# FAIREE: FAIR CLASSIFICATION WITH FINITE-SAMPLE AND DISTRIBUTION-FREE GUARANTEE

**Puheng Li**
Peking University
lphleo@pku.edu.cn

**James Zou**
Stanford University
jamesz@stanford.edu

**Linjun Zhang**
Rutgers University
linjun.zhang@rutgers.edu

## ABSTRACT

Algorithmic fairness plays an increasingly critical role in machine learning research. Several group fairness notions and algorithms have been proposed. However, the fairness guarantee of existing fair classification methods mainly depends on specific data distributional assumptions, often requiring large sample sizes, and fairness could be violated when there is a modest number of samples, which is often the case in practice. In this paper, we propose FaiREE, a fair classification algorithm that can satisfy group fairness constraints with finite-sample and distribution-free theoretical guarantees. FaiREE can be adapted to satisfy various group fairness notions (e.g., Equality of Opportunity, Equalized Odds, Demographic Parity, etc.) and achieve the optimal accuracy. These theoretical guarantees are further supported by experiments on both synthetic and real data. FaiREE is shown to have favorable performance over state-of-the-art algorithms.

## 1 INTRODUCTION

As machine learning algorithms have been increasingly used in consequential domains such as college admission Chouldechova & Roth (2018), loan application Ma et al. (2018), and disease diagnosis Fatima et al. (2017), there are emerging concerns about the algorithmic fairness in recent years. When standard machine learning algorithms are directly applied to the biased data provided by humans, the outputs are sometimes found to be biased towards certain sensitive attribute that we want to protect (race, gender, etc). To quantify the fairness in machine learning algorithms, many fairness notions have been proposed, including the individual fairness notion Biega et al. (2018), group fairness notions such as Demographic Parity, Equality of Opportunity, Predictive Parity, and Equalized Odds (Dieterich et al., 2016; Hardt et al., 2016; Gajane & Pechenizkiy, 2017; Verma & Rubin, 2018), and multi-group fairness notions including multi-calibration Hébert-Johnson et al. (2018) and multi-accuracy Kim et al. (2019). Based on these fairness notions or constraints, corresponding algorithms were designed to help satisfy the fairness constraints (Hardt et al., 2016; Pleiss et al., 2017; Zafar et al., 2017b; Krishnaswamy et al., 2021; Valera et al., 2018; Chzhen et al., 2019; Zeng et al., 2022; Thomas et al., 2019).

Among these fairness algorithms, post-processing is a popular type of algorithm which modifies the output of the model to satisfy fairness constraints. However, recent post-processing algorithms are found to lack the ability to realize accuracy–fairness trade-off and perform poorly when the sample size is limited (Hardt et al., 2016; Pleiss et al., 2017). In addition, since most fairness constraints are non-convex, some papers propose convex relaxation-based methods Zafar et al. (2017b); Krishnaswamy et al. (2021). This type of algorithms generally do not have

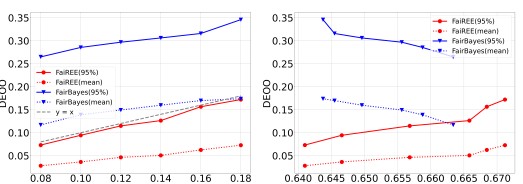

**Figure 1:** Comparison of FairBayes and FaiREE on the synthetic data with sample size = 1000. See Table 2 for detailed numerical results. Left: $DEOO$ v.s. $\alpha$, Right: DEOO v.s. Test accuracy. Here, $DEOO$ is the degree of violation to fairness constraint Equality of Opportunity and $\alpha$ is the pre-specified desired level to upper bound $DEOO$ for both methods. See Eq. (1) in Section 2 for a more detailed definition.

the theoretical guarantee of how the output satisfies the exact original fairness constraint. Another

line of research considers recalibrating the Bayes classifier by a group-dependent threshold (Valera et al., 2018; Chzhen et al., 2019; Zeng et al., 2022). However, their results require either some distributional assumptions or infinite sample size, which is hard to verify/satisfy in practice.

In this paper, we propose a post-processing algorithm FaiREE that provably achieves group fairness guarantees with only finite-sample and free of distributional assumptions (this property is also called "distribution-free" in the literature (Maritz, 1995; Clarke, 2007; Györfi et al., 2002)). To the best of our knowledge, this is the first algorithm in fair classification with a finite-sample and distribution-free guarantee. A brief pipeline of FaiREE is to first score the dataset with the given classifier, and select a candidate set based on these scores which can fit the fairness constraint with a theoretical guarantee. As there are possibly multiple classifiers that can satisfy this constraint, we further develop a distribution-free estimate of the test mis-classification error, resulting in an algorithm that produces the optimal mis-classification error given the fairness constraints. As a motivating example, Figure 1 shows that applying state-of-the-art FairBayes method in Zeng et al. (2022) on a dataset with 1000 samples results in substantial fairness violation on the test data and incorrect behavior of fairness-accuracy trade-off due to lack of fairness generalization. Our proposed FaiREE improved fairness generalization in these finite sample settings.

**Additional Related Works.** The fairness algorithms in the literature can be roughly categorized into three types: 1). Pre-processing algorithms that learn a fair representation to improve fairness (Zemel et al., 2013; Louizos et al., 2015; Lum, 2016; Adler et al., 2018; Calmon et al., 2017; Gordaliza et al., 2019; Madras et al., 2018; Kilbertus et al., 2020) 2). In-processing algorithms that optimize during training time (Calders et al., 2009; Woodworth et al., 2017; Zafar et al., 2017b;a; Agarwal et al., 2018; Russell et al., 2017; Zhang et al., 2018; Celis et al., 2019) 3). Post-processing algorithms that try to modify the output of the original method to fit fairness constraints (Kamiran et al., 2012; Feldman, 2015; Hardt et al., 2016; Fish et al., 2016; Pleiss et al., 2017; Corbett-Davies et al., 2017; Menon & Williamson, 2018; Hébert-Johnson et al., 2018; Kim et al., 2019; Deng et al., 2023).

The design of post-processing algorithms with distribution-free and finite-sample guarantees gains much attention recently due to its flexibility in practice Shafer & Vovk (2008); Romano et al. (2019), as it can be applied to any given algorithm (eg. a black-box neural network), and achieve desired theoretical guarantee with almost no assumption. One of the research areas that satisfies this property is conformal prediction (Shafer & Vovk, 2008; Lei et al., 2018; Romano et al., 2019) whose aim is to construct prediction intervals that cover a future response with high probability. In this paper, we extend this line of research beyond prediction intervals, by designing classification algorithms that satisfy certain group fairness with distribution-free and finite-sample guarantees.

**Paper Organization.** Section 2 provides the definitions and notations we use in the paper. Section 3 provides the general pipeline of FaiREE. In Section 4, we further extend the results to other fairness notions. Finally, Section 5 conducts experiments on both synthetic and real data and compares with several state-of-art algorithms to show that FaiREE has desirable performance [1].

## 2 PRELIMINARY

In this paper, we consider two types of features in classification: the standard feature $X \in \mathcal{X}$, and the sensitive attribute, which we want the output to be fair on, is denoted as $A \in \mathcal{A} = \{0, 1\}$. For the simplicity of presentation, we consider the binary classification problem with labels in $\mathcal{Y} = \{0, 1\}$. We note that our analysis can be similarly extended to the multi-class and multi-attribute setting. Under the binary classification setting, we use the score-based classifier that outputs a prediction $\widehat{Y} = \widehat{Y}(x, a) \in \{0, 1\}$ based on a score function $f(x, a) \in [0, 1]$ that depends on $X$ and $A$:

**Definition 1.** *(Score-based classifier) A score-based classifier is an indication function* $\hat{Y} = \phi(x, a) = \mathbb{1}\{f(x, a) > c\}$ *for a measurable score function* $f : \mathcal{X} \times \{0, 1\} \to [0, 1]$ *and some threshold* $c > 0$.

To address the algorithmic fairness problem, several group fairness notions have been developed in the literature. In the following, we introduce two of the popular notions, Equality of Opportunity and Equalized Odds. We will discuss other fairness notions in Section A.8 of the Appendix.

Equality of Opportunity requires comparable true positive rates across different protected groups.

---

[1] Code is available at `https://github.com/lphLeo/FaiREE`

**Definition 2.** *(Equality of Opportunity (Hardt et al., 2016)) A classifier satisfies Equality of Opportunity if it satisfies the same true positive rate among protected groups:* $\mathbb{P}_{X|A=1,Y=1}(\widehat{Y}=1) = \mathbb{P}_{X|A=0,Y=1}(\widehat{Y}=1)$.

Equalized Odds is an extension of Equality of Opportunity, requiring both false positive rate and true positive rate are similar across different attributes.

**Definition 3.** *(Equalized Odds (Hardt et al., 2016)) A classifier satisfies Equalized Odds if it satisfies the following equality:* $\mathbb{P}_{X|A=1,Y=1}(\widehat{Y}=1) = \mathbb{P}_{X|A=0,Y=1}(\widehat{Y}=1)$ *and* $\mathbb{P}_{X|A=1,Y=0}(\widehat{Y}=0) = \mathbb{P}_{X|A=0,Y=0}(\widehat{Y}=0)$.

Sometimes it is too strict to require the classifier to satisfy Equality of Opportunity or Equalized Odds exactly, which may sacrifice a lot of accuracy (as a very simple example is $f(x,a) \equiv 1$). In practice, to strike a balance between fairness and accuracy, it makes sense to relax the equality above to an inequality with a small error bound. We use the difference with respect to Equality of Opportunity, denoted by $DEOO$, to measure the disparate impact:

$$DEOO = \mathbb{P}_{X|A=1,Y=1}(\widehat{Y}=1) - \mathbb{P}_{X|A=0,Y=1}(\widehat{Y}=1). \tag{1}$$

For a classifier $\phi$, following (Zeng et al., 2022; Cho et al., 2020), $|DEOO(\phi)| \leq \alpha$ denotes an $\alpha$-tolerance fairness constraint that controls the difference between the true positive rates below $\alpha$.

Similarly, we define the following difference with Equalized Odds. Since Equalized Odds, the difference is a two-dimensional vector:

$$DEO = (\mathbb{P}_{X|A=1,Y=1}(\widehat{Y}=1) - \mathbb{P}_{X|A=0,Y=1}(\widehat{Y}=1), \mathbb{P}_{X|A=1,Y=0}(\widehat{Y}=1) - \mathbb{P}_{X|A=0,Y=0}(\widehat{Y}=1)).$$

For notational simplicity, we use the notation $\preceq$ for the element-wise comparison between vectors, that is, $DEO \preceq (\alpha_1, \alpha_2)$ if and only if $\mathbb{P}_{X|A=1,Y=1}(\widehat{Y}=1) - \mathbb{P}_{X|A=0,Y=1}(\widehat{Y}=1) \leq \alpha_1$ and $\mathbb{P}_{X|A=1,Y=0}(\widehat{Y}=1) - \mathbb{P}_{X|A=0,Y=0}(\widehat{Y}=1) \leq \alpha_2$.

**Additional Notation.** We denote the proportion of group $a$ by $p_a := \mathbb{P}(A=a)$ for $a \in \{0,1\}$; the proportion of group $Y=1$ conditioned on $A$ for $p_{Y,a} := \mathbb{P}(Y=1 \mid A=a)$; the proportion of group $Y=1$ conditioned on $A$ and $X$ for $\eta_a(x) := \mathbb{P}(Y=1 \mid A=a, X=x)$. Also, we denote by $\mathbb{P}_X(x)$ and $\mathbb{P}_{X|A=a,Y=y}(x)$ respectively the distribution function of $X$ and the distribution function of $X$ conditioned on $A$ and $Y$. The score function of standard Bayes-optimal classifier without fairness constraint is defined as $\phi^*(x,a) = \mathbb{1}\{f^*(x,a) > 1/2\}$, where $f^* \in \arg\min_f[\mathbb{P}(Y \neq \mathbb{1}\{f(x,a) > 1/2\})]$. We denote $v_{(k)}$ as the $k^{th}$ ordered value of sequence $v$ in non-decreasing order. For a set $T$, we denote $sort(T)$ as a function that returns $T$ in non-decreasing order. For a number $a \in \mathbb{R}$, we use $\lceil a \rceil$ to denote the ceiling function that maps $a$ to the least integer greater than or equal to $a$. For a positive integer $n$, we use $[n]$ to denote the set $\{1, 2, ..., n\}$.

## 3 FAIREE: A FINITE SAMPLE BASED ALGORITHM

In this section, we propose FaiREE, a general post-processing algorithm that produces a Fair classifier in a finite-sample and distribution-fREE manner, and can be applied to a wide range of group fairness notions. We will illustrate its use in Equality of Opportunity as an example in this section, and discuss more applications in later sections.

### 3.1 THE GENERAL PIPELINE OF FAIREE

Suppose we have dataset $S = S^{0,0} \cup S^{0,1} \cup S^{1,0} \cup S^{1,1}$, where $S^{y,a} = \{x_1^{y,a}, \ldots, x_{n^{y,a}}^{y,a}\}$ is the set of features associated with label $Y = y \in \{0,1\}$ and protected attribute $A = a \in \{0,1\}$. We denote the size of $S^{y,a}$ by $n^{y,a}$. Throughout the paper, we assume that $x_i^{y,a}, i \in \{1, \ldots, n^{y,a}\}$ are independently and identically distributed given $Y = y, A = a$. We define $n = n^{0,0} + n^{0,1} + n^{1,0} + n^{1,1}$ to be the total number of samples. Our goal is to post-process any given classifier to make it satisfy certain group fairness constraints.

FaiREE is a post-processing algorithm that can transform any pre-trained classification function score $f$ in order to satisfy fairness constraints. In particular, FaiREE consists of three main steps,

*scoring*, *candidate set construction*, and *candidate selection*. See Figure 2 for an illustration. We would like to note that the procedure that first chooses a candidate set of tuning parameters and then selects the best one has been commonly used in machine learning, such as in Seldonian algorithm framework to control safety and fairness Thomas et al. (2019); Giguere et al. (2022); Weber et al. (2022), the Learn then Test framework for risk control Angelopoulos et al. (2021), and in high-dimensional statistics Wang et al. (2022).

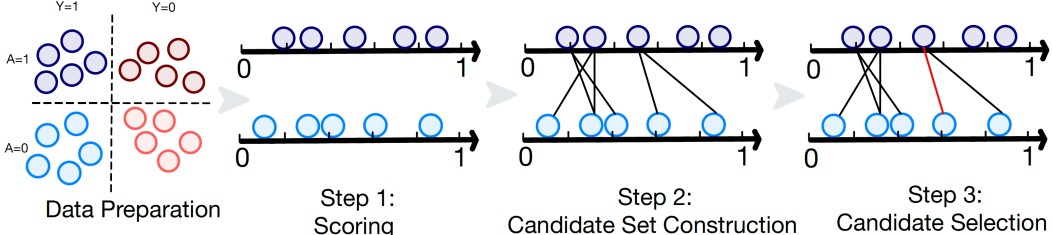

**Figure 2:** A concrete pipeline of FaiREE for Equality of Opportunity. Edges in Step 2 represent the selected candidate pair and the red edge in Step 3 represents the final optimal candidate selected from all the edges. Each pair represents two different thresholds of a single classifier.

**Step 1: Scoring.** FaiREE takes input as 1). a given fairness guarantee $\mathcal{G}$, such as Equality of Opportunity or Equalized Odds; 2). an error bound $\alpha$, which controls the violation with respect to our given fairness notion; 3). a small tolerance level $\delta$, which makes sure our final classifier satisfies our requirement with probability at least $1 - \delta$; 4). a dataset $S$.

For *scoring*, we first apply the given classifier $f$ to $S^{y,a}$ and denote the outcome $t_i^{y,a} := f(x_i^{y,a})$ as scores for each sample. These scores are then sorted within each subset in non-decreasing order respectively and obtain $T^{y,a} = \{t_{(1)}^{y,a}, \ldots, t_{(n^{y,a})}^{y,a}\}$.

**Step 2: Candidate Set Construction.** We first present a key observation for this step, which holds for many group fairness notions such as Equality of Opportunity, Equalized Odds (see details of more fairness notions in Section 3.2):

*Any classifier can fit the fairness constraint with high probability by setting the decision threshold appropriately, regardless of the data distribution.*

The insight of this observation comes from recent literature on post-processing algorithms and Neyman-Pearson classification algorithm (Fish et al., 2016; Corbett-Davies et al., 2017; Valera et al., 2018; Menon & Williamson, 2018; Tong et al., 2018; Chzhen et al., 2019). Under Equality of Opportunity, this observation is formalized in Proposition 1. We also establish similar results under other fairness notions beyond Equality of Opportunity in Section 3.2. From this observation we can build an algorithm to calculate the probability that a classifier $f$ with a certain threshold will satisfy the fairness constraint: $\text{Diff}_{\mathcal{G}}(f) \leq \alpha$, where $\text{Diff}_{\mathcal{G}}(f)$ is a generic notation to denote the violation rate of $f$ under some fairness notion $\mathcal{G}$. Then we choose the classifiers with the probability $\mathbb{P}(\text{Diff}_{\mathcal{G}}(f) > \alpha) \leq \delta$ as our candidate set $C$. This candidate set consists of a set of threshold values, with potentially different thresholds for different subpopulations.

**Step 3: Candidate Selection.** Furthermore, as there might be multiple classifiers that satisfy the given fairness constraints, we aim to choose a classifier with a small mis-classification error. To do this, FaiREE estimates the mis-classification error $err(f)$ of the classifier $f$, and chooses the one with the smallest error among the candidate set constructed in the second step.

In the rest part of the section, as an example, we consider Equality of Opportunity as our target group fairness constraint and provide our algorithm in detail.

## 3.2 APPLICATION TO EQUALITY OF OPPORTUNITY

In this section, we apply FaiREE to the fairness notion Equality of Opportunity. The following two subsections explain the steps **Candidate Set Construction** and **Candidate Selection** in detail.

### 3.2.1 CANDIDATE SET CONSTRUCTION

We first formalize our observation in the following proposition. Using the property of order statistics, the following proposition states that it is sufficient to choose the threshold of the score-based classifier from the sorted scores to control the fairness violation in a distribution-free and finite-sample manner. Here, $k^{1,a}$ is the index from which we select the threshold in $T^{1,a}$.

**Proposition 1.** *Consider* $k^{1,a} \in \{1, \ldots, n^{1,a}\}$ *for* $a \in \{0, 1\}$*, and the score-based classifier*
$$\phi(x, a) = \mathbb{1}\{f(x, a) > t^{1,a}_{(k^{1,a})}\}. \text{ Let } g_1(k, a) = \mathbb{E}[\sum_{j=k}^{n^{1,a}} \binom{n^{1,a}}{j}(Q^{1,1-a} - \alpha)^j(1 - (Q^{1,1-a} - \alpha))^{n^{1,a}-j}] \text{ with } Q^{1,a} \sim Beta(k, n^{1,a} - k + 1), \text{ then we have:}$$
$$\mathbb{P}(|DEOO(\phi)| > \alpha) \leq g_1(k^{1,1}, 1) + g_1(k^{1,0}, 0).$$

*Additionally, if* $t^{1,a}_{(k^{1,a})}$ *is a continuous random variable, the inequality above becomes tight equality.*

Here, $g_1$ is a function constructed using the property of order statistics so that $g_1(k^{1,1}, 1)$ and $g_1(k^{1,0}, 0)$ upper bound $\mathbb{P}(DEOO(\phi) > \alpha)$ and $\mathbb{P}(DEOO(\phi) < -\alpha)$ respectively. We note that $g_1$ can be efficiently compute using Monte Carlo simulations. In our experiments, we approximate $g_1$ by randomly sampling from the Beta distribution for 1000 times and achieve satisfactory approximation. This proposition ensures that the $DEOO$ of a given classifier can be controlled with high probability if we choose an appropriate threshold value when post-processing.

Based on the above proposition, we then build our classifiers for an arbitrarily given score function $f$ as below. We define $L(k^{1,0}, k^{1,1}) = g_1(k^{1,1}, 1) + g_1(k^{1,0}, 0)$. Recall that the error tolerance is $\alpha$, and $\delta$ is the tolerance level. Our candidate set is then constructed as $K = \{(k^{1,0}, k^{1,1}) \mid L(k^{1,0}, k^{1,1}) \leq \delta\}$.

Before we proceed to the theoretical guarantee for this candidate set, we introduce a bit more notation. Let us denote the size of the candidate set $K$ by $M$, and the elements in the set $K$ by $(k^{1,0}_1, k^{1,1}_1), \ldots, (k^{1,0}_M, k^{1,1}_M)$. Additionally, we let $\hat{\phi}_i(x, a) = \mathbb{1}\{f(x, a) > t^{1,a}_{(k^{1,a}_i)}\}$, for $i = 1, \ldots, M$.

To ensure that there exists at least one valid classifier (i.e. $M \geq 1$), we should have $\mathbb{E}[(Q^{1,0} - \alpha)^{n^{1,0}}] + \mathbb{E}[(Q^{1,1} - \alpha)^{n^{1,1}}] \leq \delta$, which requires a necessary and sufficient lower bound requirement on the sample size, as formulated in the following proposition:

**Theorem 1.** *If* $\min\{n^{1,0}, n^{1,1}\} \geq \lceil \frac{\log \frac{\delta}{2}}{\log(1-\alpha)} \rceil$*, for each* $i \in \{1, \ldots, M\}$ *in the candidate set, we have* $|DEOO(\hat{\phi}_i)| < \alpha$ *with probability* $1 - \delta$.

As there are at most $n^{1,0}n^{1,1}$ elements in the candidate set, the size of $K$, $M$, can be as large as $O(n^2)$. To further reduce the computational complexity, in the following part, we provide a method to shrink the candidate set.

Our construction is inspired by the following lemma, which gives the analytical form of the fair Bayes-optimal classifier under the Equality of Opportunity constraint. This Bayes-optimal classifier is defined as $\phi^*_\alpha = \arg\min_{|DEOO(\phi)| \leq \alpha} \mathbb{P}(\phi(x, a) \neq Y)$.

**Lemma 1** (Adapted from Theorem E.4 in Zeng et al. (2022))**.** *The fair Bayes-optimal classifier under Equality of Opportunity can be explicitly written as* $\phi^*_\alpha(x, a) = \mathbb{1}\{f^*(x, a) > t^*_a\}$*, then* $t^*_1 = \frac{p_1 p_{Y,1}}{2p_1 p_{Y,1} - (1/t^{1,0}_{(k)} - 2) \cdot p_0 p_{Y,0}}$.

Note that in practice, the input classifier $f$ can be the classifier trained by a classification algorithm on the training set, which means it is close to $f^*$. Thus from this observation, we can adopt a new way of building a much smaller candidate set. Note that our original candidate set is defined as : $K = \{(k^{1,0}, k^{1,1}) \mid L(k^{1,0}, k^{1,1}) \leq \delta\} = \{(k^{1,0}_1, k^{1,1}_1), \ldots, (k^{1,0}_M, k^{1,1}_M)\}$. Now, for every $1 \leq k \leq n^{1,0}$, from Lemma 1 we denote $u_1(k) = \arg\min_u |t^{1,1}_{(u)} - \frac{\hat{p}_1 \hat{p}_{Y,1}}{2\hat{p}_1 \hat{p}_{Y,1} - (1/t^{1,0}_{(k)} - 2) \cdot \hat{p}_0 \hat{p}_{Y,0}}|$, where $\hat{p}_a = \frac{n^{1,a} + n^{0,a}}{n^{0,0} + n^{0,1} + n^{1,0} + n^{1,1}}$ and $\hat{p}_{y,a} = \frac{n^{1,a}}{n^{0,a} + n^{1,a}}$. We then build our candidate set as below:

$$K' = \{(k^{1,0}, u_1(k^{1,0})) \mid L(k^{1,0}, u_1(k^{1,0})) \leq \delta\}. \tag{2}$$

This candidate set $K'$ has cardinality at most $n$. Since our next step, Candidate Selection, has computational complexity that is linear in the size of the candidate set, using the new set $K'$ would help us reduce the computational complexity from $O(n^2)$ to $O(n)$.

### 3.2.2 CANDIDATE SELECTION

In this subsection, we explain in detail how we choose the classifier with the smallest mis-classification error from the candidate set constructed in the last step. For a given pair $(k_i^{1,0}, k_i^{1,1})$ in the candidate set of index ($i \in [M]$), we need to know the rank of $t_{(k_i^{1,0})}^{1,0}$ and $t_{(k_i^{1,1})}^{1,1}$ in the sorted set $T^{0,0}$ and $T^{0,1}$ respectively in order to compute the test error where we need to consider both $y = 0$ and $1$. Specifically, we find the $k_i^{0,a}$ such that $t_{(k_i^{0,a})}^{0,a} \le t_{(k_i^{1,a})}^{1,a} < t_{(k_i^{0,a}+1)}^{0,a}$ for $a \in \{0,1\}$.

To estimate the test mis-classification error of $\hat{\phi}_i(x,a) = \mathbb{1}\{f(x,a) > t_{(k_i^{1,a})}^{1,a}\}$, we divide the error into four terms by different values of $y$ and $a$. We then estimate each part using the property of order statistics respectively, and obtain the following proposition:

**Proposition 2.** *Suppose the density functions of $f$ under $A = a, Y = 1$ are continuous. Let*
$\hat{e}_i = \frac{k_i^{1,0}}{n^{1,0}+1} \frac{n^{1,0}}{n} + \frac{k_i^{1,1}}{n^{1,1}+1} \frac{n^{1,1}}{n} + \frac{n^{0,0}+\frac{1}{2}-k_i^{0,0}}{n^{0,0}+1} \frac{n^{0,0}}{n} + \frac{n^{0,1}+\frac{1}{2}-k_i^{0,1}}{n^{0,1}+1} \frac{n^{0,1}}{n}$, *for $i = 1, 2, ..., M$. Then, there exist two constants $c_1, c_2 > 0$ such that $| \mathbb{P}(\hat{\phi}_i(x,a) \ne Y) - \hat{e}_i | \le c_1/\min(n^{0,0}, n^{0,1})$ with probability larger than $1 - c_2 \exp(-\min(n^{0,0}, n^{0,1}))$.*

The above proposition enables us to efficiently estimate the test error of the $M$ classifiers $\hat{\phi}_i$'s defined above, from which we can choose a classifier with the lowest test error $\hat{\phi}$. The algorithm is summarized in Algorithm 1. In the following, we provide the theory showing that the output of

---

**Algorithm 1:** FaiREE for Equality of Opportunity

**Input:** Training data $S = S^{0,0} \cup S^{0,1} \cup S^{1,0} \cup S^{1,1}$; the error bound $\alpha$; the tolerance level $\delta$; a given pre-trained classifier $f$

1 $T^{y,a} = \{f(x_1^{y,a}), \ldots, f(x_{n_{y,a}}^{y,a})\}$
2 $\{t_{(1)}^{y,a}, \ldots, t_{(n_{y,a})}^{y,a}\} = \text{sort}(T^{y,a})$
3 Define $g_1(k,a)$ as in Proposition 1, and let $L(k^{1,0}, k^{1,1}) = g_1(k^{1,1}, 1) + g_1(k^{1,0}, 0)$
4 Build candidate set $K'$ as in Eq. 2, and write $K' = \{(k_1^{1,0}, k_1^{1,1}), \ldots, (k_{M'}^{1,0}, k_{M'}^{1,1})\}$
5 Find $k_i^{0,0}, k_i^{0,1}$: $t_{(k_i^{0,0})}^{0,0} \le t_{(k_i^{1,0})}^{1,0} < t_{(k_i^{0,0}+1)}^{0,0}$, $t_{(k_i^{0,1})}^{0,1} \le t_{(k_i^{1,1})}^{1,1} < t_{(k_i^{0,1}+1)}^{0,1}$
6 $i_* \leftarrow \arg\min_{i \in [M']}\{\hat{e}_i\}$ ($\hat{e}_i$ is defined in Proposition 2)

**Output:** $\hat{\phi}(x,a) = \mathbb{1}\{f(x,a) > t_{(k_{i_*}^{1,a})}^{1,a}\}$

---

Algorithm 1 is approaching the optimal mis-classification error under Equality of Opportunity. The following theorem states that the final output of FaiREE has both controlled $DEOO$, and achieved almost minimum mis-classification error when the input classifier is properly chosen.

**Theorem 2.** *Given any $\alpha' < \alpha$. Set $\delta = c_0/M$ for some $c_0 > 0$, where $M$ is the candidate set size. Suppose $\min\{n^{1,0}, n^{1,1}\} \ge \lceil \frac{\log \frac{\delta}{2}}{\log(1-\alpha)} \rceil$. $\hat{\phi}$ is the output of FaiREE, then:*

*(1). $|DEOO(\hat{\phi})| < \alpha$ with probability $1 - c_0$.*

*(2). Suppose the density functions of $f$ and $f^*$ under $A = a, Y = 1$ are continuous. For any $\delta', \epsilon_0 > 0$, there exist $0 < c < 1$ and $c_1 > 0$ such that when the input classifier $f(x,a)$ satisfies $\|f - f^*\|_\infty \le \epsilon_0$ and the constructed candidate set is $K'$, we have $\mathbb{P}(\hat{\phi}(x,a) \ne Y) - \mathbb{P}(\phi_{\alpha'}^*(x,a) \ne Y) \le 2F_{(+)}^*(2\epsilon_0) + \delta'$ with probability larger than $1 - c_1 c^{\min\{n^{1,0}, n^{0,0}, n^{0,1}\}}$. ($F_{(+)}^*(x)$ is defined in Lemma 6 in the appendix.)*

Theorem 2 ensures that our classifier will approximate fair Bayes-optimal classifier if the input classifier is close to $f^*$. Here, $\alpha' < \alpha$ is any positive constant, which we adopt to promise that our candidate set is not empty.

**Remark**: We remark that FaiREE requires no assumption on data distribution except for a minimum sample size. It has the advantage over existing literature which generally imposes different assumptions to data distribution. For example, Chzhen et al. (2019) assumes $\eta(x, a)$ must surpass the level $\frac{1}{2}$ on a set of non-zero measure. Valera et al. (2018) assumes that the shifting threshold of the classifier follows the beta distribution. Also, Zeng et al. (2022)'s result only holds for population-level, and the finite-sample version is not studied.

## 4 APPLICATION TO MORE FAIRNESS NOTIONS

### 4.1 EQUALIZED ODDS

In this section, we apply our algorithm to the fairness notion Equalized Odds, which has two fairness constraints simultaneously. To ensure the two constraints, the algorithm of equalized odds should be different from Algorithm 1. We should consider all $S^{y,a}$ instead of just $S^{1,a}$ when estimating the violation to fairness constraint in the step **Candidate Set Construction**. Thus we add a function $g_0$ that deals with data with protected attribute $A = 1$, to perfect our algorithm together with $g_1$ defined in the last section.

Similar to Proposition 1, the following proposition assures that choosing an appropriate threshold during post-processing enables the high probability control of a given classifier's $DEO$.

**Proposition 3.** *Given* $k^{1,0}, k^{1,1}$ *satisfying* $k^{1,a} \in \{1, \ldots, n^{1,a}\}$ ($a = 0, 1$). *Define* $\phi(x, a) = \mathbb{1}\{f(x, a)) > t^{1,a}_{(k^{1,a})}\}$, $g_y(k, a) = \mathbb{E}[\sum_{j=k}^{n^{y,a}} \binom{n^{y,a}}{j}(Q^{y,1-a} - \alpha)^j(1 - (Q^{y,1-a} - \alpha))^{n^{y,a}-j}]$ *with* $Q^{y,a} \sim Beta(k + 1 - y, n^{y,a} - k + y)$, *then we have:*

$$\mathbb{P}(|DEO(\phi)| \preceq (\alpha, \alpha)) \geq 1 - g_1(k^{1,1}, 1) - g_1(k^{1,0}, 0) - g_0(k^{0,1}, 1) - g_0(k^{0,0}, 0).$$

Similar to Proposition 1, $g_0$ and $g_1$ jointly control the probability of $\phi$ violating the $DEO$ constraint.

---

**Algorithm 2:** FaiREE for Equalized Odds

---

**Input:** Training data $S = S^{0,0} \cup S^{0,1} \cup S^{1,0} \cup S^{1,1}$; the error bound $\alpha$; the tolerance level $\delta$; a given pre-trained classifier $f$

1 $T^{y,a} = \{f(x_1^{y,a}), \ldots, f(x_{n_{y,a}}^{y,a})\}$

2 $\{t^{y,a}_{(1)}, \ldots, t^{y,a}_{(n_{y,a})}\}$ =sort($T^{y,a}$)

3 Define $g_0(k, a)$ and $g_1(k, a)$ as in Proposition 3, $L_1(k^{1,0}, k^{1,1}) = g_1(k^{1,1}, 1) + g_1(k^{1,0}, 0)$, and $L_0(k^{0,0}, k^{0,1}) = g_0(k^{0,1}, 1) + g_0(k^{0,0}, 0)$.

4 For every $k^{1,0}, k^{1,1}$, there exists $k^{0,0}, k^{0,1}$ such that $t^{0,0}_{(k^{0,0})} \leq t^{1,0}_{(k^{1,0})} < t^{0,0}_{(k^{0,0}+1)}$, $t^{0,1}_{(k^{0,1})} \leq t^{1,1}_{(k^{1,1})} < t^{0,1}_{(k^{0,1}+1)}$.

5 Build the candidate set as $K = \{(k^{1,0}, k^{1,1}) \mid L_1(k^{1,0}, k^{1,1}) + L_0(k^{0,0}, k^{0,1}) \leq \delta\} = \{(k_1^{1,0}, k_1^{1,1}), \ldots, (k_M^{1,0}, k_M^{1,1})\}$.

6 Compute $\hat{e}_i$ as in Proposition 2), and let $i_* = \arg\min_{i \in [M]}\{\hat{e}_i\}$.

**Output:** $\hat{\phi}(x, a) = \mathbb{1}\{f(x, a) > t^{1,a}_{(k_{i_*}^{1,a})}\}$

---

Proposition 3 yields the following proposition on the $DEO$ of classifiers in the candidate set.

**Theorem 3.** *If* $\min\{n^{0,0}, n^{0,1}, n^{1,0}, n^{1,1}\} \geq \lceil \frac{\log \frac{\delta}{4}}{\log(1-\alpha)} \rceil$, *then for each* $i \in \{1, \ldots, M\}$, *we have* $|DEO(\hat{\phi}_i)| \preceq (\alpha, \alpha)$ *with probability* $1 - \delta$.

The theoretical analysis of test error is similar to the algorithm for Equality of Opportunity.

**Theorem 4.** *Given* $\alpha' < \alpha$. *Set* $\delta = c_0/M$ *for some* $c_0 > 0$, *where* $M$ *is the candidate set size.* *Suppose* $\min\{n^{0,0}, n^{0,1}, n^{1,0}, n^{1,1}\} \geq \lceil \frac{\log \frac{\delta}{4}}{\log(1-\alpha)} \rceil$. $\hat{\phi}$ *is the final output of FaiREE, then:*

*(1).* $|DEO(\hat{\phi})| \preceq (\alpha, \alpha)$ *with probability* $1 - c_0$.

*(2). Suppose the density functions of* $f^*$ *under* $A = a, Y = 1$ *are continuous. We denote* $\phi^*_{\alpha',\alpha'} = \arg\min_{|DEO(\phi)|\preceq(\alpha',\alpha')}\mathbb{P}(\phi(x, a) \neq Y)$. *For any* $\delta', \epsilon_0 > 0$, *there exist* $0 < c <$

$1$ and $c_1 > 0$ such that when the input classifier $f$ satisfies $\mid f(x, a) - f^*(x, a) \mid \leq \epsilon_0$, we have $\mathbb{P}(\hat{\phi}(x, a) \neq Y) - \mathbb{P}(\phi^*_{\alpha', \alpha'}(x, a) \neq Y) \leq 2F^*_{(+)}(2\epsilon_0) + \delta'$ with probability larger than $1 - c_1 c^{\min\{n^{1,0}, n^{1,1}, n^{0,0}, n^{0,1}\}}$. ($F^*_{(+)}(x)$ is defined in Lemma 6 in the appendix.)

## 4.2 ON COMPARING DIFFERENT FAIRNESS CONSTRAINTS

In this subsection, we further extend our algorithms to more fairness notions. The detailed technical results and derivations are deferred to Section A.8 in the appendix. Specifically, we compare the sample size requirement to make any given score function $f$ to achieve certain fairness constraint. We note that our algorithm is almost assumption-free, except for the $i.i.d.$ assumption and a necessary and sufficient condition of the sample size. Therefore, we make a chart below to recommend different fairness notions used in practice when the sample sizes are limited. We summarize our results in the following table:

**Table 1:** Sample complexity requirements for FaiREE to achieve different fairness constraints. We consider the following fairness notions: DP (Demographic Parity), EOO (Equality of Opportunity), EO (Equalized Odds), PE (Predictive Equality), EA (Equalized Accuracy), and $n^a = n^{0,a} + n^{1,a}$.

| DP | EOO | PE | EO | EA |
|---|---|---|---|---|
| $n^a \geq \lceil \frac{\log\frac{\delta}{2}}{\log(1-\alpha)} \rceil$ | $n^{1,a} \geq \lceil \frac{\log\frac{\delta}{2}}{\log(1-\alpha)} \rceil$ | $n^{0,a} \geq \lceil \frac{\log\frac{\delta}{2}}{\log(1-\alpha)} \rceil$ | $n^{y,a} \geq \lceil \frac{\log\frac{\delta}{4}}{\log(1-\alpha)} \rceil$ | $n^{y,a} \geq \lceil \frac{\log\frac{\delta}{4}}{\log(\frac{1-y+(2y-1)p_{Y,\mid y-a\mid}-\alpha}{y(2p_{Y,a}-1)+1-p_{Y,a}})} \rceil$ |

From this table, we find that Demographic Parity requires the least sample size, Equality of Opportunity and Predictive Equality need a lightly larger sample size, and Equalized Odds is the notion that requires the largest sample size among the first four fairness notions. The sample size requirement for Equalized Accuracy is similar to that of Equalized Odds, but does not have a strict dominance.

## 5 EXPERIMENTS

In this section, we conduct experiments to test and understand the effectiveness of FaiREE. For both synthetic data and real data analysis, we compare FaiREE with the following representative methods for fair classification: Reject-Option-Classification (ROC) method in Kamiran et al. (2012), Eqodds-Postprocessing (Eq) method in Hardt et al. (2016), Calibrated Eqodds-Postprocessing (C-Eq) method in Pleiss et al. (2017) and FairBayes method in Zeng et al. (2022).The first three baselines are designed to cope with Equalized Odds and the last one is for Equality of Opportunity.

## 5.1 SYNTHETIC DATA

To show the distribution-free and finite-sample guarantee of FaiREE, we generate the synthetic data from mixed distributions. Real world data are generally heavy-tailed (Resnick (1997)). Thus, we consider the following models with various heavy-tailed distributions for generating synthetic data:

**Model 1.** We generate the protected attribute $A$ and label $Y$ with probability $p_1 = \mathbb{P}(A = 1) = 0.7, p_0 = \mathbb{P}(A = 0) = 0.3, p_{y,1} = \mathbb{P}(Y = y \mid A = 1) = 0.7$ and $p_{y,0} = \mathbb{P}(Y = y \mid A = 0) = 0.4$ for $y \in \{0, 1\}$. The dimension of features is set to 60, and we generate features with $x_{i,j}^{0,0} \overset{i.i.d.}{\sim} t(3)$, where $t(k)$ denotes the $t$-distribution with degree of freedom $k$, $x_{i,j}^{0,1} \overset{i.i.d.}{\sim} \chi_1^2$, $x_{i,j}^{1,0} \overset{i.i.d.}{\sim} \chi_3^2$ and $x_{i,j}^{1,1} \overset{i.i.d.}{\sim} N(\mu, 1)$, where $\mu \sim U(0, 1)$ and the scale parameter is fixed to be 1, for $j = 1, 2, ..., 60$.

**Model 2.** We generate the protected attribute $A$ and label $Y$ with the probability, location parameter and scale parameter the same as Model 1. The dimension of features is set to 80, and we generate features with $x_{i,j}^{0,0} \overset{i.i.d.}{\sim} t(4)$, $x_{i,j}^{0,1} \overset{i.i.d.}{\sim} \chi_2^2$, $x_{i,j}^{1,0} \overset{i.i.d.}{\sim} \chi_4^2$ and $x_{i,j}^{1,1} \overset{i.i.d.}{\sim} Laplace(\mu, 1)$, for $j = 1, 2, ..., 80$.

For each model, we generate 1000 $i.i.d.$ samples, and the experimental results are summarized in Tables 2 and 3.

From these two tables, we find that our proposed FaiREE, when applied to different fairness notions Equality of Opportunity and Equality of Opportunity, is able to control the required fairness vio-

**Table 2:** Experimental studies under Model 1. Here $\overline{|DEOO|}$ denotes the sample average of the absolute value of $DEOO$ defined in Eq. (1), and $|DEOO|_{95}$ denotes the sample upper 95% quantile. $\overline{|DPE|}$ and $|DPE|_{95}$ are defined similarly for $DPE$ defined in Eq. (13). $\overline{ACC}$ is the sample average of accuracy. We use "/" in the $DPE$ line because FairBayes and FaiREE-EOO are not designed to control $DPE$.

| | Eq | C-Eq | ROC | FairBayes | | | FaiREE-EOO | | | FaiREE-EO | | |
|---|---|---|---|---|---|---|---|---|---|---|---|---|
| $\alpha$ | / | / | / | 0.08 | 0.12 | 0.16 | 0.08 | 0.12 | 0.16 | 0.08 | 0.12 | 0.16 |
| $\overline{|DEOO|}$ | 0.061 | 0.132 | 0.255 | 0.117 | 0.149 | 0.170 | 0.028 | 0.046 | 0.063 | 0.025 | 0.031 | 0.042 |
| $|DEOO|_{95}$ | 0.146 | 0.307 | 0.500 | 0.265 | 0.297 | 0.316 | 0.073 | 0.115 | 0.157 | 0.079 | 0.108 | 0.133 |
| $\overline{|DPE|}$ | 0.051 | 0.029 | 0.511 | / | / | / | / | / | / | 0.039 | 0.042 | 0.045 |
| $|DPE|_{95}$ | 0.110 | 0.091 | 0.850 | / | / | / | / | / | / | 0.075 | 0.084 | 0.106 |
| $\overline{ACC}$ | 0.472 | 0.606 | 0.637 | 0.663 | 0.656 | 0.646 | 0.621 | 0.657 | 0.669 | 0.552 | 0.562 | 0.615 |

**Table 3:** Experimental studies under Model 2, with the same notation as Table 2.

| | Eq | C-Eq | ROC | FairBayes | | | FaiREE-EOO | | | FaiREE-EO | | |
|---|---|---|---|---|---|---|---|---|---|---|---|---|
| $\alpha$ | / | / | / | 0.08 | 0.12 | 0.16 | 0.08 | 0.12 | 0.16 | 0.08 | 0.12 | 0.16 |
| $\overline{|DEOO|}$ | 0.063 | 0.105 | 0.237 | 0.251 | 0.320 | 0.324 | 0.027 | 0.047 | 0.073 | 0.028 | 0.035 | 0.047 |
| $|DEOO|_{95}$ | 0.080 | 0.137 | 0.502 | 0.676 | 0.742 | 0.765 | 0.075 | 0.112 | 0.153 | 0.077 | 0.114 | 0.143 |
| $\overline{|DPE|}$ | 0.043 | 0.107 | 0.209 | / | / | / | / | / | / | 0.041 | 0.044 | 0.056 |
| $|DPE|_{95}$ | 0.066 | 0.144 | 0.443 | / | / | / | / | / | / | 0.071 | 0.090 | 0.127 |
| $\overline{ACC}$ | 0.380 | 0.600 | 0.616 | 0.606 | 0.598 | 0.589 | 0.595 | 0.627 | 0.639 | 0.575 | 0.589 | 0.606 |

lation respectively with high probability, while all the other methods cannot. In addition, although satisfying stronger constraints, the mis-classification error FaiREE is comparable to, and sometimes better than the state-of-the-art methods.

## 5.2 REAL DATA ANALYSIS

In this section, we apply FaiREE to a real data set, Adult Census dataset (Dua et al., 2017), whose task is to predict whether a person's income is greater than $50,000. The protected attribute is gender, and the sample size is 45,222, including 32561 training samples and 12661 test samples. To facilitate the numerical study, we randomly split data into training set, calibration set and test set at each repetition and repeat for 500 times. FaiREE is compared the existing methods described in the last subsection. Again, as shown in Table 4, the proposed FaiREE method controls the fairness constraints at the desired level, and achieve small mis-classification error. More implementation details and experiments on other benchmark datasets are presented in A.9.

**Table 4:** Result of different methods on Adult Census dataset

| | Eq | C-Eq | ROC | FairBayes | | | FaiREE-EOO | | | FaiREE-EO | | |
|---|---|---|---|---|---|---|---|---|---|---|---|---|
| $\alpha$ | / | / | / | 0.07 | 0.1 | 0.14 | 0.07 | 0.1 | 0.14 | 0.07 | 0.1 | 0.14 |
| $\overline{|DEOO|}$ | 0.043 | 0.087 | 0.031 | 0.107 | 0.101 | 0.104 | 0.034 | 0.039 | 0.066 | 0.002 | 0.039 | 0.067 |
| $|DEOO|_{95}$ | 0.112 | 0.154 | 0.097 | 0.140 | 0.153 | 0.153 | 0.065 | 0.090 | 0.124 | 0.008 | 0.094 | 0.125 |
| $\overline{|DPE|}$ | 0.027 | 0.048 | 0.044 | / | / | / | / | / | / | 0.030 | 0.066 | 0.074 |
| $|DPE|_{95}$ | 0.058 | 0.105 | 0.117 | / | / | / | / | / | / | 0.056 | 0.078 | 0.086 |
| $\overline{ACC}$ | 0.815 | 0.823 | 0.691 | 0.847 | 0.847 | 0.847 | 0.845 | 0.846 | 0.847 | 0.512 | 0.845 | 0.846 |

## 6 CONCLUSION AND DISCUSSION

In this paper, we propose FaiREE, a post-processing algorithm for fair classification with theoretical guarantees in a finite-sample and distribution-free manner. FaiREE can be applied to a wide range of group fairness notions and is shown to achieve small mis-classification error while satisfying the fairness constraints. Numerical studies on both synthetic and real data show the practical value of FaiREE in achieving a superior fairness-accuracy trade-off than the state-of-the-art methods. One interesting direction of future work is to extend the FaiREE techniques to multi-group fairness notions such as multi-calibration (Hébert-Johnson et al., 2018) and multi-accuracy Kim et al. (2019).

ACKNOWLEDGEMENTS

The research of Linjun Zhang is partially supported by NSF DMS-2015378. The research of James Zou is partially supported by funding from NSF CAREER and the Sloan Fellowship.

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

## A    APPENDIX

### A.1    PROOF OF PROPOSITION 1

*Proof.* The classifier is
$$\phi = \begin{cases} \mathbb{1}\{f(x,0) > t^{1,0}_{(k^{1,0})}\}, a = 0 \\ \mathbb{1}\{f(x,1) > t^{1,1}_{(k^{1,1})}\}, a = 1 \end{cases}$$

we have:

$$
\begin{aligned}
|DEOO(\phi)| &= |\mathbb{P}(\hat{Y} = 1 \mid A = 0, Y = 1) - \mathbb{P}(\hat{Y} = 1 \mid A = 1, Y = 1)| \\
&= |\mathbb{P}(f(x,0) > t^{1,0}_{(k^{1,0})} \mid A = 0, Y = 1) - \mathbb{P}(f(x,1) > t^{1,1}_{(k^{1,1})} \mid A = 1, Y = 1)| \\
&= |1 - F^{1,0}(t^{1,0}_{(k^{1,0})}) - [1 - F^{1,1}(t^{1,1}_{(k^{1,1})})]| \\
&= |F^{1,1}(t^{1,1}_{(k^{1,1})}) - F^{1,0}(t^{1,0}_{(k^{1,0})})|
\end{aligned}
$$

Hence,

$$
\begin{aligned}
\mathbb{P}(|DEOO(\phi)| > \alpha) &= \mathbb{P}(|F^{1,1}(t^{1,1}_{(k^{1,1})}) - F^{1,0}(t^{1,0}_{(k^{1,0})})| > \alpha) \\
&= \mathbb{P}(F^{1,1}(t^{1,1}_{(k^{1,1})}) - F^{1,0}(t^{1,0}_{(k^{1,0})}) > \alpha) + \mathbb{P}(F^{1,1}(t^{1,1}_{(k^{1,1})}) - F^{1,0}(t^{1,0}_{(k^{1,0})}) < -\alpha) \\
&\triangleq A + B.
\end{aligned}
$$

We then have

$$
\begin{aligned}
A &= \mathbb{P}(F^{1,1}(t^{1,1}_{(k^{1,1})}) - F^{1,0}(t^{1,0}_{(k^{1,0})}) > \alpha) \\
&= \mathbb{P}(F^{1,0}(t^{1,0}_{(k^{1,0})}) < F^{1,1}(t^{1,1}_{(k^{1,1})}) - \alpha) \\
&\leq \mathbb{E}[\mathbb{P}(t^{1,0}_{(k^{1,0})} < F^{1,0^{-1}}(F^{1,1}(t^{1,1}_{(k^{1,1})}) - \alpha))\mathbb{1}\{F^{1,1}(t^{1,1}_{(k^{1,1})}) - \alpha > 0\} \mid t^{1,1}_{(k^{1,1})}] \\
&= \mathbb{E}\{\mathbb{P}[\text{at least } k^{1,0} \text{ of } t^{1,0}\text{'s are less than } F^{1,0^{-1}}(F^{1,1}(t^{1,1}_{(k^{1,1})}) - \alpha)]\mathbb{1}\{F^{1,1}(t^{1,1}_{(k^{1,1})}) - \alpha > 0\} \mid t^{1,1}_{(k^{1,1})}\}
\end{aligned}
$$

Following this, we obtain

$$
\begin{aligned}
A &\leq \mathbb{E}\{\sum_{j=k^{1,0}}^{n^{1,0}} \mathbb{P}[\text{exactly j of the } t^{1,0}\text{'s are less than } F^{1,0^{-1}}(F^{1,1}(t^{1,1}_{(k^{1,1})}) - \alpha)]\mathbb{1}\{F^{1,1}(t^{1,1}_{(k^{1,1})}) - \alpha > 0\} \mid t^{1,1}_{(k^{1,1})}\} \\
&= \mathbb{E}\{\sum_{j=k^{1,0}}^{n^{1,0}} \binom{n^{1,0}}{j} \mathbb{P}[t^{1,0} < F^{1,0^{-1}}(F^{1,1}(t^{1,1}_{(k^{1,1})}) - \alpha)]^j (1 - \mathbb{P}[t^{1,0} < F^{1,0^{-1}}(F^{1,1}(t^{1,1}_{(k^{1,1})}) - \alpha)])^{n^{1,0}-j} \\
&\quad \mathbb{1}\{F^{1,1}(t^{1,1}_{(k^{1,1})}) - \alpha > 0\} \mid t^{1,1}_{(k^{1,1})}\} \\
&\leq \mathbb{E}[\sum_{j=k^{1,0}}^{n^{1,0}} \binom{n^{1,0}}{j} (F^{1,1}(t^{1,1}_{(k^{1,1})}) - \alpha)^j (1 - (F^{1,1}(t^{1,1}_{(k^{1,1})}) - \alpha))^{n^{1,0}-j} \mid t^{1,1}_{(k^{1,1})}]
\end{aligned}
$$

Similarly, we have

$$
B \leq \mathbb{E}[\sum_{j=k^{1,1}}^{n^{1,1}} \binom{n^{1,1}}{j} (F^{1,0}(t^{1,0}_{(k^{1,0})}) - \alpha)^j (1 - (F^{1,0}(t^{1,0}_{(k^{1,0})}) - \alpha))^{n^{1,1}-j} \mid t^{1,0}_{(k^{1,0})}]
$$

Hence, we have

$$
\begin{aligned}
A + B \leq & \mathbb{E}[\sum_{j=k^{1,0}}^{n^{1,0}} \binom{n^{1,0}}{j} (F^{1,1}(t^{1,1}_{(k^{1,1})}) - \alpha)^j (1 - (F^{1,1}(t^{1,1}_{(k^{1,1})}) - \alpha))^{n^{1,0}-j} \mid t^{1,1}_{(k^{1,1})}] \\
& + \mathbb{E}[\sum_{j=k^{1,1}}^{n^{1,1}} \binom{n^{1,1}}{j} (F^{1,0}(t^{1,0}_{(k^{1,0})}) - \alpha)^j (1 - (F^{1,0}(t^{1,0}_{(k^{1,0})}) - \alpha))^{n^{1,1}-j} \mid t^{1,0}_{(k^{1,0})}] \\
\leq & \mathbb{E}[\sum_{j=k^{1,0}}^{n^{1,0}} \binom{n^{1,0}}{j} (Q^{1,1} - \alpha)^j (1 - (Q^{1,1} - \alpha))^{n^{1,0}-j}] \\
& + \mathbb{E}[\sum_{j=k^{1,1}}^{n^{1,1}} \binom{n^{1,1}}{j} (Q^{1,0} - \alpha)^j (1 - (Q^{1,0} - \alpha))^{n^{1,1}-j}]
\end{aligned}
$$

The last inequality holds because $F^{1,a}(t^{1,a}_{(k^{1,a})})$ is stochastically dominated by $Beta(k^{1,a}, n^{1,a} - k^{1,a} + 1)$.

If $t^{1,a}$ is continuous random variable, the equality holds.

Now we complete the proof. □

## A.2 PROOF OF LEMMA 1

We first introduce the lemma (theorem E.4 in Zeng et al. (2022)):

**Lemma 2.** *(Fair Bayes-optimal Classifiers under Equality of Opportunity). Let $E^\star = \text{DEOO}(f^\star)$. For any $\alpha > 0$, all fair Bayes-optimal classifiers $f^\star_{E,\alpha}$ under the fairness constraint $|\text{DEOO}(f)| \leq \alpha$ are given as follows:*
*- When $|E^\star| \leq \alpha$, $f^\star_{E,\alpha} = f^\star$*
*- When $|E^\star| > \alpha$, suppose $\mathbb{P}_{X|A=1,Y=1}\left(\eta_1(X) = \frac{p_1 p_{Y,1}}{2\left(p_1 p_{Y,1} - t^\star_{E,\alpha}\right)}\right) = 0$, then for all $x \in \mathcal{X}$ and $a \in \mathcal{A}$,*

$$
f^\star_{E,\alpha}(x,a) = I\left(\eta_a(x) > \frac{p_a p_{Y,a}}{2p_a p_{Y,a} + (1-2a)t^\star_{E,\alpha}}\right)
$$

*where $t^\star_{E,\alpha}$ is defined as*

$$
t^\star_{E,\alpha} = \sup\left\{t : \mathbb{P}_{Y|A=1,Y=1}\left(\eta_1(X) > \frac{p_1 p_{Y,1}}{2p_1 p_{Y,1} - t}\right) > \mathbb{P}_{Y|A=0,Y=1}\left(\eta_0(X) > \frac{p_0 p_{Y,0}}{2p_0 p_{Y,0} + t}\right) + \frac{E^\star}{|E^\star|}\alpha\right\}.
$$

Now we come back to prove Proposition 1.

*Proof.* From Lemma 2, we have

$$
t_0^* = \frac{p_0 p_{Y,0}}{2p_0 p_{Y,0} + t^*_{E,\alpha}} \tag{3}
$$

$$
t_1^* = \frac{p_1 p_{Y,1}}{2p_1 p_{Y,1} - t^*_{E,\alpha}} \tag{4}
$$

Combine Eq. (3) and (4) together and we complete the proof. □

## A.3 PROOF OF PROPOSITION 2

We first provide a lemma for the mis-classification error of the classifier in the candidate set.

**Lemma 3.** $\mid \mathbb{P}(\hat{\phi}_i(x,a) \neq Y) - [\frac{k_i^{1,0}}{n^{1,0}+1}p_0 p_{Y,0} + \frac{k_i^{1,1}}{n^{1,1}+1}p_1 p_{Y,1} + \frac{n^{0,0}+\frac{1}{2}-\mathbb{E}(k_i^{0,0})}{n^{0,0}+1}p_0(1-p_{Y,0}) +$
$\frac{n^{0,1}+\frac{1}{2}-\mathbb{E}(k_i^{0,1})}{n^{0,1}+1}p_1(1-p_{Y,1})] \mid \leq \frac{p_0(1-p_{Y,0})}{2(n^{0,0}+1)} + \frac{p_1(1-p_{Y,1})}{2(n^{0,1}+1)}$

We also have the following lemma:

**Lemma 4.** $F^{0,0}(t^{0,0}_{(k^{1,0})}) \sim Beta(k^{0,0}, n^{0,0} - k^{0,0} + 1)$, $F^{0,1}(t^{0,1}_{(k^{0,1})}) \sim Beta(k^{0,1}, n^{0,1} - k^{0,1} + 1)$.

*Proof of Lemma 4.* Since $F^{0,0}, F^{0,1}$ are the continuous cumulative distribution functions of the $t^{0,0}$'s and $t^{0,1}$'s, we have $F^{0,0}(t^{0,0}), F^{0,1}(t^{0,1}) \sim U(0,1)$, thus $F^{0,0}(t^{0,0}_{(k^{0,0})})$ is the $k^{0,0^{th}}$ order statistic of $n^{0,0}$ i.i.d samples from $U(0,1)$ and $F^{0,1}(t^{0,1}_{(k^{0,1})})$ is the $k^{0,1^{th}}$ order statistic of $n^{0,1}$ i.i.d samples from $U(0,1)$.

Thus, from the well known fact of the ordered statistics, we have $F^{0,0}(t^{0,0}_{(k^{0,0})}) \sim Beta(k^{0,0}, n^{0,0} - k^{0,0} + 1)$ and $F^{0,1}(t^{0,1}_{(k^{0,1})}) \sim Beta(k^{0,1}, n^{0,1} - k^{0,1} + 1)$.

$\square$

Now we come back to the proof of Lemma 3:

*Proof of Lemma 3.* The classifier is:

$$\hat{\phi} = \begin{cases} \mathbb{1}\{f(x,0) > t^{1,0}_{(k^{1,0})}\}, A = 0 \\ \mathbb{1}\{f(x,1) > t^{1,1}_{(k^{1,1})}\}, A = 1 \end{cases}$$

So we have the mis-classification error:

$$
\begin{aligned}
\mathbb{P}(Y \neq \hat{Y}) &= \mathbb{P}(Y = 1, \hat{Y} = 0) + \mathbb{P}(Y = 0, \hat{Y} = 1) \\
&= \mathbb{P}(Y = 1, \hat{Y} = 0, A = 0) + \mathbb{P}(Y = 1, \hat{Y} = 0, A = 1) \\
&\quad + \mathbb{P}(Y = 0, \hat{Y} = 1, A = 0) + \mathbb{P}(Y = 0, \hat{Y} = 1, A = 1) \\
&= \mathbb{P}(\hat{Y} = 0 | Y = 1, A = 0)\mathbb{P}(Y = 1, A = 0) + \mathbb{P}(\hat{Y} = 0 | Y = 1, A = 1)\mathbb{P}(Y = 1, A = 1) \\
&\quad + \mathbb{P}(\hat{Y} = 1 | Y = 0, A = 0)\mathbb{P}(Y = 0, A = 0) + \mathbb{P}(\hat{Y} = 1 | Y = 0, A = 1)\mathbb{P}(Y = 0, A = 1) \\
&= \mathbb{E}[\mathbb{P}(f(x,0) \leq t^{1,0}_{(k^{1,0})} | Y = 1, A = 0) | t^{1,0}_{(k^{1,0})}]p_0 p_{Y,0} \\
&\quad + \mathbb{E}[\mathbb{P}(f(x,1) \leq t^{1,1}_{(k^{1,1})} | Y = 1, A = 1) | t^{1,1}_{(k^{1,1})}]p_1 p_{Y,1} \\
&\quad + \mathbb{E}[\mathbb{P}(f(x,0) \geq t^{1,0}_{(k^{1,0})} | Y = 0, A = 0) | t^{1,0}_{(k^{1,0})}]p_0 (1 - p_{Y,0}) \\
&\quad + \mathbb{E}[\mathbb{P}(f(x,1) \geq t^{1,1}_{(k^{1,1})} | Y = 0, A = 1) | t^{1,1}_{(k^{1,1})}]p_1 (1 - p_{Y,1}) \\
&\leq \mathbb{E}[\mathbb{P}(f(x,0) \leq t^{1,0}_{(k^{1,0})} | Y = 1, A = 0) | t^{1,0}_{(k^{1,0})}]p_0 p_{Y,0} \\
&\quad + \mathbb{E}[\mathbb{P}(f(x,1) \leq t^{1,1}_{(k^{1,1})} | Y = 1, A = 1) | t^{1,1}_{(k^{1,1})}]p_1 p_{Y,1} \\
&\quad + \mathbb{E}[\mathbb{P}(f(x,0) \geq t^{0,0}_{(k^{0,0})} | Y = 0, A = 0) | t^{1,0}_{(k^{1,0})}]p_0 (1 - p_{Y,0}) \\
&\quad + \mathbb{E}[\mathbb{P}(f(x,1) \geq t^{0,1}_{(k^{0,1})} | Y = 0, A = 1) | t^{1,1}_{(k^{1,1})}]p_1 (1 - p_{Y,1}) \\
&= \mathbb{E}[F^{1,0}(t^{1,0}_{(k^{1,0})}) | t^{1,0}_{(k^{1,0})}]p_0 p_{Y,0} + \mathbb{E}[F^{1,1}(t^{1,1}_{(k^{1,1})}) | t^{1,1}_{(k^{1,1})}]p_1 p_{Y,1} \\
&\quad + \mathbb{E}[1 - \mathbb{E}[F^{0,0}(t^{0,0}_{(k^{0,0})}) | t^{0,0}_{(k^{0,0})}] | t^{1,0}_{(k^{1,0})}]p_0(1 - p_{Y,0}) + \mathbb{E}[1 - \mathbb{E}[F^{0,1}(t^{0,1}_{(k^{0,1})}) | t^{0,1}_{(k^{0,1})}] | t^{1,1}_{(k^{1,1})}]p_1(1 - p_{Y,1}) \\
&= \frac{k_i^{1,0}}{n^{1,0}+1}p_0 p_{Y,0} + \frac{k_i^{1,1}}{n^{1,1}+1}p_1 p_{Y,1} + \frac{n^{0,0}+1-\mathbb{E}(k_i^{0,0})}{n^{0,0}+1}p_0(1 - p_{Y,0}) + \frac{n^{0,1}+1-\mathbb{E}(k_i^{0,1})}{n^{0,1}+1}p_1(1 - p_{Y,1})
\end{aligned}
$$

The last equality comes from Lemma 4, and from the fact that $\mathbb{E}(Beta(\alpha, \beta)) = \frac{\alpha}{\alpha+\beta}$.

Similarly, we have

$$\mathbb{P}(Y \neq \hat{Y}) \geq \frac{k_i^{1,0}}{n^{1,0}+1}p_0 p_{Y,0} + \frac{k_i^{1,1}}{n^{1,1}+1}p_1 p_{Y,1} + \frac{n^{0,0} - \mathbb{E}(k_i^{0,0})}{n^{0,0}+1}p_0(1 - p_{Y,0}) + \frac{n^{0,1} - \mathbb{E}(k_i^{0,1})}{n^{0,1}+1}p_1(1 - p_{Y,1})$$

Thus, we have $| \mathbb{P}(\hat{\phi}_i(x,a) \neq Y) - [\frac{k_i^{1,0}}{n^{1,0}+1}p_0 p_{Y,0} + \frac{k_i^{1,1}}{n^{1,1}+1}p_1 p_{Y,1} + \frac{n^{0,0}+\frac{1}{2}-\mathbb{E}(k_i^{0,0})}{n^{0,0}+1}p_0(1-p_{Y,0}) + \frac{n^{0,1}+\frac{1}{2}-\mathbb{E}(k_i^{0,1})}{n^{0,1}+1}p_1(1-p_{Y,1})] | \leq \frac{p_0(1-p_{Y,0})}{2(n^{0,0}+1)} + \frac{p_1(1-p_{Y,1})}{2(n^{0,1}+1)}$.

Now we complete the proof of Lemma 3.

$\square$

Next, we come to prove Proposition 2.

**Lemma 5** (Hoeffding's inequality). *Let $X_1, \ldots, X_n$ be independent random variables. Assume that $X_i \in [m_i, M_i]$ for every $i$. Then, for any $t > 0$, we have*

$$\mathbb{P}\left\{\sum_{i=1}^{n}(X_i - \mathbb{E}X_i) \geq t\right\} \leq e^{-\frac{2t^2}{\sum_{i=1}^{n}(M_i-m_i)^2}}$$

*Proof of Proposition 2.* First, we notice that $k_i^{0,a}$ is the number of $t^{0,a}$'s such that $t^{0,a} < t_{(k_i^{1,a})}^{1,a}$, i.e.

$$k_i^{0,a} = \sum_{j=1}^{n^{0,1}} \mathbb{1}\{t_j^{0,a} < t_{(k_i^{1,a})}^{1,a}\}.$$

Thus, for a given $\epsilon > 0$, from Hoeffding's inequality, we have with probability $1 - e^{-2n^{0,a}\epsilon^2}$,

$$\frac{k_i^{0,a} - \mathbb{E}(k_i^{0,a})}{n^{0,a}} = \frac{\sum\limits_{j=1}^{n^{0,1}} \mathbb{1}\{t_j^{0,a} < t_{(k_i^{1,a})}^{1,a}\} - \sum\limits_{j=1}^{n^{0,1}} \mathbb{E}(\mathbb{1}\{t_j^{0,a} < t_{(k_i^{1,a})}^{1,a}\})}{n^{0,a}} \leq \epsilon.$$

Similarly, with probability $1 - e^{-2n^{0,a}\epsilon^2}$,

$$\frac{k_i^{0,a} - \mathbb{E}(k_i^{0,a})}{n^{0,a}} = \frac{\sum\limits_{j=1}^{n^{0,1}} \mathbb{1}\{t_j^{0,a} < t_{(k_i^{1,a})}^{1,a}\} - \sum\limits_{j=1}^{n^{0,1}} \mathbb{E}(\mathbb{1}\{t_j^{0,a} < t_{(k_i^{1,a})}^{1,a}\})}{n^{0,a}} \geq -\epsilon.$$

Thus, $| \frac{k_i^{0,a}-\mathbb{E}(k_i^{0,a})}{n^{0,a}} | \leq \epsilon$ with probability $1 - 2e^{-2n^{0,a}\epsilon^2}$.

Then we estimate $p_a$ and $p_{Y,a}$ by $\hat{p}_a = \frac{n^{1,a}+n^{0,a}}{n}$ and $\hat{p}_{Y,a} = \frac{n^{Y,a}}{n}$ ($n$ is the number of the total samples). Here, $\frac{n^{1,a}+n^{0,a}}{n} = \frac{\sum\limits_{i=1}^{n} \mathbb{1}\{Z_i^a=1\}}{n}$ and $\frac{n^{Y,a}}{n} = \frac{\sum\limits_{i=1}^{n} \mathbb{1}\{Z_i^{Y,a}=1\}}{n}$, where $Z_i^a \sim B(1,p_a)$ and $Z_i^{Y,a} \sim B(1,p_{Y,a})$. From Hoeffding's inequality (Lemma 5), we have:

$$\mathbb{P}(| \hat{p}_a - p_a | \geq \sqrt{\frac{n^{0,a}}{n}}\epsilon) \leq 2e^{-2n^{0,a}\epsilon^2},$$

$$\mathbb{P}(| \hat{p}_{Y,a} - p_{Y,a} | \geq \sqrt{\frac{n^{0,a}}{n}}\epsilon) \leq 2e^{-2n^{0,a}\epsilon^2}$$

Thus, with probability $1 - 6e^{-2n^{0,a}\epsilon^2}$, we have:

$$\begin{cases} | \hat{p}_a - p_a | \leq \sqrt{\frac{n^{0,a}}{n}}\epsilon \\ | \hat{p}_{Y,a} - p_{Y,a} | \leq \sqrt{\frac{n^{0,a}}{n}}\epsilon \\ | \frac{k_i^{0,a} - \mathbb{E}(k_i^{0,a})}{n^{0,a}} | \leq \epsilon \end{cases}$$

Hence, we have with probability $1 - 6(e^{-2n^{0,0}\epsilon^2} + e^{-2n^{0,1}\epsilon^2})$,

$$|\mathbb{P}(\hat{\phi}_i(x,a) \neq Y) - \hat{\mathbb{P}}(\hat{\phi}_i(x,a) \neq Y)|$$

$$\leq |\frac{k_i^{1,0}}{n^{1,0}+1}p_0 p_{Y,0} + \frac{k_i^{1,1}}{n^{1,1}+1}p_1 p_{Y,1} + \frac{n^{0,0}+0.5-\mathbb{E}(k_i^{0,0})}{n^{0,0}+1}p_0(1-p_{Y,0}) + \frac{n^{0,1}+0.5-\mathbb{E}(k_i^{0,1})}{n^{0,1}+1}p_1(1-p_{Y,1})$$

$$- [\frac{k_i^{1,0}}{n^{1,0}+1}\hat{p}_0 \hat{p}_{Y,0} + \frac{k_i^{1,1}}{n^{1,1}+1}\hat{p}_1 \hat{p}_{Y,1} + \frac{n^{0,0}+0.5-k_i^{0,0}}{n^{0,0}+1}\hat{p}_0(1-\hat{p}_{Y,0}) + \frac{n^{0,1}+0.5-k_i^{0,1}}{n^{0,1}+1}\hat{p}_1(1-\hat{p}_{Y,1})]|$$

$$+ \frac{p_0(1-p_{Y,0})}{2(n^{0,0}+1)} + \frac{p_1(1-p_{Y,1})}{2(n^{0,1}+1)}$$

$$\leq \epsilon[\sqrt{\frac{n^{0,0}}{n}}\frac{k_i^{1,0}}{n^{1,0}+1}(p_0+p_{Y,0}) + \sqrt{\frac{n^{0,1}}{n}}\frac{k_i^{1,1}}{n^{1,1}+1}(p_1+p_{Y,1})] + \epsilon^2(\frac{n^{0,0}}{n}\frac{k_i^{1,0}}{n^{1,0}+1} + \frac{n^{0,1}}{n}\frac{k_i^{1,1}}{n^{1,1}+1})$$

$$+ \epsilon[\frac{n^{0,0}}{n^{0,0}+1}[p_0+p_0 p_{Y,0} + \sqrt{\frac{n^{0,0}}{n}}\epsilon(\sqrt{\frac{n^{0,0}}{n}}\epsilon + p_0 + p_{Y,0}+1)]$$

$$+ \frac{n^{0,1}}{n^{0,1}+1}[p_1+p_1 p_{Y,1} + \sqrt{\frac{n^{0,1}}{n}}\epsilon(\sqrt{\frac{n^{0,1}}{n}}\epsilon + p_1 + p_{Y,1}+1)]]$$

$$+ \frac{n^{0,0}+0.5-\mathbb{E}(k_i^{0,0})}{n^{0,0}+1}\sqrt{\frac{n^{0,0}}{n}}\epsilon[\sqrt{\frac{n^{0,0}}{n}}\epsilon + p_0 + p_{Y,0}+1]$$

$$+ \frac{n^{0,1}+0.5-\mathbb{E}(k_i^{0,1})}{n^{0,1}+1}\sqrt{\frac{n^{0,1}}{n}}\epsilon[\sqrt{\frac{n^{0,1}}{n}}\epsilon + p_1 + p_{Y,1}+1] + \frac{p_0(1-p_{Y,0})}{2(n^{0,0}+1)} + \frac{p_1(1-p_{Y,1})}{2(n^{0,1}+1)}$$

$$\leq \epsilon[\sqrt{\frac{n^{0,0}}{n}}(p_0+p_{Y,0}) + \sqrt{\frac{n^{0,1}}{n}}(p_1+p_{Y,1})] + \epsilon^2(\frac{n^{0,0}}{n} + \frac{n^{0,1}}{n})$$

$$+ \epsilon[\frac{n^{0,0}}{n^{0,0}+1}[p_0+p_0 p_{Y,0} + \sqrt{\frac{n^{0,0}}{n}}\epsilon(\sqrt{\frac{n^{0,0}}{n}}\epsilon + p_0 + p_{Y,0}+1)]$$

$$+ \frac{n^{0,1}}{n^{0,1}+1}[p_1+p_1 p_{Y,1} + \sqrt{\frac{n^{0,1}}{n}}\epsilon(\sqrt{\frac{n^{0,1}}{n}}\epsilon + p_1 + p_{Y,1}+1)]]$$

$$+ \sqrt{\frac{n^{0,0}}{n}}\epsilon[\sqrt{\frac{n^{0,0}}{n}}\epsilon + p_0 + p_{Y,0}+1] + \sqrt{\frac{n^{0,1}}{n}}\epsilon[\sqrt{\frac{n^{0,1}}{n}}\epsilon + p_1 + p_{Y,1}+1]$$

$$+ \frac{p_0(1-p_{Y,0})}{2(n^{0,0}+1)} + \frac{p_1(1-p_{Y,1})}{2(n^{0,1}+1)}$$

$$\leq 2\epsilon + \epsilon^2 + \epsilon[\epsilon^2 + 4\epsilon + 2] + \epsilon^2 + 4\epsilon + \frac{p_0(1-p_{Y,0})}{2(n^{0,0}+1)} + \frac{p_1(1-p_{Y,1})}{2(n^{0,1}+1)}$$

$$= \epsilon^3 + 6\epsilon^2 + 8\epsilon + \frac{p_0(1-p_{Y,0})}{2(n^{0,0}+1)} + \frac{p_1(1-p_{Y,1})}{2(n^{0,1}+1)}$$

Thus we complete the proof. $\square$

### A.4 THEOREM FOR THE ORIGINAL CANDIDATE SET $K$

Sometimes we would use the original candidate set $K$ instead of the small set $K'$ to achieve the optimal accuracy more precisely. Now we provide our results for the candidate set $K$.

To facilitate the theoretical analysis, we first introduce the following lemma which implies that the difference between the output of the function can be controlled by the difference between the input. (i.e. the cumulative distribution function won't increase drastically.)

**Lemma 6.** *For a distribution $F$ with a continuous density function, suppose $q(x)$ denotes the quantile of $x$ under $F$, then for $x > y$, we have $F_{(-)}(x-y) \leq q(x) - q(y) \leq F_{(+)}(x-y)$, where $F_{(-)}(x)$ and $F_{(+)}(x)$ are two monotonically increasing functions, $F_{(-)}(\epsilon) > 0, F_{(+)}(\epsilon) > 0$ for any $\epsilon > 0$ and $\lim_{\epsilon \to 0} F_{(-)}(\epsilon) = \lim_{\epsilon \to 0} F_{(+)}(\epsilon) = 0$.*

*Proof of Lemma 6.* Since the domain of $q(x)$ is a closed set and $q(x)$ is continuous, we know that $q(x)$ is uniformly continuous. Thus we can easily find $F_{(+)}$ to satisfy the RHS. For $F_{(-)}$, we simply define $F_{(-)}(t) = \inf_x\{q(x+t) - q(t)\}$. Since $q(x+t) - q(t) > 0$ for $t > 0$ and the domain of $x$ is a closed set, we have $F_{(-)}(\epsilon) > 0$ for $\epsilon > 0$ and $\lim_{\epsilon \to 0} F_{(-)}(\epsilon) = 0$. Now we complete the proof. $\square$

Now we provide the following theorem.

**Theorem 5.** *Given $\alpha' < \alpha$. If $\min\{n^{1,0}, n^{1,1}\} \geq \lceil \frac{\log \frac{\delta}{2}}{\log(1-\alpha)} \rceil$. Suppose $\hat{\phi}$ is the final output of FaiREE, we have:*
*(1) $|DEOO(\hat{\phi})| < \alpha$ with probability $(1 - \delta)^M$, where $M$ is the size of the candidate set.*
*(2) Suppose the density distribution functions of $f^*$ under $A = a, Y = 1$ are continuous. When the input classifier $f$ satisfies $\mid f(x,a) - f^*(x,a) \mid \leq \epsilon_0$, for any $\epsilon > 0$ such that $F_{(+)}^*(\epsilon) \leq \frac{\alpha - \alpha'}{2} - F_{(+)}^*(2\epsilon_0)$, we have*

$$\mathbb{P}(\hat{\phi}(x,a) \neq Y) - \mathbb{P}(\phi_{\alpha'}^*(x,a) \neq Y) \leq 2F_{(+)}^*(2\epsilon_0) + 2F_{(+)}^*(\epsilon) + 2\epsilon^3 + 12\epsilon^2 + 16\epsilon + \frac{p_0(1 - p_{Y,0})}{n^{0,0} + 1} + \frac{p_1(1 - p_{Y,1})}{n^{0,1} + 1}$$

*with probability $1 - (2M + 4)(e^{-2n^{0,0}\epsilon^2} + e^{-2n^{0,1}\epsilon^2}) - (1 - F_{(-)}^{1,0}(2\epsilon))^{n^{1,0}} - (1 - F_{(-)}^{1,1}(2\epsilon))^{n^{1,1}}$.*

*Proof of Theorem 5.* The (1) of the theorem is a direct corollary from Theorem 1, now we prove the (2) of the theorem. The proof can be divided into two parts. The first part is to prove that there exist classifiers in our candidate set that are close to the fair Bayes-optimal classifier. The second part is to prove that our algorithm can successfully choose one of these classifiers with high probability.

We suppose the fair Bayes optimal classifier has the form $\phi_{\alpha'}^*(x,a) = \mathbb{1}\{f^*(x,a) > \lambda_a^*\}$. And the output classifier of our algorithm is of the form $\hat{\phi}(x,a) = \mathbb{1}\{f(x,a) > \lambda_a\}$.

For the first part, for any $\epsilon > 0$, from Lemma 6, $t^{1,a}$ has a positive probability $F_{(+)}^{1,a}(2\epsilon)$ to fall in the interval $[\lambda_a^* - \epsilon, \lambda_a^* + \epsilon]$, which implies that the probability that there exists $a \in \{0, 1\}$ such that all $t^{1,a}$'s fall out of $[\lambda_a^* - \epsilon, \lambda_a^* + \epsilon]$ is less than $(1 - F_{(+)}^{1,0}(2\epsilon))^{n^{1,0}} + (1 - F_{(+)}^{1,1}(2\epsilon))^{n^{1,1}}$. So with probability $1 - (1 - F_{(+)}^{1,0}(2\epsilon))^{n^{1,0}} - (1 - F_{(-)}^{1,1}(2\epsilon))^{n^{1,1}}$, there will exist $t^{1,a}$ in $[\lambda_a^* - \epsilon, \lambda_a^* + \epsilon]$, which we denote as $\phi_0(x,a) = \mathbb{1}\{f(x,a) > t_*^{1,a}\}$. We also denote $\phi_0^*(x,a) = \mathbb{1}\{f^*(x,a) > t_*^{1,a}\}$. Hence the gap between the classifier $\phi_0$ and the Bayes-optimal classifier will be very close. In detail, we have

$$\begin{aligned} &\mid \mathbb{P}(\phi_0(x,a) \neq Y) - \mathbb{P}(\phi_{\alpha'}^*(x,a) \neq Y) \mid \\ &\leq \mid \mathbb{P}(\phi_0(x,a) \neq Y) - \mathbb{P}(\phi_0^*(x,a) \neq Y) \mid + \mid \mathbb{P}(\phi_0^*(x,a) \neq Y) - \mathbb{P}(\phi_{\alpha'}^*(x,a) \neq Y) \mid \\ &\leq \mathbb{P}(t_*^{1,a} - \epsilon_0 \leq f^*(x,a) \leq t_*^{1,a} + \epsilon_0) + \mathbb{P}(\min\{t_*^{1,a}, \lambda_a^*\} \leq f^*(x,a) \leq max\{t_*^{1,a}, \lambda_a^*\}) \\ \text{(Lemma 6)} &\leq F_{(+)}^*(2\epsilon_0) + F_{(+)}^*(max\{t_*^{1,a}, \lambda_a^*\} - \min\{t_*^{1,a}, \lambda_a^*\}) \\ &\leq F_{(+)}^*(2\epsilon_0) + 2F_{(+)}^*(\epsilon) \end{aligned}$$

so we complete the first part of the proof.

Now we come to the second part. First, we notice that $DEOO(\phi_0)$ and $DEOO(\phi_{\alpha'}^*)$ are close to each other.

$$|| DEOO(\phi_0) | - | DEOO(\phi_{\alpha'}^*) ||$$
$$\leq || DEOO(\phi_0) | - | DEOO(\phi_0^*) || + || DEOO(\phi_0^*) | - | DEOO(\phi_{\alpha'}^*) ||$$
$$= || \mathbb{P}(f > t_*^{1,0} \mid Y = 1, A = 0) - \mathbb{P}(f > t_*^{1,1} \mid Y = 1, A = 1) |$$
$$- | \mathbb{P}(f^* > t_*^{1,0} \mid Y = 1, A = 0) - \mathbb{P}(f^* > t_*^{1,1} \mid Y = 1, A = 1) ||$$
$$+ || \mathbb{P}(f^* > t_*^{1,0} \mid Y = 1, A = 0) - \mathbb{P}(f^* > t_*^{1,1} \mid Y = 1, A = 1) |$$
$$- | \mathbb{P}(f^* > \lambda_0^* \mid Y = 1, A = 0) - \mathbb{P}(f^* > \lambda_1^* \mid Y = 1, A = 1) ||$$
$$\leq | \mathbb{P}(f > t_*^{1,0} \mid Y = 1, A = 0) - \mathbb{P}(f^* > t_*^{1,0} \mid Y = 1, A = 0) |$$
$$+ | \mathbb{P}(f > t_*^{1,1} \mid Y = 1, A = 1) - \mathbb{P}(f^* > t_*^{1,1} \mid Y = 1, A = 1) |$$
$$+ || \mathbb{P}(f^* > t_*^{1,0} \mid Y = 1, A = 0) - \mathbb{P}(f^* > t_*^{1,1} \mid Y = 1, A = 1) |$$
$$- | \mathbb{P}(f^* > \lambda_0^* \mid Y = 1, A = 0) - \mathbb{P}(f^* > \lambda_1^* \mid Y = 1, A = 1) ||$$
$$\leq \mathbb{P}(t_*^{1,0} - \epsilon_0 \leq f^*(x, a) \leq t_*^{1,0} + \epsilon_0) + \mathbb{P}(t_*^{1,1} - \epsilon_0 \leq f^*(x, a) \leq t_*^{1,1} + \epsilon_0)$$
$$+ | \mathbb{P}(f^* > t_*^{1,0} \mid Y = 1, A = 0) - \mathbb{P}(f^* > t_*^{1,1} \mid Y = 1, A = 1)$$
$$- \mathbb{P}(f^* > \lambda_0^* \mid Y = 1, A = 0) + \mathbb{P}(f^* > \lambda_1^* \mid Y = 1, A = 1) |$$
$$\leq 2F_{(+)}^*(2\epsilon_0) + \mathbb{P}(\min\{t_*^{1,a}, \lambda_a^*\} \leq f^*(x, a) \leq \max\{t_*^{1,a}, \lambda_a^*\})$$
$$(\text{Lemma 6}) \leq 2F_{(+)}^*(2\epsilon_0) + F_{(+)}^*(\max\{t_*^{1,a}, \lambda_a^*\} - \min\{t_*^{1,a}, \lambda_a^*\})$$
$$\leq 2F_{(+)}^*(2\epsilon_0) + 2F_{(+)}^*(\epsilon)$$

Thus, $| DEOO(\phi_0) | \leq | DEOO(\phi_{\alpha'}^*) | + 2F_{(+)}^*(2\epsilon_0) + 2F_{(+)}^*(\epsilon) = \alpha' + 2F_{(+)}^*(2\epsilon_0) + 2F_{(+)}^*(\epsilon)$. If $F_{(+)}^*(\epsilon) \leq \frac{\alpha - \alpha'}{2} - F_{(+)}^*(2\epsilon_0)$, then there will exist at least one feasible classifier in the candidate set.

From Lemma 3, we have the mis-classification error

$$| \mathbb{P}(\hat{\phi}_i(x, a) \neq Y) - [\frac{k_i^{1,0}}{n^{1,0}+1}p_0 p_{Y,0} + \frac{k_i^{1,1}}{n^{1,1}+1}p_1 p_{Y,1} + \frac{n^{0,0}+\frac{1}{2}-\mathbb{E}(k_i^{0,0})}{n^{0,0}+1}p_0(1 - p_{Y,0}) + \frac{n^{0,1}+\frac{1}{2}-\mathbb{E}(k_i^{0,1})}{n^{0,1}+1}p_1(1 - p_{Y,1})] | \leq \frac{p_0(1-p_{Y,0})}{2(n^{0,0}+1)} + \frac{p_1(1-p_{Y,1})}{2(n^{0,1}+1)}.$$

If we can accurately estimate the mis-classification error, than the second part is almost done. For the estimation of $\mathbb{E}(k_i^{0,0})$, we can easily use $k_i^{0,0}$. We notice that $k_i^{0,a}$ is the number of $t^{0,a}$'s such that $t^{0,a} < t_{(k_i^{1,a})}^{1,a}$, i.e. $k_i^{0,a} = \sum_{i=1}^{n^{0,1}} \mathbb{1}\{t_i^{0,a} < t_{(k_i^{1,a})}^{1,a}\}$.

Thus, from Hoeffding's inequality, we have with probability $1 - e^{-2n^{0,a}\epsilon^2}$,

$$\frac{k_i^{0,a} - \mathbb{E}(k_i^{0,a})}{n^{0,a}} = \frac{\sum_{j=1}^{n^{0,1}} \mathbb{1}\{t_j^{0,a} < t_{(k_i^{1,a})}^{1,a}\} - \sum_{j=1}^{n^{0,1}} \mathbb{E}(\mathbb{1}\{t_j^{0,a} < t_{(k_i^{1,a})}^{1,a}\})}{n^{0,a}} \leq \epsilon.$$

Similarly, with probability $1 - e^{-2n^{0,a}\epsilon^2}$,

$$\frac{k_i^{0,a} - \mathbb{E}(k_i^{0,a})}{n^{0,a}} = \frac{\sum_{j=1}^{n^{0,1}} \mathbb{1}\{t_j^{0,a} < t_{(k_i^{1,a})}^{1,a}\} - \sum_{j=1}^{n^{0,1}} \mathbb{E}(\mathbb{1}\{t_j^{0,a} < t_{(k_i^{1,a})}^{1,a}\})}{n^{0,a}} \geq -\epsilon.$$

Thus, $| \frac{k_i^{0,a} - \mathbb{E}(k_i^{0,a})}{n^{0,a}} | \leq \epsilon$ with probability $1 - 2e^{-2n^{0,a}\epsilon^2}$.

Then, it's easy to estimate $p_a$ and $p_{Y,a}$ with $\hat{p}_a = \frac{n^{1,a}+n^{0,a}}{n}$ and $\hat{p}_{Y,a} = \frac{n^{Y,a}}{n}$ ($n$ is the number of the total samples). Here, $\frac{n^{1,a}+n^{0,a}}{n} = \frac{\sum_{i=1}^{n} \mathbb{1}\{Z_i^a=1\}}{n}$ and $\frac{n^{Y,a}}{n} = \frac{\sum_{i=1}^{n} \mathbb{1}\{Z_i^{Y,a}=1\}}{n}$, where $Z_i^a \sim B(1, p_a)$ and $Z_i^{Y,a} \sim B(1, p_{Y,a})$. From Hoeffding's inequality, we have:

$$\mathbb{P}(| \hat{p}_a - p_a | \geq \sqrt{\frac{n^{0,a}}{n}}\epsilon) \leq 2e^{-2n^{0,a}\epsilon^2},$$

$$\mathbb{P}(\mid \hat{p}_{Y,a} - p_{Y,a} \mid \geq \sqrt{\frac{n^{0,a}}{n}}\epsilon) \leq 2e^{-2n^{0,a}\epsilon^2}$$

Thus, with probability $1 - 6e^{-2n^{0,a}\epsilon^2}$, we have:

$$\begin{cases} \mid \hat{p}_a - p_a \mid \leq \sqrt{\frac{n^{0,a}}{n}}\epsilon \\ \mid \hat{p}_{Y,a} - p_{Y,a} \mid \leq \sqrt{\frac{n^{0,a}}{n}}\epsilon \\ \mid \frac{k_i^{0,a} - \mathbb{E}(k_i^{0,a})}{n^{0,a}} \mid \leq \epsilon \end{cases}$$

Hence, we have with probability $1 - (2M + 4)(e^{-2n^{0,0}\epsilon^2} + e^{-2n^{0,1}\epsilon^2})$, for each $i \in \{1, \ldots, M\}$,

$$|\mathbb{P}(\hat{\phi}_i(x,a) \neq Y) - \hat{\mathbb{P}}(\hat{\phi}_i(x,a) \neq Y)|$$
$$\leq |\frac{k_i^{1,0}}{n^{1,0}+1}p_0 p_{Y,0} + \frac{k_i^{1,1}}{n^{1,1}+1}p_1 p_{Y,1} + \frac{n^{0,0}+0.5-\mathbb{E}(k_i^{0,0})}{n^{0,0}+1}p_0(1-p_{Y,0}) + \frac{n^{0,1}+0.5-\mathbb{E}(k_i^{0,1})}{n^{0,1}+1}p_1(1-p_{Y,1})$$
$$- [\frac{k_i^{1,0}}{n^{1,0}+1}\hat{p}_0\hat{p}_{Y,0} + \frac{k_i^{1,1}}{n^{1,1}+1}\hat{p}_1\hat{p}_{Y,1} + \frac{n^{0,0}+0.5-k_i^{0,0}}{n^{0,0}+1}\hat{p}_0(1-\hat{p}_{Y,0}) + \frac{n^{0,1}+0.5-k_i^{0,1}}{n^{0,1}+1}\hat{p}_1(1-\hat{p}_{Y,1})]|$$
$$+ \frac{p_0(1-p_{Y,0})}{2(n^{0,0}+1)} + \frac{p_1(1-p_{Y,1})}{2(n^{0,1}+1)}$$
$$\leq \epsilon[\sqrt{\frac{n^{0,0}}{n}}\frac{k_i^{1,0}}{n^{1,0}+1}(p_0+p_{Y,0}) + \sqrt{\frac{n^{0,1}}{n}}\frac{k_i^{1,1}}{n^{1,1}+1}(p_1+p_{Y,1})] + \epsilon^2(\frac{n^{0,0}}{n}\frac{k_i^{1,0}}{n^{1,0}+1} + \frac{n^{0,1}}{n}\frac{k_i^{1,1}}{n^{1,1}+1})$$
$$+ \epsilon[\frac{n^{0,0}}{n^{0,0}+1}[p_0 + p_0 p_{Y,0} + \sqrt{\frac{n^{0,0}}{n}}\epsilon(\sqrt{\frac{n^{0,0}}{n}}\epsilon + p_0 + p_{Y,0} + 1)]$$
$$+ \frac{n^{0,1}}{n^{0,1}+1}[p_1 + p_1 p_{Y,1} + \sqrt{\frac{n^{0,1}}{n}}\epsilon(\sqrt{\frac{n^{0,1}}{n}}\epsilon + p_1 + p_{Y,1} + 1)]]$$
$$+ \frac{n^{0,0}+0.5-\mathbb{E}(k_i^{0,0})}{n^{0,0}+1}\sqrt{\frac{n^{0,0}}{n}}\epsilon[\sqrt{\frac{n^{0,0}}{n}}\epsilon + p_0 + p_{Y,0} + 1]$$
$$+ \frac{n^{0,1}+0.5-\mathbb{E}(k_i^{0,1})}{n^{0,1}+1}\sqrt{\frac{n^{0,1}}{n}}\epsilon[\sqrt{\frac{n^{0,1}}{n}}\epsilon + p_1 + p_{Y,1} + 1]$$
$$+ \frac{p_0(1-p_{Y,0})}{2(n^{0,0}+1)} + \frac{p_1(1-p_{Y,1})}{2(n^{0,1}+1)}$$

Further, we obtain

$$|\mathbb{P}(\hat{\phi}_i(x,a) \neq Y) - \hat{\mathbb{P}}(\hat{\phi}_i(x,a) \neq Y)|$$

$$\leq \epsilon[\sqrt{\frac{n^{0,0}}{n}}(p_0 + p_{Y,0}) + \sqrt{\frac{n^{0,1}}{n}}(p_1 + p_{Y,1})] + \epsilon^2(\frac{n^{0,0}}{n} + \frac{n^{0,1}}{n})$$

$$+ \epsilon[\frac{n^{0,0}}{n^{0,0}+1}[p_0 + p_0 p_{Y,0} + \sqrt{\frac{n^{0,0}}{n}}\epsilon(\sqrt{\frac{n^{0,0}}{n}}\epsilon + p_0 + p_{Y,0} + 1)]$$

$$+ \frac{n^{0,1}}{n^{0,1}+1}[p_1 + p_1 p_{Y,1} + \sqrt{\frac{n^{0,1}}{n}}\epsilon(\sqrt{\frac{n^{0,1}}{n}}\epsilon + p_1 + p_{Y,1} + 1)]]$$

$$+ \sqrt{\frac{n^{0,0}}{n}}\epsilon[\sqrt{\frac{n^{0,0}}{n}}\epsilon + p_0 + p_{Y,0} + 1]$$

$$+ \sqrt{\frac{n^{0,1}}{n}}\epsilon[\sqrt{\frac{n^{0,1}}{n}}\epsilon + p_1 + p_{Y,1} + 1]$$

$$+ \frac{p_0(1 - p_{Y,0})}{2(n^{0,0}+1)} + \frac{p_1(1 - p_{Y,1})}{2(n^{0,1}+1)}$$

$$\leq 2\epsilon + \epsilon^2 + \epsilon[\epsilon^2 + 4\epsilon + 2] + \epsilon^2 + 4\epsilon$$

$$+ \frac{p_0(1 - p_{Y,0})}{2(n^{0,0}+1)} + \frac{p_1(1 - p_{Y,1})}{2(n^{0,1}+1)}$$

$$= \epsilon^3 + 6\epsilon^2 + 8\epsilon + \frac{p_0(1 - p_{Y,0})}{2(n^{0,0}+1)} + \frac{p_1(1 - p_{Y,1})}{2(n^{0,1}+1)}$$

Combining two parts together, we have:
with probability $1 - (2M + 4)(e^{-2n^{0,0}\epsilon^2} + e^{-2n^{0,1}\epsilon^2}) - (1 - F_{(-)}^{1,0}(2\epsilon))^{n^{1,0}} - (1 - F_{(-)}^{1,1}(2\epsilon))^{n^{1,1}}$,

$$\mathbb{P}(\hat{\phi}(x,a) \neq Y) - \mathbb{P}(\phi_{\alpha'}^*(x,a) \neq Y) \leq 2F_{(+)}^*(\epsilon) + 2F_{(+)}^*(2\epsilon_0) + 2\epsilon^3 + 12\epsilon^2 + 16\epsilon + \frac{p_0(1 - p_{Y,0})}{n^{0,0}+1} + \frac{p_1(1 - p_{Y,1})}{n^{0,1}+1}.$$

Now we complete the proof. □

## A.5   PROOF OF THEOREM 2

*Proof.* The (1) of the theorem is a direct corollary from Theorem 1, now we prove the (2) of the theorem.

It's sufficient to modify the first part of the proof of Theorem 5 and the second part simply follows Theorem 5.

For the first part, for any $\epsilon > 0$, from Lemma 6, $t^{1,0}$ has a positive probability $F_{(-)}^{1,0}(2\epsilon)$ to fall in the interval $[\lambda_0^* - \epsilon, \lambda_0^* + \epsilon]$, which implies that the probability that all $t^{1,0}$'s fall out of $[\lambda_0^* - \epsilon, \lambda_0^* + \epsilon]$ is less than $(1 - F_{(-)}^{1,0}(2\epsilon))^{n^{1,0}}$. So with probability $1 - (1 - F_{(-)}^{1,0}(2\epsilon))^{n^{1,0}}$, there will exist $t^{1,0}$ in $[\lambda_0^* - \epsilon, \lambda_0^* + \epsilon]$, which we denote as $\lambda_0$. We denote the corresponding classifier as $\mathbb{1}\{f(x,a) > \lambda_a\}$. From the proof of Theorem 5, we have with probability $1 - 4e^{-2n^{0,a}\epsilon^2} - (1 - F_{(-)}^{1,0}(2\epsilon))^{n^{1,0}}$,

$$\begin{cases} |\hat{p}_a - p_a| \leq \sqrt{\frac{n^{0,a}}{n}}\epsilon \\ |\hat{p}_{Y,a} - p_{Y,a}| \leq \sqrt{\frac{n^{0,a}}{n}}\epsilon \\ |\lambda_0 - \lambda_0^*| \leq \epsilon \end{cases}$$

We have the following equalities:

$$\begin{cases} \lambda_1^* = \dfrac{1}{2 - \dfrac{(\frac{1}{\lambda_0^*}-2)p_0 p_{Y,0}}{p_1 p_{Y,1}}} \\[4mm] \lambda_1 = \dfrac{1}{2 - \dfrac{(\frac{1}{\lambda_0}-2)\hat{p}_0 \hat{p}_{Y,0}}{\hat{p}_1 \hat{p}_{Y,1}}} \end{cases}$$

Hence,

$$\begin{cases} \dfrac{p_0 p_{Y,0}}{\lambda_0^*} + \dfrac{p_1 p_{Y,1}}{\lambda_1^*} = 2(p_0 p_{Y,0} + p_1 p_{Y,1}) \\[4mm] \dfrac{\hat{p}_0 \hat{p}_{Y,0}}{\lambda_0} + \dfrac{\hat{p}_1 \hat{p}_{Y,1}}{\lambda_1} = 2(\hat{p}_0 \hat{p}_{Y,0} + \hat{p}_1 \hat{p}_{Y,1}) \end{cases}$$

By subtracting, we have

$$(2 - \frac{1}{\lambda_1})\hat{p}_1 \hat{p}_{Y,1} - (2 - \frac{1}{\lambda_1^*})p_1 p_{Y,1} = (\frac{1}{\lambda_0} - 2)\hat{p}_0 \hat{p}_{Y,0} - (\frac{1}{\lambda_0^*} - 2)p_0 p_{Y,0}.$$

We have

$$(\frac{1}{\lambda_0} - 2)\hat{p}_0 \hat{p}_{Y,0} - (\frac{1}{\lambda_0^*} - 2)p_0 p_{Y,0}$$

$$\leq (\frac{1}{\lambda_0} - 2)(p_0 + \sqrt{\frac{n^{0,0}}{n}}\epsilon)(p_{Y,0} + \sqrt{\frac{n^{0,0}}{n}}\epsilon) - (\frac{1}{\lambda_0^*} - 2)p_0 p_{Y,0}$$

$$\leq \frac{\epsilon}{\lambda_0^*(\lambda_0^* - \epsilon)}p_0 p_{Y,0} + \sqrt{\frac{n^{0,0}}{n}}\epsilon(\frac{1}{\lambda_0^* - \epsilon} - 2)(p_0 + p_{Y,0} + \sqrt{\frac{n^{0,0}}{n}}\epsilon)$$

Similarly, we have

$$(\frac{1}{\lambda_0} - 2)\hat{p}_0 \hat{p}_{Y,0} - (\frac{1}{\lambda_0^*} - 2)p_0 p_{Y,0}$$

$$\geq -\frac{\epsilon}{\lambda_0^*(\lambda_0^* + \epsilon)}p_0 p_{Y,0} + \sqrt{\frac{n^{0,0}}{n}}\epsilon(\frac{1}{\lambda_0^* + \epsilon} - 2)(-p_0 - p_{Y,0} + \sqrt{\frac{n^{0,0}}{n}}\epsilon);$$

$$(2 - \frac{1}{\lambda_1})\hat{p}_1 \hat{p}_{Y,1} - (2 - \frac{1}{\lambda_1^*})p_1 p_{Y,1}$$

$$\leq (\frac{1}{\lambda_1^*} - \frac{1}{\lambda_1})p_1 p_{Y,1} + \sqrt{\frac{n^{0,1}}{n}}\epsilon(2 - \frac{1}{\lambda_1})(p_1 + p_{Y,1} + \sqrt{\frac{n^{0,1}}{n}}\epsilon);$$

$$(2 - \frac{1}{\lambda_1})\hat{p}_1 \hat{p}_{Y,1} - (2 - \frac{1}{\lambda_1^*})p_1 p_{Y,1}$$

$$\geq (\frac{1}{\lambda_1^*} - \frac{1}{\lambda_1})p_1 p_{Y,1} + \sqrt{\frac{n^{0,1}}{n}}\epsilon(2 - \frac{1}{\lambda_1})(-p_1 - p_{Y,1} + \sqrt{\frac{n^{0,1}}{n}}\epsilon).$$

Now, we get:

$$\begin{cases} (\frac{1}{\lambda_1^*} - \frac{1}{\lambda_1})p_1 p_{Y,1} + \sqrt{\frac{n^{0,1}}{n}}\epsilon(2 - \frac{1}{\lambda_1})(-p_1 - p_{Y,1} + \sqrt{\frac{n^{0,1}}{n}}\epsilon) \\[4mm] \leq \frac{\epsilon}{\lambda_0^*(\lambda_0^* - \epsilon)}p_0 p_{Y,0} + \sqrt{\frac{n^{0,0}}{n}}\epsilon(\frac{1}{\lambda_0^* - \epsilon} - 2)(p_0 + p_{Y,0} + \sqrt{\frac{n^{0,0}}{n}}\epsilon) \\[4mm] (\frac{1}{\lambda_1^*} - \frac{1}{\lambda_1})p_1 p_{Y,1} + \sqrt{\frac{n^{0,1}}{n}}\epsilon(2 - \frac{1}{\lambda_1})(p_1 + p_{Y,1} + \sqrt{\frac{n^{0,1}}{n}}\epsilon) \\[4mm] \geq -\frac{\epsilon}{\lambda_0^*(\lambda_0^* + \epsilon)}p_0 p_{Y,0} + \sqrt{\frac{n^{0,0}}{n}}\epsilon(\frac{1}{\lambda_0^* + \epsilon} - 2)(-p_0 - p_{Y,0} + \sqrt{\frac{n^{0,0}}{n}}\epsilon) \end{cases}$$

Hence, we have:

$$\lambda_1 - \lambda_1^*$$

$$\leq \frac{\lambda_1 \lambda_1^*}{p_1 p_{Y,1}}\left[\frac{\epsilon}{\lambda_0^*(\lambda_0^* - \epsilon)}p_0 p_{Y,0} + \sqrt{\frac{n^{0,0}}{n}}\epsilon(\frac{1}{\lambda_0^* - \epsilon} - 2)(p_0 + p_{Y,0} + \sqrt{\frac{n^{0,0}}{n}}\epsilon) - 2\sqrt{\frac{n^{0,1}}{n}}\epsilon(-p_1 - p_{Y,1} + \sqrt{\frac{n^{0,1}}{n}}\epsilon)\right]$$

and

$$\lambda_1 - \lambda_1^*$$

$$\geq \frac{\lambda_1 \lambda_1^*}{p_1 p_{Y,1}}\left[-\frac{\epsilon}{\lambda_0^*(\lambda_0^* + \epsilon)}p_0 p_{Y,0} + \sqrt{\frac{n^{0,0}}{n}}\epsilon(\frac{1}{\lambda_0^* + \epsilon} - 2)(-p_0 - p_{Y,0} + \sqrt{\frac{n^{0,0}}{n}}\epsilon) - 2\sqrt{\frac{n^{0,1}}{n}}\epsilon(p_1 + p_{Y,1} + \sqrt{\frac{n^{0,1}}{n}}\epsilon)\right]$$

Thus, we have:

$$| \lambda_1 - \lambda_1^* | \leq \frac{\frac{\epsilon}{\lambda_0^*(\lambda_0^* - \epsilon)}p_0 p_{Y,0} + \epsilon(\frac{1}{\lambda_0^* - \epsilon} - 2)(2 + \epsilon) + 4\epsilon}{p_1 p_{Y,1}}$$

Now, combined with above, we have with probability $1 - 4e^{-2n^{0,a}\epsilon^2} - (1 - F_{(-)}^{1,0}(2\epsilon))^{n^{1,0}}$,

$$\begin{cases} | \hat{p}_a - p_a | \leq \sqrt{\frac{n^{0,a}}{n}}\epsilon \\ | \hat{p}_{Y,a} - p_{Y,a} | \leq \sqrt{\frac{n^{0,a}}{n}}\epsilon \\ | \lambda_0 - \lambda_0^* | \leq \epsilon \\ | \lambda_1 - \lambda_1^* | \leq \frac{\frac{\epsilon}{\lambda_0^*(\lambda_0^* - \epsilon)}p_0 p_{Y,0} + \epsilon(\frac{1}{\lambda_0^* - \epsilon} - 2)(2 + \epsilon) + 4\epsilon}{p_1 p_{Y,1}} \end{cases}$$

From the proof of Theorem 5, we have:

If $F_{(+)}^*(\epsilon) \leq \frac{\alpha - \alpha'}{2} - F_{(+)}^*(2\epsilon_0)$, then with probability $1 - (2M + 4)(e^{-2n^{0,0}\epsilon^2} + e^{-2n^{0,1}\epsilon^2}) - (1 - F_{(-)}^{1,0}(2\epsilon))^{n^{1,0}}$,

$$\mathbb{P}(\hat{\phi}(x, a) \neq Y) - \mathbb{P}(\phi_{\alpha'}^*(x, a) \neq Y)$$

$$\leq 2F_{(+)}^*(2\epsilon_0) + F_{(+)}^*(\epsilon) + F_{(+)}^*(\frac{\frac{\epsilon}{\lambda_0^*(\lambda_0^* - \epsilon)}p_0 p_{Y,0} + \epsilon(\frac{1}{\lambda_0^* - \epsilon} - 2)(2 + \epsilon) + 4\epsilon}{p_1 p_{Y,1}}) + 2\epsilon^3 + 12\epsilon^2 + 16\epsilon.$$

Now we complete the proof. $\qquad\qquad\square$

## A.6 Fair Bayes-optimal Classifiers under Equalized Odds

**Theorem 6** (Fair Bayes-optimal Classifiers under Equalized Odds). *Let $EO^\star = \mathrm{DEO}(f^\star) = (E^\star, P^\star)$. For any $\alpha > 0$, there exist $0 < \alpha_1 \leq \alpha$ and $0 < \alpha_2 \leq \alpha$ such that all fair Bayes-optimal classifiers $f_{EO,\alpha}^\star$ under the fairness constraint $|\mathrm{DEO}(f)| \preceq (\alpha_1, \alpha_2)$ are given as below:*

- *When $|EO^\star| \preceq (\alpha_1, \alpha_2)$, $f_{EO,\alpha}^\star = f^\star$.*

- *When $E^* > \alpha_1$ or $P^* > \alpha_2$, for all $x \in \mathcal{X}$ and $a \in \mathcal{A}$, there exist $t_{1,EO,\alpha}^\star$ and $t_{2,EO,\alpha}^\star$ such that*

$$f_{EO,\alpha}^\star(x, a) =$$
$$I\left(\eta_a(x) > \frac{p_a p_{Y,a} + (2a - 1)\frac{P_{Y,a}}{1 - P_{Y,a}}t_{2,EO,\alpha}^\star}{2p_a p_{Y,a} + (2a - 1)(\frac{P_{Y,a}}{1 - P_{Y,a}}t_{2,EO,\alpha}^\star - t_{1,EO,\alpha}^\star)}\right)$$
$$+ a\tau_{EO,\alpha}^\star I\left(\eta_a(x) = \frac{p_a p_{Y,a} + (2a - 1)\frac{P_{Y,a}}{1 - P_{Y,a}}t_{2,EO,\alpha}^\star}{2p_a p_{Y,a} + (2a - 1)(\frac{P_{Y,a}}{1 - P_{Y,a}}t_{2,EO,\alpha}^\star - t_{1,EO,\alpha}^\star)}\right),$$

*Here, we assume $P_{X|A=1,Y=1}\left(\eta_1(X) = \frac{p_1 p_{Y,1} + \frac{P_{Y,1}}{1 - P_{Y,1}}t_{2,EO,\alpha}^\star}{2p_1 p_{Y,1} + \frac{P_{Y,1}}{1 - P_{Y,1}}t_{2,EO,\alpha}^\star - t_{1,EO,\alpha}^\star}\right)$*

$$= P_{X|A=0,Y=0}\left(\eta_1(X) = \frac{p_1 p_{Y,1} + \frac{P_{Y,1}}{1 - P_{Y,1}}t_{2,EO,\alpha}^\star}{2p_1 p_{Y,1} + \frac{P_{Y,1}}{1 - P_{Y,1}}t_{2,EO,\alpha}^\star - t_{1,EO,\alpha}^\star}\right) = 0 \text{ and thus } \tau_{EO,\alpha}^\star \in [0,1]$$
*can be an arbitrary constant.*

To prove Theorem 6, we first introduce the Neyman-Pearson Lemma.

**Lemma 7.** *(Generalized Neyman-Pearson lemma). Let $f_0, f_1, \ldots, f_m$ be $m+1$ real-valued functions defined on a Euclidean space $\mathcal{X}$. Assume they are $\nu$-integrable for a $\sigma$-finite measure $\nu$. Let $\phi_0$ be any function of the form*

$$\phi_0(x) = \begin{cases} 1, & f_0(x) > \sum_{i=1}^{m} c_i f_i(x) \\ \gamma(x) & f_0(x) = \sum_{i=1}^{m} c_i f_i(x) \\ 0, & f_0(x) < \sum_{i=1}^{m} c_i f_i(x) \end{cases}$$

*where $0 \leq \gamma(x) \leq 1$ for all $x \in \mathcal{X}$. For given constants $t_1, \ldots, t_m \in \mathbb{R}$, let $\mathcal{T}$ be the class of Borel functions $\phi : \mathcal{X} \mapsto \mathbb{R}$ satisfying*

$$\int_{\mathcal{X}} \phi f_i d\nu \leq t_i, \quad i = 1, 2, \ldots, m \tag{5}$$

*and $\mathcal{T}_0$ be the set of $\phi$ s in $\mathcal{T}$ satisfying (5) with all inequalities replaced by equalities. If $\phi_0 \in \mathcal{T}_0$, then $\phi_0 \in \underset{\phi \in \mathcal{T}_0}{\operatorname{argmax}} \int_{\mathcal{X}} \phi f_0 d\nu$. Moreover, if $c_i \geq 0$ for all $i = 1, \ldots, m$, then $\phi_0 \in \underset{\phi \in \mathcal{T}}{\operatorname{argmax}} \int_{\mathcal{X}} \phi f_0 d\nu$.*

Then we come to prove the theorem.

*Proof.* If $|EO^\star| \preceq (\alpha, \alpha)$, we are done since $f^\star$ is just our target classifier. Now, we assume $|EO^\star| \preceq (\alpha, \alpha)$ does not hold. Let $f$ be a classifier that gives output $\widehat{Y} = 1$ with probability $f(x, a)$ under $X = x$ and $A = a$. The mis-classification error for $f$ is

$$R(f) = \mathbb{P}(\widehat{Y} \neq Y) = 1 - \mathbb{P}(\widehat{Y} = 1, Y = 1) - \mathbb{P}(\widehat{Y} = 0, Y = 0)$$
$$= \mathbb{P}(\widehat{Y} = 1, Y = 0) - \mathbb{P}(\widehat{Y} = 1, Y = 1) + \mathbb{P}(Y = 1)$$

Thus, to minimize the mis-classification error is just equivalent to maximize $\mathbb{P}(\widehat{Y} = 1, Y = 0) - \mathbb{P}(\widehat{Y} = 1, Y = 1)$, which can be expressed as:

$$\mathbb{P}(\widehat{Y} = 1, Y = 1) - \mathbb{P}(\widehat{Y} = 1, Y = 0)$$
$$= \mathbb{P}_{X|A=1,Y=1}(\widehat{Y} = 1) p_1 p_{Y,1} + \mathbb{P}_{X|A=0,Y=1}(\widehat{Y} = 1)(1 - p_1) p_{Y,0}$$
$$\quad - \mathbb{P}_{X|A=1,Y=0}(\widehat{Y} = 1) p_1 (1 - p_{Y,1}) - \mathbb{P}_{X|A=0,Y=0}(\widehat{Y} = 1)(1 - p_1)(1 - p_{Y,0})$$
$$= p_1 \left[ p_{Y,1} \int_{\mathcal{X}} f(x, 1) d\mathbb{P}_{X|1,1}(x) - (1 - p_{Y,1}) \int_{\mathcal{X}} f(x, 1) d\mathbb{P}_{X|A=1,Y=0}(x) \right]$$
$$\quad + (1 - p_1) \left[ p_{Y,0} \int_{\mathcal{X}} f(x, 0) d\mathbb{P}_{X|A=0,Y=1}(x) - (1 - p_{Y,0}) \int_{\mathcal{X}} f(x, 0) d\mathbb{P}_{X|A=0,Y=0}(x) \right]$$
$$= \int_{\mathcal{A}} \int_{\mathcal{X}} f(x, a) M(x, a) d\mathbb{P}_X(x) d\mathbb{P}(a)$$

with

$$\begin{aligned} M(x, a) = a p_1 &\left[ p_{Y,1} \frac{d\mathbb{P}_{X|A=1,Y=1}(x)}{d\mathbb{P}_X(x)} - (1 - p_{Y,1}) \frac{d\mathbb{P}_{X|A=1,Y=0}(x)}{d\mathbb{P}_X(x)} \right] \\ + (1 - a) p_0 &\left[ p_{Y,0} \frac{d\mathbb{P}_{X|A=0,Y=1}(x)}{d\mathbb{P}_X(x)} - (1 - p_{Y,0}) \frac{d\mathbb{P}_{X|A=0,Y=0}(x)}{d\mathbb{P}_X(x)} \right]. \end{aligned} \tag{6}$$

Next, for any classifier $f$, we have,

$$DEO(f) = (\mathbb{P}_{X|A=1,Y=1}(\widehat{Y} = 1) - \mathbb{P}_{X|A=0,Y=1}(\widehat{Y} = 1), \mathbb{P}_{X|A=1,Y=0}(\widehat{Y} = 1) - \mathbb{P}_{X|A=0,Y=0}(\widehat{Y} = 1))$$
$$= \left( \int_{\mathcal{X}} f(x, 1) d\mathbb{P}_{X|A=1,Y=1}(x) - \int_{\mathcal{X}} f(x, 0) d\mathbb{P}_{X|A=0,Y=1}(x), \right.$$
$$\left. \int_{\mathcal{X}} f(x, 1) d\mathbb{P}_{X|A=1,Y=0}(x) - \int_{\mathcal{X}} f(x, 0) d\mathbb{P}_{X|A=0,Y=0}(x) \right)$$
$$= \left( \int_{\mathcal{A}} \int_{\mathcal{X}} f(x, a) H_E(x, a) d\mathbb{P}_X(x) d\mathbb{P}(a), \int_{\mathcal{A}} \int_{\mathcal{X}} f(x, a) H_P(x, a) d\mathbb{P}_X(x) d\mathbb{P}(a) \right)$$

with

$$\begin{cases} H_E(x,a) = \dfrac{ad\mathbb{P}_{X|A=1,Y=1}(x)}{p_1 d\mathbb{P}_X(x)} - \dfrac{(1-a)d\mathbb{P}_{X|A=0,Y=1}(x)}{p_0 d\mathbb{P}_X(x)} \\[2ex] H_P(x,a) = \dfrac{ad\mathbb{P}_{X|A=1,Y=0}(x)}{p_1 d\mathbb{P}_X(x)} - \dfrac{(1-a)d\mathbb{P}_{X|A=0,Y=0}(x)}{p_0 d\mathbb{P}_X(x)}. \end{cases} \tag{7}$$

Since $\displaystyle\lim_{t_2\to\infty}\frac{p_1^2 p_{Y,1}+\frac{p_{Y,1}}{1-p_{Y,1}}t_2}{2p_1^2 p_{Y,1}+\frac{p_{Y,1}}{1-p_{Y,1}}t_2-t_1} = \lim_{t_2\to\infty}\frac{p_0^2 p_{Y,0}-\frac{p_{Y,0}}{1-p_{Y,0}}t_2}{2p_0^2 p_{Y,0}-\frac{p_{Y,0}}{1-p_{Y,0}}t_2+t_1} = 1$, we have:

$$\lim_{t_2\to\infty}\mathbb{P}_{X|A=1,Y=1}\left(\eta_1(x) > \frac{p_1^2 p_{Y,1}+\frac{p_{Y,1}}{1-p_{Y,1}}t_2}{2p_1^2 p_{Y,1}+\frac{p_{Y,1}}{1-p_{Y,1}}t_2-t_1}\right)$$

$$= \lim_{t_2\to\infty}\mathbb{P}_{X|A=0,Y=1}\left(\eta_0(x) > \frac{p_0^2 p_{Y,0}-\frac{p_{Y,0}}{1-p_{Y,0}}t_2}{2p_0^2 p_{Y,0}-\frac{p_{Y,0}}{1-p_{Y,0}}t_2+t_1}\right)$$

$$= 0$$

and

$$\lim_{t_2\to\infty}\mathbb{P}_{X|A=1,Y=0}\left(\eta_1(x) > \frac{p_1^2 p_{Y,1}+\frac{p_{Y,1}}{1-p_{Y,1}}t_2}{2p_1^2 p_{Y,1}+\frac{p_{Y,1}}{1-p_{Y,1}}t_2-t_1}\right)$$

$$= \lim_{t_2\to\infty}\mathbb{P}_{X|A=0,Y=0}\left(\eta_0(x) > \frac{p_0^2 p_{Y,0}-\frac{p_{Y,0}}{1-p_{Y,0}}t_2}{2p_0^2 p_{Y,0}-\frac{p_{Y,0}}{1-p_{Y,0}}t_2+t_1}\right)$$

$$= 0$$

So there exist $t^\star_{1,EO,\alpha}$ and $t^\star_{2,EO,\alpha}$, such that:
$t^\star_{1,EO,\alpha}\frac{E^*}{|E^*|} > 0$, $t^\star_{2,EO,\alpha}\frac{P^*}{|P^*|} > 0$, and

$$\begin{cases} \dfrac{E^\star}{|E^\star|}\Big[\mathbb{P}_{X|A=1,Y=1}\big(\eta_1(x) > \dfrac{p_1^2 p_{Y,1}+\frac{p_{Y,1}}{1-p_{Y,1}}t^\star_{2,EO,\alpha}}{2p_1^2 p_{Y,1}+\frac{p_{Y,1}}{1-p_{Y,1}}t^\star_{2,EO,\alpha}-t^\star_{1,EO,\alpha}}\big) + \tau\mathbb{P}_{X|A=1,Y=1}\big(\eta_1(x) = \dfrac{p_1^2 p_{Y,1}+\frac{p_{Y,1}}{1-p_{Y,1}}t^\star_{2,EO,\alpha}}{2p_1^2 p_{Y,1}+\frac{p_{Y,1}}{1-p_{Y,1}}t^\star_{2,EO,\alpha}-t^\star_{1,EO,\alpha}}\big) \\[3ex] \qquad\qquad\qquad\qquad - \mathbb{P}_{X|A=0,Y=1}\big(\eta_0(x) > \dfrac{p_0^2 p_{Y,0}-\frac{p_{Y,0}}{1-p_{Y,0}}t^\star_{2,EO,\alpha}}{2p_0^2 p_{Y,0}-\frac{p_{Y,0}}{1-p_{Y,0}}t^\star_{2,EO,\alpha}+t^\star_{1,EO,\alpha}}\big)\Big] = \alpha_1 < \alpha \\[3ex] \dfrac{P^\star}{|P^\star|}\Big[\mathbb{P}_{X|A=1,Y=0}\big(\eta_1(x) > \dfrac{p_1^2 p_{Y,1}+\frac{p_{Y,1}}{1-p_{Y,1}}t^\star_{2,EO,\alpha}}{2p_1^2 p_{Y,1}+\frac{p_{Y,1}}{1-p_{Y,1}}t^\star_{2,EO,\alpha}-t^\star_{1,EO,\alpha}}\big) + \tau\mathbb{P}_{X|A=1,Y=0}\big(\eta_1(x) = \dfrac{p_1^2 p_{Y,1}+\frac{p_{Y,1}}{1-p_{Y,1}}t^\star_{2,EO,\alpha}}{2p_1^2 p_{Y,1}+\frac{p_{Y,1}}{1-p_{Y,1}}t^\star_{2,EO,\alpha}-t^\star_{1,EO,\alpha}}\big) \\[3ex] \qquad\qquad\qquad\qquad - \mathbb{P}_{X|A=0,Y=0}\big(\eta_0(x) > \dfrac{p_0^2 p_{Y,0}-\frac{p_{Y,0}}{1-p_{Y,0}}t^\star_{2,EO,\alpha}}{2p_0^2 p_{Y,0}-\frac{p_{Y,0}}{1-p_{Y,0}}t^\star_{2,EO,\alpha}+t^\star_{1,EO,\alpha}}\big)\Big] = \alpha_2 < \alpha \end{cases}$$

We consider the constraint,

$$\begin{cases} \dfrac{E^\star}{|E^\star|}\displaystyle\int_{\mathcal{A}}\int_{\mathcal{X}} f(x,a)H_E(x,a)d\mathbb{P}_X(x)d\mathbb{P}(a) \le \alpha_1 \\[2ex] \dfrac{F^\star}{|F^\star|}\displaystyle\int_{\mathcal{A}}\int_{\mathcal{X}} f(x,a)H_P(x,a)d\mathbb{P}_X(x)d\mathbb{P}(a) \le \alpha_2. \end{cases} \tag{8}$$

Let $f$ be the classifier of the form:

$$f_{s_1,s_2,\tau}(x,a) = \begin{cases} 1, & M(x,a) > s_1\frac{E^\star}{|E^\star|}H_E(x,a) + s_2\frac{P^\star}{|P^\star|}H_P(x,a); \\ a\tau, & M(x,a) = s_1\frac{E^\star}{|E^\star|}H_E(x,a) + s_2\frac{P^\star}{|P^\star|}H_P(x,a); \\ 0, & M(x,a) < s_1\frac{E^\star}{|E^\star|}H_E(x,a) + s_2\frac{P^\star}{|P^\star|}H_P(x,a), \end{cases} \tag{9}$$

From (6) & (7), $M(x,a) > s_1 \frac{E^\star}{|E^\star|} H_E(x,a) + s_2 \frac{P^\star}{|P^\star|} H_P(x,a)$ is equal to

$$ap_1 \left[ p_{Y,1} \frac{d\mathbb{P}_{X|A=1,Y=1}(x)}{d\mathbb{P}_X(x)} - (1-p_{Y,1}) \frac{d\mathbb{P}_{X|A=1,Y=0}(x)}{d\mathbb{P}_X(x)} \right]$$

$$+ (1-a)p_0 \left[ p_{Y,0} \frac{d\mathbb{P}_{X|A=0,Y=1}(x)}{d\mathbb{P}_X(x)} - (1-p_{Y,0}) \frac{d\mathbb{P}_{X|A=0,Y=0}(x)}{d\mathbb{P}_X(x)} \right]$$

$$> \frac{E^\star}{|E^\star|} s_1 \left( \frac{a d\mathbb{P}_{X|A=1,Y=1}(x)}{p_1 d\mathbb{P}_X(x)} - \frac{(1-a)d\mathbb{P}_{X|A=0,Y=1}(x)}{p_0 d\mathbb{P}_X(x)} \right)$$

$$+ \frac{P^\star}{|P^\star|} s_2 \left( \frac{a d\mathbb{P}_{X|A=1,Y=0}(x)}{p_1 d\mathbb{P}_X(x)} - \frac{(1-a)d\mathbb{P}_{X|A=0,Y=0}(x)}{p_0 d\mathbb{P}_X(x)} \right)$$

which is equal to

$$\begin{cases} p_1 \left[ p_{Y,1} \frac{d\mathbb{P}_{X|A=1,Y=1}(x)}{d\mathbb{P}_X(x)} - (1-p_{Y,1}) \frac{d\mathbb{P}_{X|A=1,Y=0}(x)}{d\mathbb{P}_X(x)} \right] > \frac{\frac{E^\star}{|E^\star|} s_1 d\mathbb{P}_{X|A=1,Y=1}(x) + \frac{P^\star}{|P^\star|} s_2 d\mathbb{P}_{X|A=1,Y=0}(x)}{p_1 d\mathbb{P}_X(x)} & , a=1 \\[2ex] p_0 \left[ p_{Y,0} \frac{d\mathbb{P}_{X|A=0,Y=1}(x)}{d\mathbb{P}_X(x)} - (1-p_{Y,0}) \frac{d\mathbb{P}_{X|A=0,Y=0}(x)}{d\mathbb{P}_X(x)} \right] > -\frac{\frac{E^\star}{|E^\star|} s_1 d\mathbb{P}_{X|A=0,Y=1}(x) + \frac{P^\star}{|P^\star|} s_2 d\mathbb{P}_{X|A=0,Y=0}(x)}{p_0 d\mathbb{P}_X(x)} & , a=0 \end{cases}$$

Thus,

$$M(x,a) > s_1 \frac{E^\star}{|E^\star|} H_E(x,a) + s_2 \frac{P^\star}{|P^\star|} H_P(x,a)$$

$$\iff p_a \left[ p_{Y,a} \frac{d\mathbb{P}_{X|A=a,Y=1}(x)}{d\mathbb{P}_X(x)} - (1-p_{Y,a}) \frac{d\mathbb{P}_{X|A=a,Y=0}(x)}{d\mathbb{P}_X(x)} \right] > (2a-1) \frac{\frac{E^\star}{|E^\star|} s_1 d\mathbb{P}_{X|A=a,Y=1}(x) + \frac{P^\star}{|P^\star|} s_2 d\mathbb{P}_{X|A=a,Y=0}(x)}{p_a d\mathbb{P}_X(x)}.$$

$$\iff \frac{p_{Y,a} d\mathbb{P}_{X|A=a,Y=1}(x)}{p_{Y,a} d\mathbb{P}_{X|A=a,Y=1}(x) + (1-p_{Y,a})d\mathbb{P}_{X|A=a,Y=0}(x)} > \frac{p_a^2 p_{Y,a} + (2a-1)\frac{p_{Y,a}}{1-p_{Y,a}} t_2}{2p_a^2 p_{Y,a} + (2a-1)(\frac{p_{Y,a}}{1-p_{Y,a}} t_2 - t_1)}.$$

where $t_1 = 2\frac{E^\star}{|E^\star|} s_1, \quad t_2 = 2\frac{P^\star}{|P^\star|} s_2$.

$$\iff \eta_a(x) > \frac{p_a^2 p_{Y,a} + (2a-1)\frac{p_{Y,a}}{1-p_{Y,a}} t_2}{2p_a^2 p_{Y,a} + (2a-1)(\frac{p_{Y,a}}{1-p_{Y,a}} t_2 - t_1)}.$$

As a result, $f_{s_1,s_2,\tau}(x,a)$ in (9) can be written as

$$f_{t_1,t_2,\tau}(x,a) = \mathbb{1}\{\eta_a(x) > \frac{p_a^2 p_{Y,a} + (2a-1)\frac{p_{Y,a}}{1-p_{Y,a}} t_2}{2p_a^2 p_{Y,a} + (2a-1)(\frac{p_{Y,a}}{1-p_{Y,a}} t_2 - t_1)}\} + a\tau \mathbb{1}\{\eta_a(x) = \frac{p_a^2 p_{Y,a} + (2a-1)\frac{p_{Y,a}}{1-p_{Y,a}} t_2}{2p_a^2 p_{Y,a} + (2a-1)(\frac{p_{Y,a}}{1-p_{Y,a}} t_2 - t_1)}\}.$$

$$(10)$$

Further, the constraint (8) for $f$ in (10) is equivalent to

$$\begin{cases} \frac{E^\star}{|E^\star|} [\mathbb{P}_{X|A=1,Y=1}(\eta_1(x) > \frac{p_1^2 p_{Y,1} + \frac{p_{Y,1}}{1-p_{Y,1}} t_2}{2p_1^2 p_{Y,1} + \frac{p_{Y,1}}{1-p_{Y,1}} t_2 - t_1}) + \tau \mathbb{P}_{X|A=1,Y=1}(\eta_1(x) = \frac{p_1^2 p_{Y,1} + \frac{p_{Y,1}}{1-p_{Y,1}} t_2}{2p_1^2 p_{Y,1} + \frac{p_{Y,1}}{1-p_{Y,1}} t_2 - t_1}) \\ \qquad\qquad\qquad\qquad\qquad - \mathbb{P}_{X|A=0,Y=1}(\eta_0(x) > \frac{p_0^2 p_{Y,0} - \frac{p_{Y,0}}{1-p_{Y,0}} t_2}{2p_0^2 p_{Y,0} - \frac{p_{Y,0}}{1-p_{Y,0}} t_2 + t_1})] \le \alpha_1 \\[3ex] \frac{P^\star}{|P^\star|} [\mathbb{P}_{X|A=1,Y=0}(\eta_1(x) > \frac{p_1^2 p_{Y,1} + \frac{p_{Y,1}}{1-p_{Y,1}} t_2}{2p_1^2 p_{Y,1} + \frac{p_{Y,1}}{1-p_{Y,1}} t_2 - t_1}) + \tau \mathbb{P}_{X|A=1,Y=0}(\eta_1(x) = \frac{p_1^2 p_{Y,1} + \frac{p_{Y,1}}{1-p_{Y,1}} t_2}{2p_1^2 p_{Y,1} + \frac{p_{Y,1}}{1-p_{Y,1}} t_2 - t_1}) \\ \qquad\qquad\qquad\qquad\qquad - \mathbb{P}_{X|A=0,Y=0}(\eta_0(x) > \frac{p_0^2 p_{Y,0} - \frac{p_{Y,0}}{1-p_{Y,0}} t_2}{2p_0^2 p_{Y,0} - \frac{p_{Y,0}}{1-p_{Y,0}} t_2 + t_1})] \le \alpha_2 \end{cases}$$

$$(11)$$

Now, let $\mathcal{T}_{\alpha_1,\alpha_2}$ be the class of Borel functions $f$ that satisfy (8) and $\mathcal{T}_{\alpha_1,\alpha_2,0}$ be the set of $f$-s in $\mathcal{T}_\alpha$ that satisfy (8) with all the inequalities being replaced by equalities.

From the definition of $t_{1,EO,\alpha}^\star$ and $t_{2,EO,\alpha}^\star$, clearly $f_{t_{1,EO,\alpha}^\star, t_{2,EO,\alpha}^\star, \tau}(x,a) \in \mathcal{T}_{\alpha_1,\alpha_2,0}$. Further, we have $s_1^* = t_{1,EO,\alpha}^\star \frac{E^\star}{2|E^\star|} > 0$ and $s_2^* = t_{2,EO,\alpha}^\star \frac{P^\star}{2|P^\star|} > 0$.

Hence, from Generalized Neyman-Pearson lemma, we have:

$$f_{t_{1,EO,\alpha}^\star, t_{2,EO,\alpha}^\star, \tau}(x,a) \in \underset{f \in \mathcal{T}_{\alpha_1,\alpha_2}}{argmax} \int_{\mathcal{A}} \int_{\mathcal{X}} f_{t_1,t_2,\tau}(x,a) M(x,a) d\mathbb{P}_X(x) d\mathbb{P}(a)$$

Now we complete our proof. $\qquad\square$

### A.7 PROOF OF PROPOSITION 3

*Proof.* The classifier is

$$\phi = \begin{cases} \mathbb{1}\{f(x,0) > t^{1,0}_{(k^{1,0})}\}, a = 0 \\ \mathbb{1}\{f(x,1) > t^{1,1}_{(k^{1,1})}\}, a = 1 \end{cases}$$

we have:

$$|DEO(\phi)| = (|F^{1,1}(t^{1,1}_{(k^{1,1})}) - F^{1,0}(t^{1,0}_{(k^{1,0})})|, |F^{0,1}(t^{1,1}_{(k^{1,1})}) - F^{0,0}(t^{1,0}_{(k^{1,0})})|)$$

From Proposition 1, we have $\mathbb{P}(|F^{1,1}(t^{1,1}_{(k^{1,1})}) - F^{1,0}(t^{1,0}_{(k^{1,0})})| > \alpha) \le \mathbb{E}[\sum_{j=k^{1,0}}^{n^{1,0}} \binom{n^{1,0}}{j} (Q^{1,1} - \alpha)^j (1 - (Q^{1,1} - \alpha))^{n^{1,0}-j}] + \mathbb{E}[\sum_{j=k^{1,1}}^{n^{1,1}} \binom{n^{1,1}}{j} (Q^{1,0} - \alpha)^j (1 - (Q^{1,0} - \alpha))^{n^{1,1}-j}]$

Also,

$$\begin{aligned} &\mathbb{P}(|F^{0,1}(t^{1,1}_{(k^{1,1})}) - F^{0,0}(t^{1,0}_{(k^{1,0})})| > \alpha) \\ =&\mathbb{P}(F^{0,1}(t^{1,1}_{(k^{1,1})}) - F^{0,0}(t^{1,0}_{(k^{1,0})}) > \alpha) + \mathbb{P}(F^{0,1}(t^{1,1}_{(k^{1,1})}) - F^{0,0}(t^{1,0}_{(k^{1,0})}) < -\alpha) \\ \triangleq& A + B \end{aligned}$$

And we have
$$\begin{aligned} A =& \mathbb{P}(F^{0,1}(t^{1,1}_{(k^{1,1})}) - F^{0,0}(t^{1,0}_{(k^{1,0})}) > \alpha) \\ =& \mathbb{P}(F^{0,0}(t^{1,0}_{(k^{1,0})}) < F^{0,1}(t^{1,1}_{(k^{1,1})}) - \alpha) \\ \le& \mathbb{P}(F^{0,0}(t^{0,0}_{(k^{0,0})}) < F^{0,1}(t^{0,1}_{(k^{0,1}+1)}) - \alpha) \\ \le& \mathbb{E}[\mathbb{P}(t^{0,0}_{(k^{0,0})} < F^{0,0^{-1}}(F^{0,1}(t^{0,1}_{(k^{0,1}+1)}) - \alpha))\mathbb{1}\{F^{0,1}(t^{0,1}_{(k^{0,1}+1)}) - \alpha > 0\} \mid t^{0,1}_{(k^{0,1}+1)}] \\ =& E\{\mathbb{P}[\text{at least } k^{0,0} \text{ of } t^{0,0}\text{'s are less than } F^{0,0^{-1}}(F^{0,1}(t^{0,1}_{(k^{0,1}+1)}) - \alpha)]\mathbb{1}\{F^{0,1}(t^{0,1}_{(k^{0,1}+1)}) - \alpha > 0\} \mid t^{0,1}_{(k^{0,1}+1)}\} \\ =& E\{\sum_{j=k^{0,0}}^{n^{0,0}} \mathbb{P}[\text{exactly j of the } t^{0,0}\text{'s are less than } F^{0,0^{-1}}(F^{0,1}(t^{0,1}_{(k^{0,1}+1)}) - \alpha)]\mathbb{1}\{F^{0,1}(t^{0,1}_{(k^{0,1}+1)}) - \alpha > 0\} \mid t^{0,1}_{(k^{0,1}+1)}\} \\ =& E\{\sum_{j=k^{0,0}}^{n^{0,0}} \binom{n^{0,0}}{j} \mathbb{P}[t^{0,0} < F^{0,0^{-1}}(F^{0,1}(t^{0,1}_{(k^{0,1}+1)}) - \alpha)]^j (1 - \mathbb{P}[t^{0,0} < F^{0,0^{-1}}(F^{0,1}(t^{0,1}_{(k^{0,1}+1)}) - \alpha)])^{n^{0,0}-j} \\ & \mathbb{1}\{F^{0,1}(t^{0,1}_{(k^{0,1}+1)}) - \alpha > 0\} \mid t^{0,1}_{(k^{0,1}+1)}\} \\ \le& \mathbb{E}[\sum_{j=k^{0,0}}^{n^{0,0}} \binom{n^{0,0}}{j} (F^{0,1}(t^{0,1}_{(k^{0,1}+1)}) - \alpha)^j (1 - (F^{0,1}(t^{0,1}_{(k^{0,1}+1)}) - \alpha))^{n^{0,0}-j} \mid t^{0,1}_{(k^{0,1}+1)}] \end{aligned}$$

Similarly, we have

$$B \le \mathbb{E}[\sum_{j=k^{0,1}}^{n^{0,1}} \binom{n^{0,1}}{j} (F^{0,0}(t^{0,0}_{(k^{0,0}+1)}) - \alpha)^j (1 - (F^{0,0}(t^{0,0}_{(k^{0,0}+1)}) - \alpha))^{n^{0,1}-j} \mid t^{0,0}_{(k^{0,0}+1)}]$$

Hence, we have

$$\begin{aligned} A + B \le& \mathbb{E}[\sum_{j=k^{0,0}}^{n^{0,0}} \binom{n^{0,0}}{j} (F^{0,1}(t^{0,1}_{(k^{0,1}+1)}) - \alpha)^j (1 - (F^{0,1}(t^{0,1}_{(k^{0,1}+1)}) - \alpha))^{n^{0,0}-j} \mid t^{0,1}_{(k^{0,1}+1)}] \\ &+ \mathbb{E}[\sum_{j=k^{0,1}}^{n^{0,1}} \binom{n^{0,1}}{j} (F^{0,0}(t^{0,0}_{(k^{0,0}+1)}) - \alpha)^j (1 - (F^{0,0}(t^{0,0}_{(k^{0,0}+1)}) - \alpha))^{n^{0,1}-j} \mid t^{0,0}_{(k^{0,0}+1)}] \end{aligned}$$

Since $F^{0,a}(t^{0,a}_{(k^{0,a}+1)})$ is stochastically dominated by $Beta(k^{0,a}+1, n^{0,a} - k^{0,a})$, we complete the proof. □

## A.8 ALGORITHMS FOR OTHER GROUP FAIRNESS CONSTRAINTS

In addition to Equality of Opportunity and Equalized Odds, there are other common fairness constraints and we can extend FaiREE to them.

**Definition 4** (Demographic Parity). *A classifier satisfies Demographic Parity if its prediction $\widehat{Y}$ is statistically independent of the sensitive attribute $A$ :*

$$\mathbb{P}(\widehat{Y} = 1 \mid A = 1) = \mathbb{P}(\widehat{Y} = 1 \mid A = 0)$$

**Definition 5** (Predictive Equality). *A classifier satisfies Predictive Equality if it achieves the same TNR (or FPR) among protected groups:*

$$\mathbb{P}_{X|A=1,Y=0}(\widehat{Y} = 1) = \mathbb{P}_{X|A=0,Y=0}(\widehat{Y} = 1)$$

**Definition 6** (Equalized Accuracy). *A classifier satisfies Equalized Accuracy if its mis-classification error is statistically independent of the sensitive attribute $A$:*

$$\mathbb{P}(\widehat{Y} \neq Y \mid A = 1) = \mathbb{P}(\widehat{Y} \neq Y \mid A = 0)$$

Similar to $DEOO$, we can define the following measures:

$$DDP = \mathbb{P}_{X|A=1}(\widehat{Y} = 1) - \mathbb{P}_{X|A=0}(\widehat{Y} = 1) \tag{12}$$

$$DPE = \mathbb{P}_{X|A=1,Y=0}(\widehat{Y} = 1) - \mathbb{P}_{X|A=0,Y=0}(\widehat{Y} = 1) \tag{13}$$

$$DEA = \mathbb{P}(\widehat{Y} \neq Y \mid A = 1) - \mathbb{P}(\widehat{Y} \neq Y \mid A = 0). \tag{14}$$

### A.8.1 FAIREE FOR DEMOGRAPHIC PARITY

---

**Algorithm 3:** FaiREE for Demographic Parity

**Input:**
Training data: $S = S^{0,0} \cup S^{0,1} \cup S^{1,0} \cup S^{1,1}$
$\alpha$: error bound
$\delta$: small tolerance level
$f$: a classifier

1  $T^{y,a} = \{f(x^{y,a}_1), \ldots, f(x^{y,a}_{n_{y,a}})\}$
2  $\{t^{y,a}_{(1)}, \ldots, t^{y,a}_{(n_{y,a})}\}$ =sort($T^{y,a}$)
3  $T^y = T^{y,0} \cup T^{y,1}$
4  $\{t^y_{(1)}, \ldots, t^y_{(n_y)}\}$ =sort($T^y$)
5  Define $g(k,a) = \mathbb{E}[\sum_{j=k}^{n^a} \binom{n^a}{j}(Q^{1-a} - \alpha)^j (1 - (Q^{1-a} - \alpha))^{n^a-j}]$ with
   $Q^a \sim Beta(k, n^a - k + 1)$, $L(k^0, k^1) = g(k^0, 0) + g(k^1, 1)$
6  Build candidate set $K = \{(k^0, k^1) \mid L(k^0, k^1) \leq \delta\} = \{(k^0_1, k^1_1), \ldots, (k^0_M, k^1_M)\}$
7  Find $k^{y,0}_i$: $t^{y,0}_{(k^{y,0}_i)} \leq t^0_{(k^0_i)} < t^{y,0}_{(k^{y,0}_i+1)}, t^{y,1}_{(k^{y,1}_i)} \leq t^1_{(k^1_i)} < t^{y,1}_{(k^{y,1}_i+1)}, y \in \{0,1\}$
8  $i_* \leftarrow \underset{i \in [M]}{\arg\min}\{\hat{e}_i\}$ ($\hat{e}_i$ is defined in Proposition 2)

**Output:** $\hat{\phi}(x,a) = \mathbb{1}\{f(x,a) > t^a_{(k^a_{i_*})}\}$

---

Similar to the algorithm for Equality of Opportunity, we have the following propositions and assumption:

**Proposition 4.** *Given $k^0, k^1$ satisfying $k^a \in \{1, \ldots, n^a\}$ $(a = 0, 1)$. Define $\phi(x, a) = \mathbb{1}\{f(x, a) > t_{(k^a)}^a\}$, $g(k, a) = \mathbb{E}[\sum_{j=k}^{n^a} \binom{n^a}{j}(Q^{1-a} - \alpha)^j(1 - (Q^{1-a} - \alpha))^{n^a - j}]$ with $Q^a \sim Beta(k, n^a - k + 1)$, then we have:*

$$\mathbb{P}(|DDP(\phi)| > \alpha) \leq g(k^0, 0) + g(k^1, 1).$$

*If $t^a$ is continuous random variable, the equality holds.*

**Theorem 7.** *If $\min\{n^0, n^1\} \geq \lceil \frac{\log \frac{\delta}{2}}{\log(1-\alpha)} \rceil$, we have $|DDP(\hat{\phi}_i)| < \alpha$ with probability $1 - \delta$, for each $i \in \{1, \ldots, M\}$.*

**Theorem 8.** *Given $\alpha' < \alpha$. If $\min\{n^0, n^1\} \geq \lceil \frac{\log \frac{\delta}{2}}{\log(1-\alpha)} \rceil$. Suppose $\hat{\phi}$ is the final output of FaiREE, we have:*
*(1) $|DDP(\hat{\phi})| \leq \alpha$ with probability $(1 - \delta)^M$, where $M$ is the size of the candidate set.*
*(2) Suppose the density distribution functions of $f^*$ under $A = a, Y = 1$ are continuous. $\phi_{DDP,\alpha}^* = \arg\min_{|DDP(\phi)| \leq \alpha} \mathbb{P}(\phi(x, a) \neq Y)$. When the input classifier $f$ satisfies $\|f(x, a) - f^*(x, a)\|_\infty \leq \epsilon_0$, for any $\epsilon > 0$ such that $F_{(+)}^*(\epsilon) \leq \frac{\alpha - \alpha'}{2} - F_{(+)}^*(2\epsilon_0)$, we have*

$$| \mathbb{P}(\hat{\phi}(x, a) \neq Y) - \mathbb{P}(\phi_{\alpha'}^*(x, a) \neq Y) | \leq 2F_{(+)}^*(2\epsilon_0) + 2F_{(+)}^*(\epsilon) + 2\epsilon^3 + 12\epsilon^2 + 16\epsilon$$

*with probability $1 - (2M + 4)(e^{-2n^{1,0}\epsilon^2} + e^{-2n^{1,1}\epsilon^2} + e^{-2n^{0,0}\epsilon^2} + e^{-2n^{0,1}\epsilon^2})$.*

### A.8.2 FaiREE for Predictive Equality

---
**Algorithm 4:** FaiREE for Predictive Opportunity

---
**Input:**
Training data: $S = S^{0,0} \cup S^{0,1} \cup S^{1,0} \cup S^{1,1}$
$\alpha$: error bound
$\delta$: small tolerance level
$f$: a classifier
1  $T^{y,a} = \{f(x_1^{y,a}), \ldots, f(x_{n_{y,1}}^{y,a})\}$
2  $\{t_{(1)}^{y,a}, \ldots, t_{(n_{y,a})}^{y,a}\}$ =sort$(T^{y,a})$
3  Define $g_0(k, a) = \mathbb{E}[\sum_{j=k}^{n^{0,a}} \binom{n^{0,a}}{j}(Q^{0,1-a} - \alpha)^j(1 - (Q^{0,1-a} - \alpha))^{n^{0,a} - j}]$ with
   $Q^{0,a} \sim Beta(k, n^{0,a} - k + 1)$, $L(k^{0,0}, k^{0,1}) = g_0(k^{0,0}, 0) + g_0(k^{0,1}, 1)$
4  Build candidate set $K = \{(k^{0,0}, k^{0,1}) \mid L(k^{0,0}, k^{0,1}) \leq \delta\} = \{(k_1^{0,0}, k_1^{0,1}), \ldots, (k_M^{0,0}, k_M^{0,1})\}$
5  Find $k_i^{1,0}, k_i^{1,1}$: $t_{(k_i^{1,0})}^{1,0} \leq t_{(k_i^{0,0})}^{0,0} < t_{(k_i^{1,0}+1)}^{1,0}$, $t_{(k_i^{1,1})}^{1,1} \leq t_{(k_i^{0,1})}^{0,1} < t_{(k_i^{1,1}+1)}^{1,1}$
6  $i_* \leftarrow \arg\min_{i \in [M]}\{\hat{e}_i\}$ ($\hat{e}_i$ is defined in Proposition 2)
   **Output:** $\hat{\phi}(x, a) = \mathbb{1}\{f(x, a) > t_{(k_{i_*}^{0,a})}^{0,a}\}$

---

Similar to the algorithm for Equality of Opportunity, we have the following propositions and assumption:

**Proposition 5.** *Given $k^{0,0}, k^{0,1}$ satisfying $k^{1,a} \in \{1, \ldots, n^{0,a}\}$ $(a = 0, 1)$. Define $\phi(x, a) = \mathbb{1}\{f(x, a) > t_{(k^{0,a})}^{0,a}\}$, $g_0(k, a) = \mathbb{E}[\sum_{j=k}^{n^{0,a}} \binom{n^{0,a}}{j}(Q^{0,1-a} - \alpha)^j(1 - (Q^{0,1-a} - \alpha))^{n^{0,a} - j}]$ with $Q^{0,a} \sim Beta(k, n^{0,a} - k + 1)$, then we have:*

$$\mathbb{P}(|DPE(\phi)| > \alpha) \leq g_0(k^{0,0}, 0) + g_0(k^{0,1}, 1).$$

*If $t^{0,a}$ is continuous random variable, the equality holds.*

**Theorem 9.** *If $\min\{n^{0,0}, n^{0,1}\} \geq \lceil \frac{\log \frac{\delta}{2}}{\log(1-\alpha)} \rceil$, we have $|DPE(\hat{\phi}_i)| < \alpha$ with probability $1 - \delta$, for each $i \in \{1, \ldots, M\}$.*

**Theorem 10.** *Given $\alpha' < \alpha$. If $\min\{n^{0,0}, n^{0,1}\} \geq \lceil \frac{\log \frac{\delta}{2}}{\log(1-\alpha)} \rceil$. Suppose $\hat{\phi}$ is the final output of FaiREE, we have:*
*(1) $|DPE(\hat{\phi})| \leq \alpha$ with probability $(1-\delta)^M$, where $M$ is the size of the candidate set.*
*(2) Suppose the density distribution functions of $f^*$ under $A = a, Y = 1$ are continuous. $\phi^*_{DPE,\alpha} = \arg\min_{|DPE(\phi)| \leq \alpha} \mathbb{P}(\phi(x,a) \neq Y)$. When the input classifier $f$ satisfies $\|f(x,a) - f^*(x,a)\|_\infty \leq \epsilon_0$, for any $\epsilon > 0$ such that $F^*_{(+)}(\epsilon) \leq \frac{\alpha - \alpha'}{2} - F^*_{(+)}(2\epsilon_0)$, we have*

$$| \mathbb{P}(\hat{\phi}(x,a) \neq Y) - \mathbb{P}(\phi^*_{\alpha'}(x,a) \neq Y) | \leq 2F^*_{(+)}(2\epsilon_0) + 2F^*_{(+)}(\epsilon) + 2\epsilon^3 + 12\epsilon^2 + 16\epsilon$$

*with probability $1 - (2M + 4)(e^{-2n^{1,0}\epsilon^2} + e^{-2n^{1,1}\epsilon^2}) - (1 - F^{0,0}_{(-)}(2\epsilon))^{n^{0,0}} - (1 - F^{0,1}_{(-)}(2\epsilon))^{n^{0,1}}$.*

### A.8.3 FaiREE for Equalized Accuracy

---

**Algorithm 5:** FaiREE for Equalized Accuracy

---

**Input:**
Training data: $S = S^{0,0} \cup S^{0,1} \cup S^{1,0} \cup S^{1,1}$
$\alpha$: error bound ($\alpha > |p_{Y,1} - p_{Y,0}|$)
$\delta$: small tolerance level
$f$: a classifier

1 $T^{y,a} = \{f(x_1^{y,a}), \ldots, f(x_{n_{y,a}}^{y,a})\}$
2 $\{t_{(1)}^{y,a}, \ldots, t_{(n_{y,a})}^{y,a}\}$ =sort($T^{y,a}$)
3 Define $g_1(k, a) = \mathbb{E}[\sum_{j=k}^{n^{1,a}} \binom{n^{1,a}}{j} (\frac{p_{Y,1-a}Q^{1,1-a} - \alpha}{p_{Y,a}})^j (1 - \frac{p_{Y,1-a}Q^{1,1-a} - \alpha}{p_{Y,a}})^{n^{1,a}-j}]$ with

    $Q^{1,a} \sim Beta(k, n^{1,a} - k + 1), L_1(k^{1,0}, k^{1,1}) = g_1(k^{1,1}, 1) + g_1(k^{1,0}, 0)$
4 Define $g_0(k, a) =$

    $\mathbb{E}[\sum_{j=k}^{n^{0,a}} \binom{n^{0,a}}{j} (\frac{(1-p_{Y,1-a})Q^{0,1-a} + p_{Y,1-a} - p_{Y,a} - \alpha}{1 - p_{Y,0}})^j (1 - \frac{(1-p_{Y,1-a})Q^{0,1-a} + p_{Y,1-a} - p_{Y,a} - \alpha}{1 - p_{Y,a}})^{n^{0,a}-j}]$

    with $Q^{0,a} \sim Beta(k + 1, n^{0,a} - k), L_0(k^{0,0}, k^{0,1}) = g_0(k^{0,1}, 1) + g_0(k^{0,0}, 0)$
5 Find $k^{0,0}, k^{0,1}$: $t_{(k^{0,0})}^{0,0} \leq t_{(k^{1,0})}^{1,0} < t_{(k^{0,0}+1)}^{0,0}, t_{(k^{0,1})}^{0,1} \leq t_{(k^{1,1})}^{1,1} < t_{(k^{0,1}+1)}^{0,1}$.
6 Build candidate set

    $K = \{(k^{1,0}, k^{1,1}) \mid L_1(k^{1,0}, k^{1,1}) + L_0(k^{0,0}, k^{0,1}) \leq \delta\} = \{(k_1^{1,0}, k_1^{1,1}), \ldots, (k_M^{1,0}, k_M^{1,1})\}$
7 $i_* \leftarrow \arg\min_{i \in [M]}\{\hat{e}_i\}$ ($\hat{e}_i$ is defined in Proposition 2)

**Output:** $\hat{\phi}(x, a) = \mathbb{1}\{f(x,a) > t_{(k_{i_*}^{1,a})}^{1,a}\}$

---

Similar to the algorithm for Equalized Odds, we have the following propositions and assumption:

**Proposition 6.** *Given $k^{1,0}, k^{1,1}$ satisfying $k^{1,a} \in \{1, \ldots, n^{1,a}\}$ ($a = 0, 1$) and $\alpha > |p_{Y,1} - p_{Y,0}|$. Define $\phi(x, a) = \mathbb{1}\{f(x,a) > t_{(k^{1,a})}^{1,a}\}$, $g_1(k, a) = \mathbb{E}[\sum_{j=k}^{n^{1,a}} \binom{n^{1,a}}{j} (\frac{p_{Y,1-a}Q^{1,1-a} - \alpha}{p_{Y,a}})^j (1 - \frac{p_{Y,1-a}Q^{1,1-a} - \alpha}{p_{Y,a}})^{n^{1,a}-j}]$ with $Q^{1,a} \sim Beta(k, n^{1,a} - k + 1)$, $q_{Y,a}(\alpha) = \frac{(1-p_{Y,1-a})Q^{0,1-a} + p_{Y,1-a} - p_{Y,a} - \alpha}{1 - p_{Y,a}}$, and $g_0(k, a) = \mathbb{E}[\sum_{j=k}^{n^{0,a}} \binom{n^{0,a}}{j} (q_{Y,a}(\alpha))^j (1 - q_{Y,a}(\alpha))^{n^{0,a}-j}]$ with $Q^{0,a} \sim Beta(k + 1, n^{0,a} - k)$. Then we have:*

$$\mathbb{P}(|DPE(\phi)| > \alpha) \leq g_1(k^{1,1}, 1) + g_1(k^{1,0}, 0) + g_0(k^{0,1}, 1) + g_0(k^{0,0}, 0).$$

**Assumption 1.** $n^{0,0} \geq \lceil \frac{\log \frac{\delta}{4}}{\log(1 - \frac{\alpha}{1-p_{Y,0}})} \rceil, n^{0,1} \geq \lceil \frac{\log \frac{\delta}{4}}{\log(1 - \frac{\alpha}{1-p_{Y,1}})} \rceil, n^{1,0} \geq \lceil \frac{\log \frac{\delta}{4}}{\log(\frac{p_{Y,1} - \alpha}{p_{Y,0}})} \rceil, n^{1,1} \geq \lceil \frac{\log \frac{\delta}{4}}{\log(\frac{p_{Y,0} - \alpha}{p_{Y,1}})} \rceil$, *in which $\lceil \cdot \rceil$ denotes the ceiling function.*

**Theorem 11.** *Under Assumption 1, we have $|DEA(\hat{\phi}_i)| \leq \alpha$ with probability $1 - \delta$, for each $i \in \{1, \ldots, M\}$.*

**Corollary 1.** *Under Assumption 1, we have $|DEA(\hat{\phi})| \leq \alpha$ with probability $(1-\delta)^M$, where $M$ is the size of the candidate set.*

## A.9 IMPLEMENTATION DETAILS AND ADDITIONAL EXPERIMENTS

From Lemma 2, we adopt a new way of building a much smaller candidate set. Note that our shrunk candidate set for Equality of Opportunity is:

$$K' = \{(k^{1,0}, u_1(k^{1,0})) \mid L_1(k^{1,0}, u_1(k^{1,0})) \leq \delta\}.$$

Since Equalized Odds constraint is an extension of Equality of Opportunity, our target classifier should be in $K'$.

To select our target classifier, it's sufficient to add a condition of similar false positive rate between privileged and unprivileged groups. Specifically, we choose our final candidate set as below:

$$K'' = \{(k^{1,0}, u_1(k^{1,0})) \mid L_1(k^{1,0}, u_1(k^{1,0})) \leq \delta, L_0(k^{0,0}, k^{1,0}) \leq \delta\}.$$

We also did experiments on other benchmark datasets.

First, we apply FaiREE to German Credit dataset Kamiran & Calders (2009), whose task is to predict whether a bank account holder's credit is *good* or *bad*. The protected attribute is gender, and the sample size is 1000, with 800 training samples and 200 test samples. To facilitate the numerical study, we randomly split data into training set, calibration set, and test set at each repetition and repeat 500 times.

**Table 5:** Result of different methods on German Credit dataset

| | Eq | C-Eq | ROC | FairBayes | | | FaiREE-EOO | | | FaiREE-EO | | |
|---|---|---|---|---|---|---|---|---|---|---|---|---|
| $\alpha$ | / | / | / | 0.07 | 0.1 | 0.14 | 0.07 | 0.1 | 0.14 | 0.07 | 0.1 | 0.14 |
| $\overline{|DEOO|}$ | 0.078 | 0.093 | 0.100 | 0.126 | 0.125 | 0.126 | 0.020 | 0.029 | 0.034 | 0.001 | 0.025 | 0.048 |
| $|DEOO|_{95}$ | 0.179 | 0.127 | 0.267 | 0.160 | 0.182 | 0.186 | 0.066 | 0.097 | 0.130 | 0.004 | 0.084 | 0.120 |
| $\overline{|DPE|}$ | 0.109 | 0.072 | 0.138 | / | / | / | / | / | / | 0.041 | 0.063 | 0.092 |
| $|DPE|_{95}$ | 0.235 | 0.114 | 0.334 | / | / | / | / | / | / | 0.059 | 0.097 | 0.133 |
| $\overline{ACC}$ | 0.707 | 0.720 | 0.591 | 0.722 | 0.723 | 0.723 | 0.717 | 0.729 | 0.745 | 0.702 | 0.721 | 0.722 |

Then, we apply FaiREE to Compas Score dataset Angwin et al. (2016), whose task is to predict whether a person will conduct crime in the future. The protected attribute is gender, and the sample size is 5278, with 4222 training samples and 1056 test samples. To facilitate the numerical study, we randomly split data into training set, calibration set and test set at each repetition and repeat for 500 times.

**Table 6:** Result of different methods on Compas Score dataset

| | Eq | C-Eq | ROC | FairBayes | | | FaiREE-EOO | | | FaiREE-EO | | |
|---|---|---|---|---|---|---|---|---|---|---|---|---|
| $\alpha$ | / | / | / | 0.07 | 0.1 | 0.14 | 0.07 | 0.1 | 0.14 | 0.07 | 0.1 | 0.14 |
| $\overline{|DEOO|}$ | 0.083 | 0.642 | 0.070 | 0.077 | 0.109 | 0.136 | 0.026 | 0.033 | 0.042 | 0.027 | 0.049 | 0.098 |
| $|DEOO|_{95}$ | 0.101 | 0.684 | 0.174 | 0.145 | 0.186 | 0.250 | 0.066 | 0.097 | 0.131 | 0.068 | 0.090 | 0.140 |
| $\overline{|DPE|}$ | 0.025 | 0.291 | 0.067 | / | / | / | / | / | / | 0.026 | 0.048 | 0.060 |
| $|DPE|_{95}$ | 0.047 | 0.332 | 0.148 | / | / | / | / | / | / | 0.057 | 0.092 | 0.114 |
| $\overline{ACC}$ | 0.629 | 0.664 | 0.652 | 0.658 | 0.658 | 0.659 | 0.654 | 0.660 | 0.672 | 0.623 | 0.654 | 0.669 |

We further generate a synthetic model where trained classifiers are more informative.

**Model 3.** We generate the protected attribute $A$ and label $Y$ with the probability, location parameter and scale parameter the same as Model 1. The dimension of features is set to 60, and we

generate features with $x_{i,j}^{0,0} \overset{i.i.d.}{\sim} t(1)$, $x_{i,j}^{0,1} \overset{i.i.d.}{\sim} t(4)$, $x_{i,j}^{1,0} \overset{i.i.d.}{\sim} \chi_1^2$ and $x_{i,j}^{1,1} \overset{i.i.d.}{\sim} \chi_4^2$, for $j = 1, 2, ..., 60$.

**Table 7:** Experimental studies under Model 3. Here $\overline{|DEOO|}$ denotes the sample average of the absolute value of $DEOO$ defined in Eq. (1), and $|DEOO|_{95}$ denotes the sample upper 95% quantile. $\overline{|DPE|}$ and $|DPE|_{95}$ are defined similarly for $DPE$ defined in Eq. (13). $\overline{ACC}$ is the sample average of accuracy. We use "/" in the $DPE$ line because FairBayes and FaiREE-EOO are not designed to control $DPE$.

|  | Eq | C-Eq | ROC | FairBayes | | | FaiREE-EOO | | | FaiREE-EO | | |
|---|---|---|---|---|---|---|---|---|---|---|---|---|
| $\alpha$ | / | / | / | 0.04 | 0.06 | 0.08 | 0.04 | 0.06 | 0.08 | 0.04 | 0.06 | 0.08 |
| $\overline{|DEOO|}$ | 0.041 | 0.020 | 0.034 | 0.048 | 0.062 | 0.076 | 0.018 | 0.025 | 0.033 | 0.015 | 0.032 | 0.034 |
| $|DEOO|_{95}$ | 0.115 | 0.042 | 0.070 | 0.080 | 0.113 | 0.143 | 0.038 | 0.054 | 0.076 | 0.036 | 0.058 | 0.069 |
| $\overline{|DPE|}$ | 0.093 | 0.106 | 0.062 | / | / | / | / | / | / | 0.026 | 0.034 | 0.046 |
| $|DPE|_{95}$ | 0.272 | 0.191 | 0.101 | / | / | / | / | / | / | 0.039 | 0.055 | 0.077 |
| $\overline{ACC}$ | 0.887 | 0.921 | 0.834 | 0.898 | 0.901 | 0.912 | 0.946 | 0.950 | 0.963 | 0.900 | 0.914 | 0.933 |

### A.9.1 COMPARISON WITH MORE ALGORITHMS

In this subsection, we further compare FaiREE with more baseline algorithms that are designed for achieving Equalized Odds or Equality of Opportunity, including pre-processing algorithms (Fairdecision in Kilbertus et al. (2020) and LAFTR in Madras et al. (2018)) and in-processing algorithms (Meta-cl in Celis et al. (2019) and Adv-debias in Zhang et al. (2018)) under synthetic settings Model 1 and Model 2 described in Section 5.1, and the real dataset Adult Census in Section 5.2. The results are summarized in Tables 8, 9, and 10.

From the experimental results, we can find that FaiREE has favorable results over these baseline methods, with respect to fairness and accuracy. In particular, the experimental results indicate that while the baseline methods are designed to minimize the fairness violation as much as possible (i.e. set $\alpha = 0$), these methods are unable to have an exact control of the fairness violation to a desired level $\alpha$. For example, in the analysis of Adult Census dataset, the 95% quantile of the DEOO fairness violations of Fairdecision is 0.078, and that of LAFTR, Meta-cl and Adv-debias are all above 0.2. Moreover, our results found that If we allow the same fairness violation of DEOO and DEP for our proposed method FaiREE, we have a much higher accuracy (0.845) compared to the accuracy of those four baseline methods.

**Table 8:** Results of different methods on Adult Census dataset. Here $\overline{|DEOO|}$ denotes the sample average of the absolute value of $DEOO$ defined in Eq. (1), and $|DEOO|_{95}$ denotes the sample upper 95% quantile. $\overline{|DPE|}$ and $|DPE|_{95}$ are defined similarly for $DPE$ defined in Eq. (13). $\overline{ACC}$ is the sample average of accuracy. We use "/" in the $DPE$ line because Fairdecision and FaiREE-EOO are not designed to control $DPE$.

|  | Fairdecision | LAFTR | Meta-cl | Adv-debias | FaiREE-EOO | | | FaiREE-EO | | |
|---|---|---|---|---|---|---|---|---|---|---|
| $\alpha$ | / | / | / | / | 0.07 | 0.1 | 0.14 | 0.07 | 0.1 | 0.14 |
| $\overline{|DEOO|}$ | 0.041 | 0.124 | 0.172 | 0.199 | 0.034 | 0.039 | 0.066 | 0.002 | 0.039 | 0.067 |
| $|DEOO|_{95}$ | 0.078 | 0.203 | 0.253 | 0.248 | 0.065 | 0.090 | 0.124 | 0.008 | 0.094 | 0.125 |
| $\overline{|DPE|}$ | / | 0.044 | 0.194 | 0.074 | / | / | / | 0.030 | 0.066 | 0.074 |
| $|DPE|_{95}$ | / | 0.083 | 0.271 | 0.094 | / | / | / | 0.056 | 0.078 | 0.086 |
| $\overline{ACC}$ | 0.772 | 0.822 | 0.688 | 0.791 | 0.845 | 0.846 | 0.847 | 0.512 | 0.845 | 0.846 |

**Table 9:** Experimental studies under Model 1, with the same notation as Table 8

|  | Fairdecision | LAFTR | Meta-cl | Adv-debias | FaiREE-EOO | | | FaiREE-EO | | |
|---|---|---|---|---|---|---|---|---|---|---|
| $\alpha$ | / | / | / | / | 0.08 | 0.12 | 0.16 | 0.08 | 0.12 | 0.16 |
| $\|DEOO\|$ | 0.072 | 0.081 | 0.028 | 0.062 | 0.028 | 0.046 | 0.063 | 0.025 | 0.031 | 0.042 |
| $\|DEOO\|_{95}$ | 0.177 | 0.145 | 0.108 | 0.226 | 0.073 | 0.115 | 0.157 | 0.079 | 0.108 | 0.133 |
| $\overline{\|DPE\|}$ | / | 0.061 | 0.118 | 0.179 | / | / | / | 0.039 | 0.042 | 0.045 |
| $\|DPE\|_{95}$ | / | 0.104 | 0.272 | 0.412 | / | / | / | 0.075 | 0.084 | 0.106 |
| $\overline{ACC}$ | 0.616 | 0.533 | 0.620 | 0.645 | 0.621 | 0.657 | 0.669 | 0.552 | 0.562 | 0.615 |

**Table 10:** Experimental studies under Model 2, with the same notation as Table 8.

|  | Fairdecision | LAFTR | Meta-cl | Adv-debias | FaiREE-EOO | | | FaiREE-EO | | |
|---|---|---|---|---|---|---|---|---|---|---|
| $\alpha$ | / | / | / | / | 0.08 | 0.12 | 0.16 | 0.08 | 0.12 | 0.16 |
| $\|DEOO\|$ | 0.675 | 0.450 | 0.094 | 0.096 | 0.027 | 0.047 | 0.073 | 0.028 | 0.035 | 0.047 |
| $\|DEOO\|_{95}$ | 0.744 | 0.633 | 0.208 | 0.263 | 0.075 | 0.112 | 0.153 | 0.077 | 0.114 | 0.143 |
| $\overline{\|DPE\|}$ | / | 0.502 | 0.120 | 0.140 | / | / | / | 0.041 | 0.044 | 0.056 |
| $\|DPE\|_{95}$ | / | 0.686 | 0.312 | 0.418 | / | / | / | 0.071 | 0.090 | 0.127 |
| $\overline{ACC}$ | 0.584 | 0.647 | 0.606 | 0.628 | 0.595 | 0.627 | 0.639 | 0.575 | 0.589 | 0.606 |

## A.10   SUPPLEMENTARY FIGURES

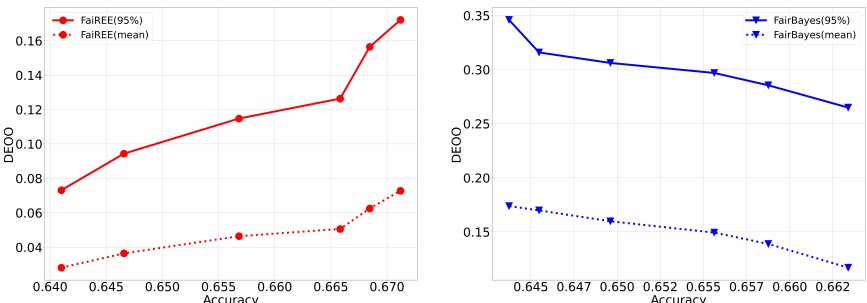

**Figure 3:** DEOO v.s. Accuracy, as a complementary figure for Figure 1

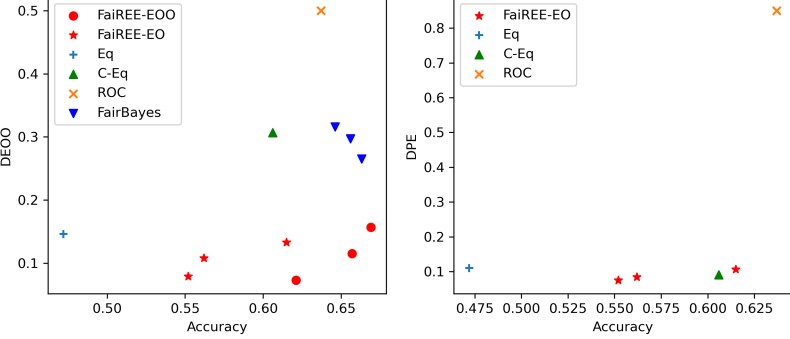

**Figure 4:** DEOO v.s. Accuracy & DPE v.s. Accuracy for Model 1

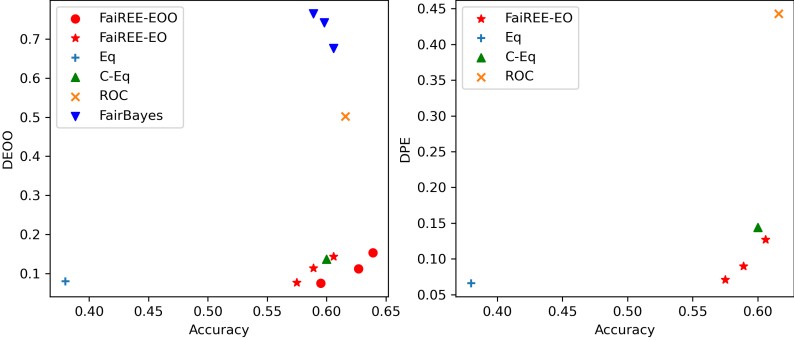

**Figure 5:** DEOO v.s. Accuracy & DPE v.s. Accuracy for Model 2

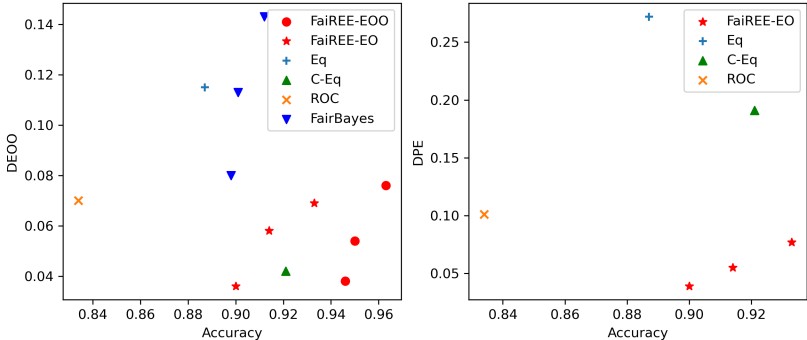

**Figure 6:** DEOO v.s. Accuracy & DPE v.s. Accuracy for Model 3

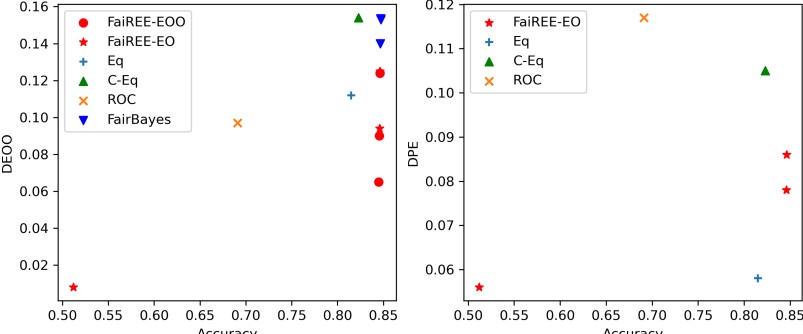

**Figure 7:** DEOO v.s. Accuracy & DPE v.s. Accuracy for Adult Census dataset

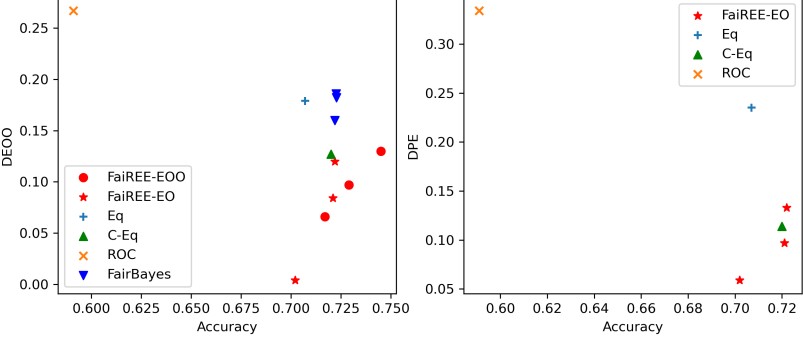

**Figure 8:** DEOO v.s. Accuracy & DPE v.s. Accuracy for German Credit dataset

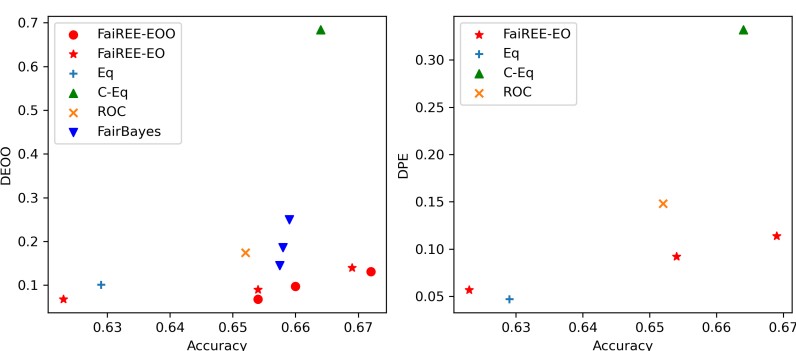

**Figure 9:** DEOO v.s. Accuracy & DPE v.s. Accuracy for Compas Score dataset

