# OpenReview forum: "FaiREE: fair classification with finite-sample and distribution-free guarantee"
_ICLR.cc/2023/Conference — ICLR 2023 poster_

### Official Review · Reviewer_6zkG · 2022-10-25

**Confidence:** 4
**Correctness:** 3
**Technical Novelty And Significance:** 3
**Empirical Novelty And Significance:** 3
**Recommendation:** 8

**Clarity, Quality, Novelty And Reproducibility:**

The contribution of the paper is novel and significant, and the paper is well-written.

**Strength And Weaknesses:**

Strength:

The proposed post-processing algorithm satisfies two theoretical guarantees (Thm 2 and 3), i.e., a) The output classifier satisfies fairness constraint with high probability, and b) if the input classifier is optimal, the output classifier will be near optimal with high probability. To my knowledge, these theoretical guarantees are novel and critical in trustworthy machine-learning applications.


Weaknesses:

The second guarantee of the proposed post-processing algorithm in Theorem 2 requires a strong assumption that the input classifier is Bayes-optimal. Is there a way to bound the performance of the proposed algorithm with any classifier $f$ using the approximation error between $f$ and $f^*$?

In addition, if the classifier $f$ is first trained using some dataset $S_{train}$, all the theoretical results only hold when the dataset $S$ used in the post-processing is independent of $S_{train}$. Is it correct? Thus, a train/validation/test split is still needed in this setting, and the sample complexity results in table 2 should be interpreted as the validation sample complexity since it does not include the sample needed for training $f$.

Minor comments:

Figure 1 is really difficult for readers to digest at the beginning of the paper. Readers need to understand $\alpha$ and DEOO to appreciate the results, so maybe deferring it to later sections will be better. Also, figures are more readable than tables, why not include more figures in the main body and move those tables to the appendix?

Typo: 2 line above Paper Organization, should be “group” instead of “grup”


**Summary Of The Paper:**

This paper proposes a new post-processing algorithm (FaiREE) to satisfy group fairness constraints in classification with finite-sample and distribution-free theoretical guarantees. FaiREE can be adapted to satisfy various group fairness notions, e.g., Equality of Opportunity, Equalized Odds, and Demographic Parity. Synthetic and real data experiments further support these theoretical guarantees.

**Summary Of The Review:**

This paper tackles an interesting problem with critical societal impacts, and I would recommend accepting the paper.

---

> ### Author Response · Authors · 2022-11-14
> **Response to Reviewer 6zkG**
>
> Thank you for your positive review and your constructive comments and suggestions. We have revised our paper according to your comments. We respond to your questions below and would appreciate it if you could let us know if our response addresses your concerns.
>
> >**Q1**: The second guarantee of the proposed post-processing algorithm in Theorem 2 requires a strong assumption that the input classifier is Bayes-optimal. Is there a way to bound the performance of the proposed algorithm with any classifier f using the approximation error between f and f∗?
>
> **A1**: Thank you for pointing this out. Our analysis can be directly extended to the case where the input is close to the Bayes-optimal. We have modified the statement and proof for both Theorems 2 and 4. Please see Page 6 and 7 of the revised paper.
>
> ---
>
> >**Q2**: In addition, if the classifier $f$ is first trained using some dataset Strain, all the theoretical results only hold when the dataset S used in the post-processing is independent of Strain. Is it correct? Thus, a train/validation/test split is still needed in this setting, and the sample complexity results in table 2 should be interpreted as the validation sample complexity since it does not include the sample needed for training $f$.
>
> **A2**: Thank you for your valuable comment! We would like to clarify that the minimum sample size requirement is needed to make any given $f$ to satisfy the fairness constraint (i.e., part (i) in Theorem 2 or 4). Your comment is right when we care about accuracy (i.e., the part (2) in Theorem 2 or 4), in which case we could split the data: use the first half as the training set to find an $f$ that is close to $f^*$, and use the second half as the calibration set. In light of your suggestions, we add more clarifications in Section 4.2, hoping to make the point clearer.
>
> ---
>
> >**Q3**: Figure 1 is really difficult for readers to digest at the beginning of the paper. Readers need to understand α and DEOO to appreciate the results, so maybe deferring it to later sections will be better. Also, figures are more readable than tables, why not include more figures in the main body and move those tables to the appendix?
>
> **A3**: Thank you for your suggestion. We have changed Figure 1 to be more readable by aligning the colors and also substituting the right part of the figure with a trade-off plot between fairness and accuracy. For all the tables in the paper, we have also visualized them as fairness-accuracy trade-off plots. For the reason of page limit, we temporarily put them in the appendix (in Pages 31 and 32).
>
> ---
>
> >**Q4**: Typo: 2 lines above Paper Organization, should be “group” instead of “grup”
>
> **A4**:  Thank you for your careful reading. We have corrected this typo.
>
> ---

---

### Official Review · Reviewer_sJJW · 2022-10-25

**Confidence:** 4
**Correctness:** 4
**Technical Novelty And Significance:** 3
**Empirical Novelty And Significance:** 3
**Recommendation:** 6

**Clarity, Quality, Novelty And Reproducibility:**

Clarity: The writing is good.
Quality: The theoretical guarantee is good, but the empirical results are not that powerful.
Novelty: The proposed approach is interesting and novel.

**Strength And Weaknesses:**

Strength:
- The considered problem of fair classification with finite samples is relevant.
- The approach is interesting that includes candidate set construction and selection.
- The empirical results show that previous fair classification has limitations with finite samples.

Weaknesses:
- The empirical performance of FairREE does not have an advantage compared to baselines. For example, in Figure 1, FairBayes can achieve higher accuracy than FairREE for small alpha.
- The motivation of considering post-processing approaches seems not enough to me. Why do you select post-processing instead of pre/in-processing methods for the finite sample setting?

**Summary Of The Paper:**

The paper proposes FaiREE, a post-processing algorithm for fair classification with theoretical guarantees in a finite-sample and distribution-free manner. FaiREE can be applied to a wide range of group fairness notions, and is shown to achieve small mis-classification error while the fairness constraints. Numerical studies on both synthetic and real data shows the practical value of FaiREE in achieving a superior fairness-accuracy trade-off than the state-of-the-art methods.

**Summary Of The Review:**

The paper proposes FaiREE, a post-processing algorithm for fair classification with theoretical guarantees in a finite-sample and distribution-free manner. The approach is interesting that includes candidate set construction and selection. However, the empirical performance of FairREE does not have an advantage compared to baselines. Overall, I tend to reject the paper for the current version.

---

> ### Author Response · Authors · 2022-11-14
> **Response to Reviewer sJJW**
>
> Thank you for your valuable suggestions. We respond to your questions below and we would appreciate it if you could tell us whether our response answers your questions.
>
> >**Q1**: The empirical performance of FaiREE does not have an advantage compared to baselines. For example, in Figure 1, FairBayes can achieve higher accuracy than FaiREE for small alpha.
>
> **A1**: Thank you for your comment, but we are afraid there is a misunderstanding here. We would like to note that in Figure 1, FairBayes achieves higher accuracy than FaiREE because FairBayes is UNABLE to satisfy the fairness constraint with finite samples. In the left part of Figure 1, the FairBayes has a very poor performance in fairness for a small alpha, eg. when $\alpha=0.12$, the 95% quantile of DEOO for the FairBayes is around 0.3, much larger than the desired level 0.12, while FaiREE satisfies the fairness constraint very well.
>
> In addition, Table 3 actually shows our proposed algorithm has better accuracy: when the fairness violation of our method and the baseline methods are similar, eg. when $\alpha=0.16$, our method FaiREE-EO has the 95% quantiles of DEOO and DPE violations at 0.133 and 0.106 respectively, and at the same time achieve the accuracy 0.669. In contrast, the baseline method C-Eq has a similar fairness violation, but a worse accuracy at only 0.406.
>
>
> ---
>
> >**Q2**: The motivation of considering post-processing approaches seems not enough to me. Why do you select post-processing instead of pre/in-processing methods for the finite sample setting?
>
> **A2**: We would like to point out that to the best of our knowledge, our method is the **first** algorithm in the literature that can produce a classifier that satisfies fairness constraints with only finite sample and without making any distributional assumptions. As we pointed out in the literature review, the existing methods generally require an asymptotic setting (infinite sample size) and certain distributional assumptions such as normal distribution, bounded density, etc.
> We focus on post-processing because it is applicable to a broader classes of ML algorithms, especially algorithms where we don’t have whitebox access or ability to make modifications, which is the case for most ML methods deployed in practice.
>
>
> ---

---

> > ### Comment · Reviewer_sJJW · 2022-11-14
> > **Response to the authors**
> >
> > Thanks for the response.
> >
> > For Q1, I got the meaning. I think the original figure was confusing and you have changed Figure 1 in the revised pdf.
> >
> > For Q2, my point is that your motivation is "However, recent post-processing algorithms are found to lack the ability to realize accuracy–fairness trade-off and perform poorly when the sample size is limited", and hence, consider designing a post-processing algorithm with only finite samples. But why do you only discuss the ability to exist post-processing algorithms? If I do not misunderstand, the response seems to claim that all prior methods, including pre-processing, in-processing, and post-processing, can not satisfy fairness constraints with only finite samples (am I right?). Then for this stronger claim, I think you need to do a better discussion in the introduction/related section, and show some empirical proofs.

---

> > > ### Author Response · Authors · 2022-11-17
> > > **(2/2) Further Response to Reviewer sJJW**
> > >
> > > **Table R2**: Experimental studies under Model 1, with the same notation as Table R1.
> > >
> > > |           |$\mid$Fairdecision  |$\mid$LAFTR  |$\mid$Meta-cl |$\mid$Adv-debias      |    $\mid$   |FaiREE-EOO|    |    $\mid$  |FaiREE-EO|     |
> > > |------------|-------------|-------------------|------------------|----------------|---------------|-----------------|------------------|--------------|-------------|-------------------|
> > > |$\alpha$| $\mid\quad\quad\quad$/ | $\mid\quad$ / |  $\mid\quad\quad$       /   |  $\mid\quad\quad$/  |   $\mid$0.08  |   $\mid\quad\quad$0.12  |  $\mid$0.16  |  $\mid$0.08  |  $\mid\quad$0.12 | $\mid$0.16|
> > > |$\mid\overline{DEOO}\mid$| $\mid\quad\quad$0.072  |  $\mid$0.081  |  $\mid\quad$0.028   |  $\mid\quad$0.062  |   $\mid$0.028  |  $\mid\quad\quad$0.046 |  $\mid$0.063  |  $\mid$0.025  |   $\mid\quad$0.031  |  $\mid$0.042 |
> > > |${\|{DEOO}\|}_{95}$| $\mid\quad\quad$0.177   |   $\mid$0.145  |   $\mid\quad$0.108   |  $\mid\quad$0.226  |   $\mid$0.073  |  $\mid\quad\quad$0.115  |  $\mid$0.157  |  $\mid$0.079  |  $\mid\quad$0.108  |  $\mid$0.133  |
> > > |$\overline{\|DPE\|}$| $\mid\quad\quad\quad$/  |  $\mid$0.061  |  $\mid\quad$0.118  | $\mid\quad$0.179   |   $\mid\quad$/   |  $\mid\quad\quad$ /  | $\mid\quad$/  |  $\mid$0.039  |  $\mid\quad$0.042  |  $\mid$0.045|
> > > |${\|{DPE}\|}_{95}$| $\mid\quad\quad\quad$/  |  $\mid$0.104  |  $\mid\quad$0.272  |  $\mid\quad$0.412   |  $\mid\quad$/  |  $\mid\quad\quad$ /  |  $\mid\quad$/  |  $\mid$0.075  |  $\mid\quad$0.084   |  $\mid$0.106|
> > > |$\overline{ACC}$| $\mid\quad\quad$0.616  |  $\mid$0.533  |  $\mid\quad$0.620  |  $\mid\quad$0.645     |  $\mid$0.621   | $\mid\quad\quad$0.657  | $\mid$0.669  |  $\mid$0.552 |  $\mid\quad$0.562  | $\mid$0.615 |
> > >
> > > ---
> > >
> > > **Table R3**: Experimental studies under Model 2, with the same notation as Table R1.
> > >
> > > |           |$\mid$Fairdecision  |$\mid$LAFTR  |$\mid$Meta-cl |$\mid$Adv-debias      |    $\mid$   |FaiREE-EOO|    |    $\mid$  |FaiREE-EO|     |
> > > |------------|-------------|-------------------|------------------|----------------|---------------|-----------------|------------------|--------------|-------------|-------------------|
> > > |$\alpha$| $\mid\quad\quad\quad$/ | $\mid\quad$ / |  $\mid\quad\quad$       /   |  $\mid\quad\quad$/  |   $\mid$0.08  |   $\mid\quad\quad$0.12  |  $\mid$0.16  |  $\mid$0.08  |  $\mid\quad$0.12 | $\mid$0.16|
> > > |$\mid\overline{DEOO}\mid$| $\mid\quad\quad$0.675  |  $\mid$0.450  |  $\mid\quad$0.094   |  $\mid\quad$0.096  |   $\mid$0.027  |  $\mid\quad\quad$0.047 |  $\mid$0.073  |  $\mid$0.028  |   $\mid\quad$0.035  |  $\mid$0.047 |
> > > |${\|{DEOO}\|}_{95}$| $\mid\quad\quad$0.744   |   $\mid$0.633  |   $\mid\quad$0.208   |  $\mid\quad$0.263  |   $\mid$0.075  |  $\mid\quad\quad$0.112  |  $\mid$0.153  |  $\mid$0.077  |  $\mid\quad$0.114  |  $\mid$0.143  |
> > > |$\overline{\|DPE\|}$| $\mid\quad\quad\quad$/  |  $\mid$0.502  |  $\mid\quad$0.120  | $\mid\quad$0.140   |   $\mid\quad$/   |  $\mid\quad\quad$ /  | $\mid\quad$/  |  $\mid$0.041  |  $\mid\quad$0.044  |  $\mid$0.056|
> > > |${\|{DPE}\|}_{95}$| $\mid\quad\quad\quad$/  |  $\mid$0.686  |  $\mid\quad$0.312  |  $\mid\quad$0.418   |  $\mid\quad$/  |  $\mid\quad\quad$ /  |  $\mid\quad$/  |  $\mid$0.071  |  $\mid\quad$0.090   |  $\mid$0.127|
> > > |$\overline{ACC}$| $\mid\quad\quad$0.584  |  $\mid$0.647  |  $\mid\quad$0.606  |  $\mid\quad$0.628     |  $\mid$0.595   | $\mid\quad\quad$0.627  | $\mid$0.639  |  $\mid$0.575 |  $\mid\quad$0.589  | $\mid$0.606 |
> > >
> > > ---
> > >
> > > **Reference**
> > >
> > > [1] Madras, D., Creager, E., Pitassi, T., & Zemel, R. (2018, July). Learning adversarially fair and transferable representations. In International Conference on Machine Learning (pp. 3384-3393). PMLR.
> > >
> > > [2] Kilbertus, N., Rodriguez, M. G., Schölkopf, B., Muandet, K., & Valera, I. (2020, June). Fair decisions despite imperfect predictions. In International Conference on Artificial Intelligence and Statistics (pp. 277-287). PMLR.
> > >
> > > [3] Zhang, B. H., Lemoine, B., & Mitchell, M. (2018, December). Mitigating unwanted biases with adversarial learning. In Proceedings of the 2018 AAAI/ACM Conference on AI, Ethics, and Society (pp. 335-340).
> > >
> > > [4] Celis, L. E., Huang, L., Keswani, V., & Vishnoi, N. K. (2019, January). Classification with fairness constraints: A meta-algorithm with provable guarantees. In Proceedings of the conference on fairness, accountability, and transparency (pp. 319-328).

---

> > > > ### Author Response · Authors · 2022-11-18
> > > > **Follow-up**
> > > >
> > > > Dear sJJW,
> > > >
> > > > We would like to follow up to see if our response addresses your additional concerns or if you have any further questions. We would really appreciate the opportunity to discuss this further if our response has not addressed your concerns. Thank you!

---

> > > ### Author Response · Authors · 2022-11-17
> > > **(1/2) Further Response to Reviewer sJJW**
> > >
> > > Thank you for your further comments!
> > >
> > > We are glad that Fig 1 now becomes clear to you, and thank for your further clarification of Q2. We would like to remark that proving our algorithm guarantees fairness in a distribution-free and finite-sample manner is a major **theoretical** contribution of our paper, and to the best of our knowledge, we are the first paper in the algorithmic fairness literature to achieve this, as existing works do not have such theoretical guarantees.
> > >
> > > In light of your suggestion, we further compare FaiREE with more baseline methods that belong to pre-processing algorithms (including Fairdecision in [1] and LAFTR in [2]) and in-processing algorithms (including Meta-cl in [3] and Adv-debias in [4]), under synthetic settings Model 1 and Model 2 described in Section 5.1, and the real dataset Adult Census in Section 5.2. The results are summarized in the following tables, which were also updated in the revised pdf file.
> > >
> > > The results indicate that while the baseline methods are designed to minimize the fairness violation as much as possible (i.e. set alpha=0), these methods are unable to have exact control of the fairness violation to a desired level alpha. For example, in the analysis of Adult Census dataset, the 95% quantile of the DEOO fairness violations of Fairdecision is 0.078, and that of LAFTR, Meta-cl, and Adv-debias are all above 0.2. Moreover, our results found that If we allow the same fairness violation of DEOO and DPE for our proposed method FaiREE, we have a much higher accuracy (0.845) compared to the accuracy of those four baseline methods.
> > >
> > > ---
> > >
> > > **Table R1**: Results of different methods on Adult Census dataset. Here $\overline{|DEOO|}$ denotes the sample average of the absolute value of $DEOO$ defined in Eq.(1) in the paper, and $\mid DEOO\mid_{95}$ denotes the sample upper 95\% quantile. $\overline{|DPE|}$ and ${|DPE|}_{95}$ are defined similarly for $DPE$ defined in Eq. (13) in the paper. $\overline{ACC}$ is the sample average of accuracy. We use "/" in the $DPE$ line because Fairdecision and FaiREE-EOO are not designed to control $DPE$.
> > > |           |$\mid$Fairdecision  |$\mid$LAFTR  |$\mid$Meta-cl |$\mid$Adv-debias      |    $\mid$   |FaiREE-EOO|    |    $\mid$  |FaiREE-EO|     |
> > > |------------|-------------|-------------------|------------------|----------------|---------------|-----------------|------------------|--------------|-------------|-------------------|
> > > |$\alpha$| $\mid\quad\quad\quad$/ | $\mid\quad$ / |  $\mid\quad\quad$       /   |  $\mid\quad\quad$/  |   $\mid$0.07  |   $\mid\quad\quad$0.1  |  $\mid$0.14  |  $\mid$0.07  |  $\mid\quad$0.1 | $\mid$0.14|
> > > |$\mid\overline{DEOO}\mid$| $\mid\quad\quad$0.041  |  $\mid$0.124  |  $\mid\quad$0.172   |  $\mid\quad$0.199  |   $\mid$0.034  |  $\mid\quad\quad$0.039 |  $\mid$0.066  |  $\mid$0.002  |   $\mid\quad$0.039  |  $\mid$0.067 |
> > > |${\|{DEOO}\|}_{95}$| $\mid\quad\quad$0.078   |   $\mid$0.203  |   $\mid\quad$0.253   |  $\mid\quad$0.248  |   $\mid$0.065  |  $\mid\quad\quad$0.090  |  $\mid$0.124  |  $\mid$0.008  |  $\mid\quad$0.094  |  $\mid$0.125  |
> > > |$\overline{\|DPE\|}$| $\mid\quad\quad\quad$/  |  $\mid$0.044  |  $\mid\quad$0.194  | $\mid\quad$0.074   |   $\mid\quad$/   |  $\mid\quad\quad$ /  | $\mid\quad$/  |  $\mid$0.030  |  $\mid\quad$0.066  |  $\mid$0.074 |
> > > |${\|{DPE}\|}_{95}$| $\mid\quad\quad\quad$/  |  $\mid$0.083  |  $\mid\quad$0.271  |  $\mid\quad$0.094   |  $\mid\quad$/  |  $\mid\quad\quad$ /  |  $\mid\quad$/  |  $\mid$0.056  |  $\mid\quad$0.078   |  $\mid$0.086 |
> > > |$\overline{ACC}$| $\mid\quad\quad$0.772  |  $\mid$0.822  |  $\mid\quad$0.688  |  $\mid\quad$0.791     |  $\mid$0.845   | $\mid\quad\quad$0.846  | $\mid$0.847  |  $\mid$0.512 |  $\mid\quad$0.845  | $\mid$0.846 |
> > >
> > > ---

---

### Official Review · Reviewer_8rQh · 2022-11-03

**Confidence:** 4
**Correctness:** 3
**Technical Novelty And Significance:** 2
**Empirical Novelty And Significance:** 2
**Recommendation:** 3

**Clarity, Quality, Novelty And Reproducibility:**

Clarity: The main algorithm and its analysis were explained carefully in the paper. However, I thought that the author should have explained the experimental details in more detail. In particular, why does the accuracy become quite low on certain datasets for smaller values of $\alpha$? Moreover, it would also have been nice to see a pareto-curve showing the trade-off between fairness and accuracy.

Novelty: I think the idea of using threshold based classification for post-processing decisions of a classifier is quite well-established, and this is not a novel contribution. But the authors did provide an interesting application of the recent results on fair Bayes classification. I also did not understand the claim about distribution free approach. Since lemma 1 uses knowledge of posterior distribution, isn't this dependent on the underlying distribution?

**Strength And Weaknesses:**

Strengths:
- The proposed method is quite general as a post-processing method and probably can be adapted for other fairness measures as well. I also like proposition 1 which provides a bound on the fairness violation of a score-based classifier given a threshold.
- The experimental results show improved fairness guarantees compared to other post-processing based fair classifiers.

Weaknesses:
  - I think the proposed bound only holds when the dataset is not imbalanced. This is evident in the statement of theorem 2 where the bound scales with the minimum number of samples across different group, outcome pairs. Since most of the datasets encountered in fair classification are under-represented, this is a major concern.
- The main weakness of the paper is experimental evaluation. Table 3 shows improvement in fairness but for smaller values of $\alpha$ (e.g. 0.08 or 0.12) accuracy is significantly lower for FairEE-EO. The same observation holds for the Compass dataset (in the appendix). This makes me think that the proposed algorithm doesn’t provide right trade-off between accuracy and fairness.
- Most of the guarantees are in-sample fairness guarantees. The authors didn’t provide generalization guarantees even though the lack of generalization of  post-processing based methods was emphasized in the motivation.
- Finally, in order to implement the candidate set construction efficiently, one needs to implement functions $g_1$ (from proposition 1) efficiently. It was not mentioned how these statistics can be evaluated efficiently.

**Summary Of The Paper:**


This paper considers the problem of fair classification by post-processing the decisions of an existing (possibly unfair) classifier. The authors focus on two metrics — equality of opportunity and equalized odds. The main idea is to set the decision threshold appropriately based on the protected group and true outcome.

The proposed algorithm consists of two main steps — (a) candidate set construction, and (b) candidate selection. For candidate set construction, the authors first provide a bound that measures the probability that the fairness constraint is violated for a given threshold. Based on this lemma and a recent result from fair Bayes classification, the algorithm selects a set of candidate threshold pairs. In the next step, the threshold pair with the best empirical performance is selected as the final pair of thresholds.

The proposed algorithm is evaluated on two simulated datasets, and several standard datasets on fair classification and compared against existing post-processing based fair classifiers. The results show that the proposed algorithm (FaiR-EE) has improved fairness guarantee, but it doesn’t seem to provide any Pareto improvement in terms of the fairness-accuracy trade-off.

**Summary Of The Review:**

Overall, I think the paper considers an interesting approach for designing post-processing based fair classifier. However, the main proposed algorithm has some requirements (e.g. balanced data, knowledge of $g_1$, etc). Moreover, the experimental section seems weak and suggests that the proposed approach doesn't provide right trade-off between fairness and accuracy.

---

> ### Author Response · Authors · 2022-11-14
> **(2/2) Response to Reviewer 8rQh**
>
> >**Q5**:  The main algorithm and its analysis were explained carefully in the paper. However, I thought that the author should have explained the experimental details in more detail. In particular, why does the accuracy become quite low on certain datasets for smaller values of α? Moreover, it would also have been nice to see a pareto-curve showing the trade-off between fairness and accuracy.
>
> **A5**: The accuracy becomes quite low for smaller values of alpha is a natural phenomenon in algorithmic fairness: smaller alpha means a more stringent fairness constraint, implying fewer candidate functions we can choose and therefore lower the accuracy. This phenomenon may be exhibited differently in different datasets because different datasets have different data distributions and the unconstrained Bayes-optimal solution for some datasets may violate the fairness constraints at different levels. If the unconstrained Bayes-optimal solution has a small violation, then after the post-processing of our algorithm, the accuracy can remain high; on the other hand, if the unconstrained optimal solution has a severe fairness violation, after we map it to a solution that satisfies the fairness constraints, the solution generally produces low accuracy.
>
> In addition, we have added two plots showing the trade-off between fairness and accuracy in the appendix (Page 31) and also combined them into one figure to substitute the right part of Figure 1 and thus make it more readable. Currently, the left part of Figure 1 is DEOO v.s. $\alpha$ for FaiREE and Fairbayes (DEOO is rigorously controlled by FaiREE under different $\alpha$), and the right part is DEOO v.s. test accuracy for FaiREE and Fairbayes (tradeoff can be directly observed).
>
> ---
>
> >**Q6**: I think the idea of using threshold based classification for post-processing decisions of a classifier is quite well-established, and this is not a novel contribution. But the authors did provide an interesting application of the recent results on fair Bayes classification. I also did not understand the claim about distribution free approach. Since lemma 1 uses knowledge of posterior distribution, isn't this dependent on the underlying distribution?
>
> **A6**: We would like to point out that to the best of our knowledge, our method is the **first** algorithm in the literature that can produce a classifier that satisfies fairness constraints with only finite sample and without making any distributional assumptions. As we pointed out in the literature review, the existing methods generally require an asymptotic setting (infinite sample size) and certain distributional assumptions such as normal distribution, bounded density, etc.
> The term "distribution-free" refers to the case where our algorithm does not require assumptions for the distribution of the data (also see the remark at the bottom of Page 6 in the paper). This term has been commonly used in the machine learning literature, such as [1, 2, 3]. To address your concern, we further clarify the meaning of "distribution-free" in the introduction (Section 1, Paragraph 3, Line 2).
>
> ---
>
> **Reference**
>
> [1] Maritz, J. S. (1995). Distribution-free statistical methods (Vol. 17). CRC Press.
>
> [2] Clarke, K. A. (2007). A simple distribution-free test for nonnested model selection. Political Analysis, 15(3), 347-363.
>
> [3] Györfi, L., Kohler, M., Krzyzak, A., & Walk, H. (2002). A distribution-free theory of nonparametric regression (Vol. 1). New York: Springer.

---

> > ### Author Response · Authors · 2022-11-17
> > **Follow-up**
> >
> > Dear Reviewer 8rQh,
> >
> > We would like to follow up to see if our response addresses your concerns or if you have any further questions. We would really appreciate the opportunity to discuss this further if our response has not addressed your concerns. Thank you!

---

> ### Author Response · Authors · 2022-11-14
> **(1/2) Response to Reviewer 8rQh**
>
> Thank you for reviewing our paper and for your valuable feedback. Below, we address your concerns point by point and we’ve revised our paper according to your suggestions. We would appreciate it if you could let us know whether your concerns are addressed by our response.
>
> >**Q1**: I think the proposed bound only holds when the dataset is not imbalanced. This is evident in the statement of theorem 2 where the bound scales with the minimum number of samples across different group, outcome pairs. Since most of the datasets encountered in fair classification are under-represented, this is a major concern.
>
> **A1**: We are sorry to hear your misunderstanding. Our theorem 2 only requires a minimum sample size assumption min\{$n^{1,0}, n^{1,1}$\} $\geq \lceil \frac{\log (\delta / 2)} {\log (1-\alpha)}\rceil$, where $\delta$ and $\alpha$ are two adjustable parameters for the algorithm. As a concrete example, if we choose $\delta$ = 0.1 and $\alpha$ = 0.1, then the minimum size requirement for n ^{1,0} and n^{1,1} is simply $log(\delta / 2) / log(1 - \alpha) \approx$ 29 for FaiREE-EOO, which means our proposed algorithm works even for heavily imbalanced data such as n^{1,0} = 30 and n^{1, 1} = 2000.
>
> ---
>
> >**Q2**: The main weakness of the paper is experimental evaluation. Table 3 shows improvement in fairness but for smaller values of α (e.g. 0.08 or 0.12) accuracy is significantly lower for FairEE-EO. The same observation holds for the Compass dataset (in the appendix). This makes me think that the proposed algorithm doesn’t provide right trade-off between accuracy and fairness.
>
> **A2**: We would like to point out that the accuracy of FairEE-EO is worse than the baseline methods because the baseline methods are UNABLE to control fairness with finite samples. As illustrated in our Table 3, the three baseline methods, Eq, C-Eq, and ROC all have large DEOO: the 95% quantile of their DEOO is at least 0.146, and that of ROC is even as large as 0.5, while our proposed algorithms all control the fairness constraints at desired levels.  Therefore, it is very natural that our accuracy is lower than those baseline methods when $\alpha$ is smaller because our algorithm satisfies the fairness constraint exactly and exhibits the accuracy-fairness trade-off. In addition, Table 3 actually shows our proposed algorithm has better accuracy: when the fairness violation of our method and the baseline methods are similar, eg. when $\alpha=0.16$, our method FaiREE-EO has the 95% quantiles of DEOO and DPE violations at 0.133 and 0.106 respectively, and at the same time achieve the accuracy 0.669. In contrast, the baseline method C-Eq has a similar fairness violation, but a worse accuracy at only 0.406.
>
>
> ---
>
> >**Q3**: Most of the guarantees are in-sample fairness guarantees. The authors didn’t provide generalization guarantees even though the lack of generalization of post-processing based methods was emphasized in the motivation.
>
> **A3**:  We are afraid this is another misunderstanding. Our whole paper considers the out-of-sample fairness guarantees. In our theorem statement for fairness guarantees, e.g., Theorem 2.(1), the quantity of focus is DEOO. The DEOO is defined in Eq. (1), and the probabilities in the definition are taken over the test distribution $X\mid A=1, Y=1$ and $X\mid A=0, Y=1$ respectively. In addition, our theorem statement for accuracy is also for test distributions. For example, in Theorem 2.(2), the error term we consider is the probability that the output classifier of FaiREE makes an incorrect prediction on the test set. More specifically, our theoretical guarantee is established for the test error $P(\hat{\phi}(x,a) \neq Y)$, where $\hat{\phi}$ is the output classifier of FaiREE, and Y is the label in the test distribution.
>
> ---
>
> >**Q4**: Finally, in order to implement the candidate set construction efficiently, one needs to implement functions $g_1$(from proposition 1) efficiently. It was not mentioned how these statistics can be evaluated efficiently.
>
> **A4**: We would like to note that $g_1$ can be implemented efficiently, as it only requires generating a few random numbers. More specifically, the expectation in $g_1$ can be efficiently implemented by Monte Carlo simulations and random sampling from a Beta distribution. In our experiments, only 1000 times of sampling is sufficient to obtain a very high approximation accuracy, which on average takes 0.15 seconds on 12th Gen Intel Core i9 (laptop) with 14 cores to compute under the setting of 1000 sample size.
>
> ---

---

### Official Review · Reviewer_P2N4 · 2022-11-04

**Confidence:** 3
**Correctness:** 4
**Technical Novelty And Significance:** 3
**Empirical Novelty And Significance:** 3
**Recommendation:** 6

**Clarity, Quality, Novelty And Reproducibility:**

The writing quality of this paper can be significantly improved. Here I just listed some:

- Page 1, paragraph 1: “To address this fairness problem, many fairness notions have been proposed.” Are these fairness notions proposed to just formulate the fairness measures or mitigate the unfairness?
- Page 1, paragraph 1: “Based on these fairness notions or constraints, corresponding algorithms…… Among these fairness algorithms…” Where are the examples?
- Page 1, paragraph 2: “most fairness constraints are non-convex, some papers propose convex relaxation-based methods” The convex relaxations are designed for in-processing mitigation algorithms. How did those relaxations motivate your research problem?
- Page 2, paragraph 1: achieve group fairness notions -> achieve group fairness constraints; fairness classification -> fair classification
- Page 1, Figure 1: consider combine the subfigures into one trade-off graphic for accuracy and fairness violation
- Page 2, paragraph 3: “grup fairness” typo
- Proposition 2, $\phi \neq y$ is ill-defined, as $\phi(x, a)$ has appeared in Proposition 1.

**Strength And Weaknesses:**

Strengths:
- The scope and motivation of this paper is very clear. The proposed algorithm seems to be a novel work in post-processing fairness mitigations.
- The authors demonstrate how the proposed algorithm can be extended to multi-group fairness notions, which will be useful in practice.
- The proposed algorithm has strong theoretical guarantee to bound the error of post-processed classifier.

Weaknesses:
- The writing of this paper is not in a good shape. There are many typos. The wordings are sometimes confusing. The heavy notations make it not easy to understand the paper. In this sense, I don't think this paper is ready for publication. Please see the comments below.

**Summary Of The Paper:**

This paper aims to address one challenge in post-processing fairness mitigation algorithms: they require either distributional assumptions or access to infinite data examples. This paper presents a framework that takes in any score-based black-box classifier and outputs group-wise thresholds to correct for the fairness violations. The advantage over existing literature is it doesn’t require additional assumption on data distribution except for a minimum sample size.

There are three steps in the FaiREE framework. First, score each data example using the given black-box classifier. Second, conform a candidate set of thresholds that promise the fairness constraints. Finally, select the classifier that has the smallest mis-classification error from the candidate set. The key improvement is that the authors provide an upper bound of the probability of fairness violations and lend the upper bound to shrink the candidate set.

The FaiREE framework can be extended to multi-group fairness notions. Experiments are conducted on both a synthetic dataset and the Adult dataset to validate the effectiveness of the post-processing algorithm.

**Summary Of The Review:**

I did not have the time to go over any of the proofs, but I do have some concerns that this paper is not ready for publication. I would be open to adjust my score after receiving author's response.

Questions:
1. Does Theorem 2 require the input classifier f to be Bayes-optimal? If true, how practical is this additional assumption?
2. The big-$O$ notation after the w.p. phrases in Theorem 2 and Theorem 4 does not make sense to me. How do you interpret the big-O inside a probability?
3. In the experiments, the baseline methods are compared on fairness notions of which the algorithms are not designed to mitigate the violations. For example, Eq is compared on predictive equality and equalized accuracy constraints. Is it a fair comparison?
4. In Table 2 and Table 3, the accuracy of classifiers are around 0.5. Is that informative on the synthetic dataset?

---------
The authors have properly addressed most of my concerns and made significant improvements over the draft. In consequence, I updated the score from 5 to 6 accordingly.

---

> ### Author Response · Authors · 2022-11-14
> **(2/2) Response to Reviewer P2N4**
>
> >**Q8**: The big-O notation after the w.p. phrases in Theorem 2 and Theorem 4 does not make sense to me. How do you interpret the big-O inside a probability?
>
>  **A8**: We would like to note that big-O after the w.p. phrases is commonly used in theoretical machine learning papers, such as [1, 2, 3] (e.g., see Lemma 2.1 in [1], Theorem 1.6 in [2] and Theorem 3 in [3]). In light of your suggestions, we modified the statement of the theorems without using the notation of big-O. For example, in current Theorem 2, we write "there exists $c_1 > 0$, such that …with probability larger than $1 - c_1 \cdot c^{\min(n^{1,0}, n^{0,0}, n^{0,1})}$".
>
> ---
>
> >**Q9**: In the experiments, the baseline methods are compared on fairness notions of which the algorithms are not designed to mitigate the violations. For example, Eq is compared on predictive equality and equalized accuracy constraints. Is it a fair comparison?
>
>  **A9**: In our experiments, the baseline method Eq [4] is designed to fit the fairness notion Equalized Odds, which requires both Equality of Opportunity (equal false negative rate across different groups, the violation to which is denoted by DEOO) and Predictive Equality (equal false positive rate across different groups, the violation to which is denoted by DPE). Our FaiREE-EO is designed for the same purpose.
> We didn’t compare equalized accuracy for Eq in our experiments.
>
> ---
>
> >**Q10**: In Table 2 and Table 3, the accuracy of classifiers are around 0.5. Is that informative on the synthetic dataset?
>
>  **A10**: Thank you for your comment. In the revision, we added another set of synthetic dataset analysis, which shows that our algorithm can perform better than other methods when the dataset is informative (i.e., has high accuracy). The results are summarized below.
>
> **Table R1**: Experimental studies under Model 3. Here $\overline{|DEOO|}$ denotes the sample average of the absolute value of $DEOO$ defined in Eq.(1) in the paper, and $\mid DEOO\mid_{95}$ denotes the sample upper 95\% quantile. $\overline{|DPE|}$ and ${|DPE|}_{95}$ are defined similarly for $DPE$ defined in Eq. (13) in the paper. $\overline{ACC}$ is the sample average of accuracy. We use "/" in the $DPE$ line because FairBayes and FaiREE-EOO are not designed to control $DPE$.
>
>
> |           |$\mid$   Eq   | $\mid$C-Eq  | $\mid$ROC |  $\mid$        | FairBayes |         |    $\mid$   |FaiREE-EOO|   |    $\mid$ |FaiREE-EO|     |
> |------------|------------|------------|------------|--------|---------|---------|---------|-----|---------|---------|---------|---------|
> |$\alpha$| $\mid\quad$   / | $\mid\quad$ / |  $\mid\quad$       /   |  $\mid$0.04  | $\mid\quad$0.06  |   $\mid$0.08  |   $\mid$0.04  |   $\mid\quad\quad$0.06  |  $\mid$0.08  |  $\mid$0.04  |  $\mid\quad$0.06 | $\mid$0.08|
> |$\mid\overline{DEOO}\mid$| $\mid$0.041  |  $\mid$0.020  |  $\mid$0.034   |  $\mid$0.048  |  $\mid\quad$0.062   |   $\mid$0.076  |   $\mid$0.018  |  $\mid\quad\quad$0.025 |  $\mid$0.033  |  $\mid$0.015  |   $\mid\quad$0.032  |  $\mid$0.034 |
> |${\|{DEOO}\|}_{95}$| $\mid$0.115   |   $\mid$0.042  |   $\mid$0.070   |  $\mid$0.080 | $\mid\quad$0.113  |  $\mid$0.143  |   $\mid$0.038  |  $\mid\quad\quad$0.054  |  $\mid$0.076  |  $\mid$0.036  |  $\mid\quad$0.058  |  $\mid$0.069  |
> |$\overline{\|DPE\|}$| $\mid$0.093  |  $\mid$0.106  |  $\mid$0.062  | $\mid\quad$/  |  $\mid\quad\quad$/  |  $\mid\quad$/  |   $\mid\quad$/   |  $\mid\quad\quad$ /  | $\mid\quad$/  |  $\mid$0.026  |  $\mid\quad$0.034  |  $\mid$0.046|
> |${\|{DPE}\|}_{95}$| $\mid$0.272  |  $\mid$0.191  |  $\mid$0.101  |  $\mid\quad$/  |  $\mid\quad\quad$/  |  $\mid\quad$/  |  $\mid\quad$/  |  $\mid\quad\quad$ /  |  $\mid\quad$/  |  $\mid$0.039  |  $\mid\quad$0.055   |  $\mid$0.077|
> |$\overline{ACC}$| $\mid$0.887  |  $\mid$0.921  |  $\mid$0.834  |  $\mid$0.898   |  $\mid\quad$0.901   |  $\mid$0.912   |  $\mid$0.946   | $\mid\quad\quad$0.950  | $\mid$0.963  |  $\mid$0.900  |  $\mid\quad$0.914  | $\mid$0.933|
>
> ---
>
>
> **Reference**
>
> [1] Arora, S., Li, Y., Liang, Y., Ma, T., & Risteski, A. (2016). A latent variable model approach to pmi-based word embeddings. Transactions of the Association for Computational Linguistics, 4, 385-399.
>
> [2] Chen, Y., Fan, J., Ma, C., & Yan, Y. (2021). Bridging convex and nonconvex optimization in robust PCA: Noise, outliers and missing data. The Annals of Statistics, 49(5), 2948-2971.
>
> [3] Tripuraneni, N., Jin, C., & Jordan, M. (2021, July). Provable meta-learning of linear representations. In International Conference on Machine Learning (pp. 10434-10443). PMLR.
>
> [4] Hardt, M., Price, E., & Srebro, N. (2016). Equality of opportunity in supervised learning. Advances in neural information processing systems, 29.

---

> > ### Author Response · Authors · 2022-11-17
> > **Follow-up**
> >
> > Dear Reviewer P2N4,
> >
> > We would like to follow up to see if our response addresses your concerns or if you have any further questions. We would really appreciate the opportunity to discuss this further if our response has not addressed your concerns. Thank you!

---

> > > ### Comment · Reviewer_P2N4 · 2022-11-18
> > > **Thanks for the author response.**
> > >
> > > I took a look at the highlighted revision. I found that most of my major concerns have been corrected in the updated draft. I will update my rating accordingly.

---

> > > > ### Author Response · Authors · 2022-11-18
> > > > **Thank you**
> > > >
> > > > Thank you for your reply! We are happy to see that our response address most of your concerns. Thanks again for your valuable feedback to help us improve our paper and for being willing to update your rating.

---

> > > > > ### Comment · Reviewer_P2N4 · 2022-12-04
> > > > > **The rating is updated**
> > > > >
> > > > > I am not doing so because I was waiting the other reviewers' responses.

---

> > > > > > ### Author Response · Authors · 2022-12-04
> > > > > > **Thank you**
> > > > > >
> > > > > > Thank you for updating the score, and thank you again for your insightful comments and valuable feedback!

---

> > > > ### Author Response · Authors · 2022-12-04
> > > > **A gentle ping**
> > > >
> > > > Dear Reviewer P2N4,
> > > >
> > > > We wanted to send a gentle reminder as the discussion stage is about to end in a week (we understand this is a busy time of year). If our response has addressed your concerns, we would appreciate it if you could update your score as you kindly offered to do. If not, please feel free to ask us additional questions and we are happy to answer them!

---

> ### Author Response · Authors · 2022-11-14
> **(1/2) Response to Reviewer P2N4**
>
> Thank you for your comprehensive review and your valuable feedback to help us improve our paper. We have revised our paper based on your feedback. We have also improved the writing and corrected typos in the paper. We detail our response below and please kindly let us know if our response addresses your concerns.
>
> > **Q1**: Page 1, paragraph 1: "To address this fairness problem, many fairness notions have been proposed." Are these fairness notions proposed to just formulate the fairness measures or mitigate the unfairness?
>
> **A1**: These fairness notions are proposed to quantify certain fairness measures, based on which optimization methods are then used to mitigate the unfairness. In light of your comments, we have changed the sentence to "To quantify the fairness in machine learning algorithms, many fairness notions have been proposed…"
>
> ---
>
> > **Q2**: Page 1, paragraph 1: "Based on these fairness notions or constraints, corresponding algorithms…… Among these fairness algorithms…" Where are the examples?
>
> **A2**: Thank you for your comments. In our original submission, we cited related papers in the "Additional Related Works" section. We have added the citations and examples to this paragraph as you suggested.
>
> ---
>
> > **Q3**: Page 1, paragraph 2: "most fairness constraints are non-convex, some papers propose convex relaxation-based methods" The convex relaxations are designed for in-processing mitigation algorithms. How did those relaxations motivate your research problem?
>
> **A3**: As we mentioned in the sample paragraph, the convex-relaxation-based algorithms in the literature only satisfy the relaxed (convex) constraints, and they generally do not have the theoretical guarantee of how the output satisfies the exact original fairness constraint. In our paper, our proposed algorithm is provable to satisfy the desired fairness constraint with only finite sample and without distributional assumptions (i.e., distribution-free).
>
>
>
> ---
>
> > **Q4**: Page 2, paragraph 1: achieve group fairness notions -> achieve group fairness constraints; fairness classification -> fair classification;  Page 2, paragraph 3: "grup fairness" typo
>
> **A4**: Thank you for pointing these out. We have modified these typos.
>
> ---
>
> > **Q5**: Page 1, Figure 1: consider combining the subfigures into one trade-off graphic for accuracy and fairness violation
>
> **A5**: Thank you for this suggestion. We have added two subfigures showing the trade-off between fairness and accuracy in the appendix (Page 31) and also combined them into one figure to substitute the right part of Figure 1 for it to be clearer. Currently, the left part of Figure 1 is DEOO v.s. $\alpha$ for FaiREE and Fairbayes (DEOO is rigorously controlled by FaiREE under different $\alpha$), and the right part is DEOO v.s. test accuracy for FaiREE and Fairbayes (tradeoff can be directly observed).
>
> ---
>
> >**Q6**: Proposition 2, $\phi_i \neq y$ is ill-defined, as $\phi(x,a)$ has appeared in Proposition 1.
>
> **A6**: In our original submission, we defined $\phi_i$ in the paragraph before Proposition 2. In light of your comment, we have modified the notation and written $\phi_i$ as $\phi_i(x,a)$ to make the notation clearer.
>
> ---
>
> >**Q7**: Does Theorem 2 require the input classifier f to be Bayes-optimal? If true, how practical is this additional assumption?
>
>  **A7**: Thank you for pointing this out. Our analysis can be directly extended to the case where the input is close to the Bayes-optimal. We have modified the statement and proof for both Theorems 2 and 4. Please see Pages 6 and 7 of the revised paper.
>
> ---

---

### Author Response · Authors · 2022-11-14
**Summary of Paper Revision**

We sincerely appreciate all reviewers for their time and their insightful and constructive feedback. We have carefully updated the paper to address all of the reviewers’ comments. Please see the newly uploaded file, where the changes in the text are highlighted in orange. Below, we summarize the key changes:

1. Improved part (2) of Theorem 2 so that we can bound the performance of the proposed algorithm with any classifier f using the approximation error between $f$ and $f^{*}$, which makes this Theorem much more practical. (Reviewers P2N4 and 6zkG)
2. Added new synthetic experimental results by generating a new dataset on which trained classifiers are more informative, and showed that FaiREE has favorable performance on the new dataset. (Reviewer P2N4)
3. Added citations between "Based on these fairness notions or constraints, corresponding algorithms……" and "Among these fairness algorithms…" to show examples of the previous algorithms. (Reviewer P2N4)
4. Added two subfigures of trade-off graphic for accuracy and fairness violation and combined them into one figure to substitute the right part of Figure 1 to make it more readable and understandable. (Reviewers P2N4, 8rQh and sJJW)
5. Visualized all the tables in the paper using fairness-accuracy trade-off plots. (Reviewer 6zkG)
6. Make clearer the definition of big-O in Theorem 2 and the term "distribution-free" (Reviewers P2N4 and 8rQh); make clearer the meaning of the minimum sample size requirements in Table 2 (Reviewer 6zkG).
7. Fixed typos.

---

### Decision · Program_Chairs · 2023-01-20

**Decision:**

Accept: poster

**Justification For Why Not Higher Score:**

The presentation of the paper is to be improved (there are a number of typos), and the experimental evaluation looks weak.

**Justification For Why Not Lower Score:**

The proposed framework is clearly presented, seems to be novel, and has strong theoretical guarantees to bound the error of the post-processed classifier.

**Metareview: Summary, Strengths And Weaknesses:**

This paper proposes a framework to address a challenge in post-processing fairness mitigation algorithms, which require either distributional assumptions or access to infinite data examples. The proposed framework can take in any score-based black-box classifier and outputs group-wise thresholds to correct fairness violations. The proposed framework is clearly presented, seems to be novel, and has strong theoretical guarantees to bound the error of the post-processed classifier. On the other hand, the presentation is to be improved (there are a number of typos), and the experimental evaluation looks weak.

**Note From Pc:**

if the above contains the word "oral" or "spotlight" please see: "oral" presentation means -> notable-top-5% and "spotlight" means -> notable-top-25%. As stated in our emails, we are disassociating presentation type from AC recommendations